



# The influence of Atlantic climate variability on the long-term development of Mediterranean cold-water coral mounds (Alboran Sea, Melilla Mound Field)

Robin Fentimen[1], Eline Feenstra[1], Andres Rüggeberg[1], Efraim Hall[1], Valentin Rime[1], Torsten Vennemann[2], Irka Hajdas[3], Antonietta Rosso[4], David Van Rooij[5], Thierry Adatte[2], Hendrik Vogel[6], Norbert Frank[7], Thomas Krengel[7], Anneleen Foubert[1]

[1] Department of Geosciences, University of Fribourg, Fribourg, CH-1700, Switzerland
[2] Institute of Earth Surface Dynamics, University of Lausanne, Lausanne, CH-1015, Switzerland
[3] Laboratory of Ion Beam Physics, ETH Zürich, Zürich, CH-8093, Switzerland
[4] Department of Biological, Geological and Environmental Sciences, University of Catania, Catania, 95128, Italy
[5] Department of Geology, Ghent University, Ghent, 9000, Belgium
[6] Institute of Geological Sciences and Oeschger Centre for Climate Change Research, University of Bern, Bern, CH-3012, Switzerland
[7] Institute of Environmental Physics, University of Heidelberg, Heidelberg, D-69120, Germany

*Correspondence to*: Robin Fentimen (robin.fentimen@unifr.ch)

**Abstract.** This study provides a detailed reconstruction of climatic events affecting a cold-water coral mound located within the East Melilla Coral Province (Southeast Alboran Sea) over the last 300 ky. Based on benthic foraminiferal assemblages, macrofaunal quantification, grain size analysis, sediment geochemistry, and foraminiferal stable isotope compositions, a reconstruction of environmental conditions prevailing in the region is proposed. The variations in planktonic and benthic $\delta^{18}O$ values indicate that cold-water coral mound formation follows global climatic variability. Cold-water corals develop during both interglacial and glacial periods, although interglacial conditions would have allowed better proliferation. Environmental conditions during glacial periods, particularly during the Last Glacial Maximum, appear to better suit the ecological requirements of the erect cheilostome bryozoan *Buskea dichotoma*. Benthic foraminiferal assemblages suggest that high organic carbon flux characterized interglacial periods. Results from this study imply that increased influence of warm and moist Atlantic air masses during interglacial periods led to increased fluvial discharge, providing nutrients for cold-water corals. Important interglacial Atlantic Water mass inflow further promoted strong Alboran Gyres, and thus mixing between surface and intermediate water masses. Increased turbulence and nutrient supply would have hence provided suitable conditions for coral development. In contrast, benthic foraminiferal assemblages and grain size distributions suggest that the benthic environment received less organic matter during glacial periods, whilst bottom flow velocity was reduced in comparison to interglacial periods. During glacial periods, arid continental conditions combined to more stratified water masses caused a dwindling of coral communities in the southeastern Alboran Sea, although aeolian dust input may have allowed these to survive. In contrast to Northeast Atlantic counterparts, coral mound build-up in the southeastern Alboran Sea occurs during glacial as well as during interglacial periods and at very low aggradation rates (between 1 and 9 cm.ky$^{-1}$). We propose that *Buskea dichotoma* plays an important role in long-term mound formation at the East Melilla Coral Province, noticeably during glacial periods.





## 1. Introduction

Cold-water coral (CWC) reefs are diverse ecosystems that are common on Earth (Freiwald et al., 2004; Roberts et al., 2009). The most important reef building CWC species in the Atlantic Ocean and Mediterranean Sea are the
scleractinians *Desmophyllum pertusum* (formerly known as *Lophelia pertusa*, see Addamo et al., 2016) and *Madrepora oculata* (Roberts et al., 2009). These predominantly suspension-feeding organisms depend on nutrient supply and enhanced hydrodynamic regimes (White et al., 2005; Mienis et al., 2007; Carlier et al., 2009; Davies et al., 2009; Roberts et al., 2009; Hanz et al., 2019). The role of internal waves (i.e. waves that occur at the interface between two water masses of different densities) on the proliferation of CWCs is important, since these oscillations
increase turbulence and hence nutrient supply (White et al., 2005; Davies et al., 2009; Pomar et al., 2012; Wang et al., 2019). Physico-chemical properties of the ambient water (e.g. salinity, temperature, dissolved oxygen concentrations, pH, density) also affect CWC growth (Freiwald et al., 2004; Dullo et al., 2008; Davies and Guinotte, 2011; Hanz et al., 2019). If favourable conditions are maintained over longer periods, successive reef generations build CWC mounds through the interaction between coral growth and sediment accumulation (Wilson, 1979;
Roberts et al., 2006; Foubert and Henriet, 2009; Roberts et al., 2009). Consequently, CWC mounds can reach considerable heights of over 300 m and spread for kilometres in width and length at their base (De Mol et al., 2002; Kenyon et al., 2003; Huvenne et al., 2005). Mound development may span from thousands to millions of years and attain important mound aggradation rates, e.g 290 cm.ky$^{-1}$ in the Porcupine Seabight (Frank et al., 2009; López Correa et al., 2012; Stalder et al., 2015; Wienberg et al., 2018). As such, CWC mounds are valuable environmental
and climatic archives, although mound formation is generally discontinuous (Rüggeberg et al., 2007; Roberts et al., 2009). Moreover, the sensitivity of CWCs to climate change renders them useful to monitor variations in environmental conditions (e.g., water mass variability, surface productivity, bottom current velocity; Rüggeberg et al., 2007; Huvenne et al., 2009; Hebbeln et al., 2016; Wienberg et al., 2018).

The long-term development of CWC mounds was first studied in the Northeast Atlantic Ocean, where it is recognized to be driven by large-scale changes in oceanographic conditions (e.g. Dorschel et al., 2005; Frank et al., 2011, Wienberg et al., 2018). Corals along the Irish margin grow during interglacial and interstadial times, whilst their development declines during glacial periods (Dorschel et al., 2005; Kano et al., 2007; Rüggeberg et al., 2007; Eisele et al., 2008). Cold-water coral mound development along the Irish margin depends on the strength of the
Mediterranean Outflow Water (MOW) and the influence of internal waves (Mohn et al., 2014; Raddatz et al., 2014; Hebbeln et al., 2016). The strong influence of MOW during interglacial and interstadial times and the resulting enhanced turbulence induced by internal waves provides the correct balance between nutrient and sediment supply (Mohn et al., 2014; Raddatz et al., 2014). In contrast, during glacial times, weak MOW flow lowers nutrient supply and increases sediment smothering, causing coral retreat (Dorschel et al., 2005; Rüggeberg et al., 2007; Mohn et al.,
2014). In the Northwest Atlantic Ocean, CWC mounds also form during interglacial periods, when stronger hydrodynamic regimes and better-oxygenated waters dominate the region (Matos et al., 2015; 2017). At lower latitudes in the East Atlantic, off the coast of Mauritania and in the Gulf of Cádiz, coral mounds form essentially




during glacial times (Wienberg et al., 2009; Eisele et al., 2011), although they also developed at lower aggradation rates during the last interglacial (Marine Isotope Stage 5; Wienberg et al. 2018).


In the Mediterranean Sea, CWC mound provinces are mostly concentrated in the Alboran Sea, along the Moroccan margin (Fink et al., 2013; 2015; Lo Iacono et al., 2014; Stalder et al., 2015; 2018; Terhzaz et al., 2018; Wang et al., 2019; Rachid et al., 2020). The largest CWC mound field in this region is the Melilla Mound Field, covering an area greater than 500 km$^2$ parallel to the margin (Comas and Pinheiro, 2010; Lo Iacono et al., 2014). It is divided into two

provinces, the West and East Melilla Coral Provinces, respectively situated 7 km northwest and 35 km northeast of the Cape Tres Forcas (Hebbeln, 2019; Fig. 1). Several recent studies suggest an environmental forcing on CWC mound formation during the last 30 ky at the Melilla mounds (Fink et al., 2013; Stalder et al., 2015; 2018; Wang et al., 2019; Feenstra, 2020). Mound aggradation rates reach their highest values (75-420 cm.ky$^{-1}$) during the Early Holocene and Bølling-Allerød interstadial. In contrast, mound formation halted during the Younger-Dryas,

demonstrating low mound aggradation rates (30-50 cm.ky$^{-1}$; Fink et al., 2013; Stalder et al., 2015; Wang et al., 2019; Feenstra, 2020). Based on benthic foraminiferal assemblages, Stalder et al. (2015) suggest that cold/dense well oxygenated bottom water conditions favoured CWC development, whilst Wang et al. (2019) relate the intensified coral proliferation to high surface productivity combined with strong turbulence induced by internal waves.

Although the development of the East Melilla mounds during the last 30 ky is well documented, the long-term development and environmental forcing affecting these mounds remain unknown. The aims of this study are: 1) to constrain the influence of climate variability on mound formation in the East Melilla region over the last 300 ky, and 2) to assess long-term coral mound formation in the area and compare it to North Atlantic counterparts.

## 2. Study area

### 2.1 Geological setting

The Alboran domain is structurally complex and its geodynamics are still debated (Duggen et al. 2008). Extension and subsidence occurred during the Early to Middle Miocene (Comas et al., 1999; Faccenna et al., 2004; Do Couto et al., 2016). As a result of the extension in the area, the Alboran Sea oceanic crust has been thinned, with a minimum thickness of 13 km in some parts (López Casado et al., 2001). The Alboran Sea is the westernmost basin of the

Mediterranean Sea, and is closely connected to the Atlantic Ocean by the Strait of Gibraltar. The Alboran Sea is approximately 400 km long, with a width of 200 km, an average depth of 1300 m and a maximum depth of 1800 m (Olivet et al., 1973; Comas et al., 1999). The Alboran Sea's metamorphic basement is intruded by a number of volcanic plateaus and seamounts formed through the extensional processes that took place between 17 and 8 million years ago (Comas et al., 1999; Duggen et al., 2008). One of these shallow volcanic plateaus, the Banc des

Provençaux (ca. 200 m depth), extends in a series of 3 ridges colonized by CWCs, named "Brittlestar ridges" (BRI, BRII, BRIII) (Comas et al., 2009). They are part of the larger East Melilla Coral Province nestled at depths of between 250 and 450 m. The ridges are 3 to 20 km in length, whilst the mounds vary in height from 50 to 150 m



(Hebbeln et al., 2019). These mounds have mostly dead corals with scarce living corals at their summits and erosional moats at their base, supporting the presence of dynamic currents that influenced the area (Hebbeln et al., 110    2019) (Fig.1).

## 2.2 Oceanography

Low salinity (ca. 36.5 psu), low density Atlantic Water enters the Mediterranean through the Strait of Gibraltar. This inflowing water mass mixes with Mediterranean water while crossing the Strait of Gibraltar to form the Modified Atlantic Water (MAW), the dominant surface water mass in the Alboran Sea (La Violette, 1983; Millot, 2009). In 115    addition, evaporation also exceeds river runoff and precipitation; hence MAW becomes saltier and denser journeying east and finally sinks in the Levantine, Aegean, Adriatic and Liguro-Provençal sub-basins (Millot et al., 2006). Intermediate waters consist of the highly saline (ca. 38.5 psu) and warm (ca. 13.5 °C) Levantine Intermediate Water (LIW) that forms in the Levantine basin and flows from East to West, entering the western Mediterranean through the Straits of Sicily to finally exit through the Strait of Gibraltar (Millot, 2013). Levantine Intermediate Water 120    contributes to ca. 70 % of the total outflow of Mediterranean Outflow Water (MOW; Millot, 2013). The LIW flows between 200 and 600 m water depth, whilst the core of the LIW is situated at approximately 400 m depth (Millot, 2009).

It is important to note that the LIW receives contributions from other intermediate water masses before it enters the 125    western Mediterranean and hence has different characteristics to the LIW in the eastern Mediterranean (Millot, 2013). Moreover, intermediate waters appear to differ between the North and South Alboran Sea (Fig. 2). The LIW flows essentially along the Spanish margin, whilst Shelf Water (ShW), i.e. a mixture of MAW and Western Mediterranean Deep Water (WMDW), dominates intermediate depths along the Moroccan margin (Ercilla et al., 2016). Brittlestar Ridge I lies in the depth range of the ShW (Fig. 2). The deepest water mass, flowing under LIW 130    and ShW, is WMDW, which forms in the Gulf of Lions and flows westward to finally exit through the Strait of Gibraltar and contribute to MOW (Millot et al., 2006). In the Alboran Sea, WMDW circulates principally along the Moroccan margin (Hernandez-Molina et al., 2002; Ercilla et al., 2016).

The surface MAW extends down to approximately 200 m depth (Katz, 1972) and enters the Northeast Alboran Sea 135    as a jet (1.6 Sv; 1 Sv = $10^6$.m$^3$.s$^{-1}$; Lanoix, 1974). This jet triggers the formation of the quasi-permanent anti-cyclonic Western Alboran Gyre that contributes to mixing between surface MAW and underlying LIW (Heburn and La Violette, 1990; Lafuente et al., 1998). When the waters of the Western Alboran Gyre reach the African coast, they separate into two branches: one flows back westward along the coast towards the Strait of Gibraltar while the other flows towards the eastern part of the basin to form the Eastern Alboran Gyre (La Violette, 1983; Viúdez and Tintoré, 140    1995). This second non-permanent gyre also contributes to the mixing process between surface and intermediate water masses. The Banc des Provençaux and Brittlestar Ridge I are situated in the path of the eastward circulating branch/Eastern Alboran Gyre (Lanoix, 1974; Viúdez and Tintoré, 1995; Fig. 1). The mixing between surface and intermediate water masses occurs down to ca. 300 m water depth (Heburn and La Violette, 1990). The Strait of





Gibraltar is a shallow (ca. 300 m depth) and narrow (ca. 20 km wide) crossing point for entering lower salinity

MAW and exiting higher salinity MOW (Heburn and La Violette, 1990; Millot, 2009). Thus, the Strait of Gibraltar plays a key role in controlling water mass exchanges between the semi-enclosed Mediterranean Sea and the Atlantic Ocean. The importance of the water exchange varies between glacial and interglacial periods as a function of sea level change. Moreover, the narrow width and depth of the Strait of Gibraltar, together with the geometry of the Alboran basin and the Coriolis force, affects the formation, mean position and shape of the Alboran gyres (Heburn

and La Violette, 1990). Thus, this will in turn affect mixing between surface and intermediate water masses in the Alboran Sea.

### 3. Material and methods

#### 3.1 Sample collection

This study is based on the multiproxy analysis of gravity core MD13-3462G (35°26.531′N, 2°31.073′W; 327 m

depth; 926 cm long) recovered during the EUROFLEETS cruise MD194 Gateway 'The Mediterranean-Atlantic Gateway Code: The Late Pleistocene Carbonate Mound Record' on board of the R/V *Marion-Dufresne II* (Van Rooij et al., 2013). Cores were split frozen and sedimentary facies descriptions were made at the University of Fribourg prior to sampling. These descriptions include the detailed investigation of texture, grain-size and colour of the matrix sediment, together with the identification and assessment of the preservation state of major macrofaunal components

(Fig. 3). All data was plotted using the ggplot2 package for R (Wickham, 2016; R Core Team, 2018).

#### 3.2 Macrofaunal quantification

X-ray Computed Tomography (CT) imaging was carried out on whole-round sections using a Siemens *Somatom Definition AS64* at the Institute of Forensic Sciences at the University of Bern (Switzerland). Core sections were scanned using an X-ray source operating at 120 kV. The images were reconstructed with a slice thickness of 0.6 mm

taking into account an increment of 0.3 mm. The pixel resolution of the slices is 0.3 mm. The *Avizo 9.4* software was used to visualize, segment and quantify the volumes of the main macrofaunal components (coral, bryozoan and bivalve/brachiopod fragments). Prior to segmentation, images were filtered to remove noise in the matrix, using a non-local means filter. Brachiopods and bivalves were segmented manually. Corals, matrix, pores and bryozoans were segmented through the combination of dual thresholding and watershed segmentation. Labelled fragments

smaller than 5 voxels were filtered prior to quantification. The material statistics module was used to quantify the volume % of faunal fragments per slice and the same volume of interest was selected for each core section.

#### 3.3 Geochemical logging

Geochemical logging was performed using the *Itrax* high-resolution X-ray fluorescence (XRF) core scanner on split cores at the Institute of Geological Sciences, University of Bern (Switzerland). Measurements were taken at 5 mm

intervals using an integration time of 20 s at 30 kV and 45 mA. To counter potentially biased measurements linked to





the uneven surface of CWC cores, such as the direct measurement of air or of CWC skeletons, a post treatment of the dataset was carried out. X-ray fluorescence values with Argon counts higher than 6000, representing the measurement of air and thereby more porous/cracked media not representative for changes in sediment composition, were removed from the final dataset. In this study, we use the $Log_{10}$ normalized ratios (Gregory et al., 2019) Ti/Al and Si/Al as proxies for aeolian input, whilst the $Log_{10}$ Zr/Al and Rb/Al are used as proxies for fluvial input. Indeed, titanium enrichment is considered a typical indicator of increased Saharan dust influence (Frigola et al., 2008; Itambi et al., 2009; Rodrigo-Gámiz et al., 2011), as aeolian deposits tend to concentrate heavy minerals that are rich in elements such as titanium or zirconium (Balsam et al., 1995; Itambi et al., 2009, Rodrigo-Gamíz et al., 2011). Silicates make up an important part of Saharan material, whilst they are rare in Alboran sediments (Caquineau et al., 2002; Masqué et al., 2003). Thus, in the same way as for titanium, enrichment in silica can be used as a proxy for increased aeolian input originating from the Sahara (Rodrigo-Gámiz et al., 2011; Feenstra, 2020). Since rubidium is common in aluminosilicate minerals contained in fluvial material, the Rb/Al ratio is used as an indicator of terrestrial run-off in the western Mediterranean (Calvert and Pedersen, 2007; Martinez-Ruiz et al., 2015; Feenstra, 2020). Though zirconium is generally considered as a proxy for aeolian input for the same reasons as Titanium (Rodrigo-Gámiz et al., 2011), it has been shown that sediments originating from major Moroccan rivers are considerably enriched in zirconium (Stanley et al., 1975). We hence use the Zr/Al and Rb/Al ratios as regional proxies for fluvial input.

### 3.4 Grain-size analysis and organic geochemistry

Grain-size of the siliciclastic fraction was analysed using the *Malvern Mastersizer 3000* at the Department of Geology, Ghent University (Belgium). The core was sampled with a small spoon ($1 \text{ cm}^3$) every 5 cm. Large clasts (>1 cm), such as coral or bryozoan fragments, were removed prior to analysis. Samples were placed in 35 % $H_2O_2$ to remove organic matter and boiled until the reaction ended. Following this first step, samples were boiled in 10 % HCl for 2 minutes to dissolve $CaCO_3$. Prior to measurement, samples were placed in 2 % sodium polymetaphosphate and boiled to assure complete disaggregation. Any remaining particle larger than 2 mm was sieved out before measurement. Eighty seven size classes were measured (from 0.01 to 2000 μm). Each sample was measured three times and results were then averaged. Mean grain-size of the siliciclastic fraction $\overline{GS}$ (Folk and Ward, 1957) was calculated on the entire dataset with the *Rysgran* package for R (Gilbert et al., 2015; R Core Team, 2018). The sortable silt mean size $\overline{SS}$, as defined by McCave et al. (1995; i.e. the mean of the 10-63 μm grain size range), was also calculated following the same procedure. Furthermore, following McCave and Hall (2006), the percentage of sortable silt (SS%) in the total <63 μm fraction was calculated. This percentage, together with the sortable silt mean size, was used as a proxy for bottom current velocity (McCave and Hall, 2006; Toucanne et al., 2012). It has to be mentioned that the use of $\overline{SS}$ as a proxy for bottom current velocity on cores recovered from CWC mounds may be biased (e.g. Eisele et al., 2011). Indeed, the baffling effect of coral framework can locally reduce bottom current velocity and favour the deposition of fine sediments (Huvenne et al., 2009; Titschack et al., 2009; Fentimen et al., 2020), thus leading to an underestimation of $\overline{SS}$ during periods with high CWC content. Because of this, only relative increases in $\overline{SS}$ are considered in combination with results obtained from other proxies.



Total Organic Carbon (TOC, weight%) and Mineral Carbon (MinC, weight%) contents were determined on matrix sediments every 10 cm using the Rock-Eval6 technique at the laboratory of Sediment Geochemistry at the University

of Lausanne (Fantasia et al., 2019). Following Jiang et al. (2017), the percentage of carbonates was calculated as $CaCO_3(\%) = 7.976 \times MinC$. The RockEval6 technique produces an Oxygen and Hydrogen index, respectively corresponding to the quantity of $CO_2$ relative to TOC and the quantity of pyrolyzable organic compounds relative to TOC (Fantasia et al., 2019). These two indices give an indication about the origin of the organic matter present in the samples (Van Krevelen, 1993).

**3.5 Microfaunal and macrofaunal investigations**

The core was sampled (sliced) every 10 cm for micropalaeontological analysis. Samples were weighed dry, washed through a 63 μm mesh sieve and dried at 30 °C. Each fraction was then dry sieved through a series of 63, 125 and 2000 μm mesh sieves and weighed. A target number of 300 benthic foraminifera were identified from the fraction larger than 125 μm for each sample. If the residue contained more than 300 specimens, it was split using a dry

microsplitter. Relative abundances (percentages) of benthic species were calculated from the total benthic foraminiferal assemblage. The benthic foraminiferal density was calculated by dividing the total number of foraminifera of a given sample by the sample fraction's weight. The diversity Shannon index (H') was computed using the PRIMER6 software (Clarke and Gorley, 2006).

Samples prepared for micropaleontological analysis were further used to identify bryozoan species/genera at the Department of Biological, Geological and Environmental Sciences, University of Catania (Italy) on the 125 μm to 2 mm and >2 mm sized fractions. Key intervals with high bryozoan content, previously identified by CT imagery, were selected. Dominant scleractinian corals and main brachiopod and bivalve species were identified at the lowest taxonomic level possible on the 2 mm sized fraction at the Department of Geosciences, University of Fribourg

(Switzerland).

**3.6 Oxygen and Carbon stable isotope analysis**

Stable oxygen and carbon isotope compositions were measured on 5 to 12 specimens of the planktonic foraminifera *Globigerina bulloides* and the benthic foraminifera *Cibicides lobatulus* from the size fraction 212-250 μm in order to prevent any ontogenic effect on the measurements (Schiebel and Hemleben, 2017). The specimens were first cleaned

three times with distilled water in an ultrasonic bath for 2 seconds. The measurements were then made using a *Thermo Fisher Scientific GasBench II* connected to a *Thermo Finnigan Delta Plus XL* isotope ratio mass spectrometer at the Stable Isotope Laboratory of the University of Lausanne (Switzerland) according to the method adapted from Spötl and Vennemann (2003). Results are reported in the conventional δ-values in permil (‰) relative to the Vienna Pee Dee Belemnite (VPDB) standard. Analytical standard deviations (1σ) average 0.04 ‰ for $\delta^{13}C$ and

0.06 ‰ for $\delta^{18}O$ values based on 8 replicate analyses of standards in each sequence of 40 samples.



### 3.7 Radiometric dating

Radiocarbon dating was performed on benthic foraminifera from 3 samples from the upper first meter of core MD13-3462G at the Laboratory of Ion Beam Physics, ETH Zürich, Switzerland (Table 1). The epibenthic foraminifera species *Discanomalina coronata*, *Cibicides lobatulus* and *Cibicides refulgens* were picked in order to obtain between

4 and 10 mg of pure carbonate. The samples were first dissolved in phosphoric acid. The resulting extracted $CO_2$ was then converted to graphite and measured by Accelerator Mass Spectrometry (AMS) technique using the *MICADAS* dedicated instrument (Synal et al., 2007). Results were corrected for [13]C and calibrated using the Marine13 calibration curve (Reimer et al., 2013) and the software OxCal v4.2.4 (Ramsey, 2017). A reservoir age of $390 \pm 80$ years was applied to all ages (Siani et al., 2000).


Uranium-series dating was carried out on 10 CWC fragments (*D. pertusum* and *M. oculata*) using a multicollector inductively coupled plasma source mass spectrometer MC-ICPMS (*Thermo Fisher Scientific Neptune^plus*) coupled with a dissolver (*Aridus I*) at the Institute of Environmental Physics, Heidelberg University (Table 2). In order to constrain the chronostratigraphy of the core, well-preserved coral fragments were selected at the upper and lower

boundaries of coral-rich units. These were identified based on visual core descriptions and CT-analysis (macrofaunal quantification; Fig. 3). Coral fragments were physically cleaned with a *Dremel*® drill tool and by sand blasting, and further chemically cleaned using a weak acid leaching prior to measurements. The detailed sample protocol is described by Frank et al. (2004), while spectrometry and chemical U and Th extraction and purification followed Wefing et al. (2017). Uranium-series coral ages were used to calculate mound aggradation rates.

## 4. Results

### 4.1 Chronostratigraphy

The chronostratigraphy of core MD13-3462G is based on the combination of the coral ages (U-series dating), the planktonic and benthic stable oxygen isotope records, and the foraminiferal radiocarbon ages for the top first meter of the core (Fig. 3). Coral ages have been widely used to define the chronology of cores recovered from coral

mounds. This approach provides satisfying results although age reversals down core have to be taken into account (e.g. Rüggeberg et al., 2007; Frank et al., 2009; Matos et al., 2017). Indeed, reefs are fragile structures and can collapse, topple and fragment through the action of bioerosion, strong bottom currents, and gravity-driven processes, resulting in transport and redeposition of coral fragments (Beuck et al., 2005; Dorschel et al., 2007; White et al., 2007). In contrast, constructing a continuous age model based on stable isotope records is generally considered

untrustworthy for cores collected from coral mounds since sedimentation is intermittent (Dorschel et al., 2005). However, coral ages at the upper and lower boundaries of coral build-up phases in core MD13-3462G (e.g. at 390 and 507 cm depth) correspond to changes in the stable oxygen isotope records (Fig. 3), which in turn match the changes between Marine Isotope Stages (MIS; Lisiecki and Raymo, 2005). As such, the stable oxygen isotope records can, in the case of core MD13-3462G and in conjunction with coral ages, indicate important stratigraphic





boundaries (Fig. 3). This is particularly relevant during times when CWCs did not grow and hence cannot serve to construct a timeframe.

The coral ages indicate that core MD13-3462G extends approximately from 300 ka BP (Marine Isotope Stage 9) to the Holocene (Fig. 3, Table 2). The stratigraphic boundaries from the base of the core to ca. 600 cm depth were defined based on the coral ages as planktonic stable oxygen isotope compositions show little variation. The boundaries of MIS 8 are the most poorly defined (Fig. 3). Due to difficulties to define precisely the stratigraphy of this section of the core, it will not be considered in detail during this study. In contrast, the planktonic and benthic $\delta^{18}$O values and the coral ages do constrain the stratigraphic boundaries from MIS 6 to MIS 1 (Fig. 3). Low planktonic and benthic $\delta^{18}$O values correspond to interglacial periods, whilst high planktonic and benthic $\delta^{18}$O values correspond to the last glacial periods of MIS 6, 2 and 2 (Fig. 3). These boundaries are confirmed by the coral and foraminiferal ages (Fig. 3; Tables 1 and 2). Highest planktonic and benthic $\delta^{18}$O values (3.5 and 4 ‰) correspond to MIS 4 whilst average planktonic and benthic $\delta^{18}$O values between 2 and 3 ‰ correspond to MIS 3.

### 4.2 Sediment characterization

The sediment in core MD13-3462G consists mostly of macrofaunal remains (essentially corals and bryozoans) surrounded by a clay- to silt-sized carbonate/siliciclastic matrix. No important variation in the matrix sediment is observed throughout the core. Carbonate content varies from ca. 10 to 86 %, whilst average values generally range between 40 and 60 % (Fig. 4). Total organic carbon content in the sediment varies between 0.16 and 1.13 wt% (Fig. 4). The highest TOC value is measured during late MIS 3 (1.13 wt%), whilst the lowest is recorded during MIS 8 (0.16 wt%; Fig. 4). The most important shifts to higher TOC values are observed during MIS 5, MIS 3 and at the transition between MIS 2 and MIS 1 (Fig. 4). High TOC values correspond to interglacials, whilst low values correspond to glacials (Fig. 4). The sediment samples are further characterized by high Oxygen index values (> 200 mg C0$_2$/g TOC; Supplementary data), indicating that the organic matter is oxidized and of essentially terrestrial origin (Espitalié et al., 1985).

The mean grain size of the siliciclastic fraction ($\overline{GS}$) varies between ca. 6 and 14 µm (Fig. 4), whilst $\overline{SS}$ varies between ca. 19 and ca. 26 µm (Fig. 4). Trends in $\overline{SS}$ follow those of $\overline{GS}$ (Fig. 4). Overall, a decrease in $\overline{SS}$ and $\overline{GS}$ is associated to intervals marking the transitions from interglacial to glacial periods (Fig. 4). Conversely, an increasing trend is observed from ca. 550 to ca. 375 cm depth, corresponding to the passage from the later phases of MIS 6 to the end of MIS 5 (Fig. 4). This trend is mirrored in $\overline{GS}$ (Fig. 4). A sharp decrease in $\overline{SS}$ and $\overline{GS}$ marks the passage from MIS 3 to MIS 2 and the later phase of MIS 2 (Fig. 4). The percentage of sortable silt (SS%) increases with $\overline{SS}$ (Fig. 5). As discussed by McCave and Hall (2006) and McCave et al. (2017), the straight-line relationship (slope of ca. 0.125 µm/% and an intercept at 0% of ca. 17.5 µm) between $\overline{SS}$ and SS% is indicative of a sorting process induced by bottom currents (Fig. 5).





### 4.3 Macrofauna

The major macrofaunal fragments present in the core are scleractinian corals, bryozoans, brachiopods and bivalves (Fig. 3; Fig. 6). Sea urchins, gastropods, serpulids and gorgonian fragments are more sporadically distributed. The dominant coral species in the core is the scleractinian *D. pertusum*. In the upper 20 cm, *D. pertusum* is replaced by *M. oculata* (Fig. 3; Fig. 6). A third and solitary species, *Desmophyllum dianthus*, is scarcely distributed (Fig. 3). High CWC content is observed during interglacial periods, whilst low content characterizes glacial periods (Fig. 3).

During MIS 3 coral content shows a more staggered distribution, with a range of values from less than 10 vol% to ca. 27 vol% (Fig. 3).

In total 23 genera of bryozoans were identified. *Buskea dichotoma* is by far the dominant bryozoan species (Fig. 6). Accessory species/genera are mainly represented by *Reteporella sparteli*, *Tubuliporina* sp. and *Palmiskenea* sp.

Bryozoan content varies in general between 10 and 20 vol% (Fig. 3). Very high content is, however, observed during MIS 2, reaching near to 70 vol%. The fragments, although delicate and fragile, are well preserved, large sized and unworn (Fig. 6). Bryozoans are absent during most of MIS 5. This absence corresponds to the interval when coral content is the most important (Fig. 3). Conversely, the maximum abundance of bryozoans during MIS 2 correlates to a minimum in coral content (Fig. 3).


Brachiopods are mainly represented by the co-occurrence of the species *Gryphus vitreus* and *Terebratulina retusa* (Fig. 6). These two brachiopods are regularly associated to the bivalve *Bathyarca pectunculoides* (Fig. 6). These three invertebrates have been formerly reported from Mediterranean CWC environments. *Gryphus vitreus* and *Terebratulina retusa* are also recorded from Pleistocene CWC deposits from Rhodes, Greece (Bromley, 2005),

whilst *Bathyarca pectunculoides* was found at the Santa Maria di Leuca CWC province (Mastrototaro et al., 2010; Negri and Corselli, 2016). *Gryphus vitreus* was also found associated to "white corals" between 235 and 255 m depth off the coast of the Hyères Islands, France (Emig and Arnaud, 1988). Although being fragile, the shells are well preserved (Fig. 6). The brachiopod/bivalves concentrate as layers; hence they demonstrate a non-continuous distribution (Fig. 3 and 6). They reach their highest abundance during glacial periods, in particular at the end of MIS

3 (30 vol% at 80 cm). Brachiopods and bivalves are completely absent during the last two interglacial periods (Fig. 3).

### 4.4 Benthic foraminiferal assemblages

Shannon diversity ranges between ca. 2.8 at 652 cm and 3.6 at 782 cm (Fig. 7). High Shannon diversity values between 3.4 and 3.6 are recorded during interglacial periods (Fig. 7). The lowest Shannon diversity values (between

2.8 and 3.0) are associated to glacial periods (Fig. 7). A total number of 166 benthic foraminifera species were recognized (Annex 1). The most abundant species are *Bolivina spathulata*, *Bulimina marginata*, *Bulimina striata*, *Cassidulina laevigata*, *Cibicides lobatulus*, *Discanomalina coronata*, *Gavelinopsis praegeri*, *Globocassidulina subglobosa*, *Hyalinea balthica*, *Miliolinella subrotunda*, *Trifarina angulosa* and *Uvigerina mediterranea*.





The three Buliminid species *B. aculeata*, *B. marginata* and *B. striata* demonstrate the same distribution trends and were thus grouped together as *Bulimina* spp. All Miliolids were grouped together for the same reason. The species *M. subrotunda* makes up more than half of the total abundance of the Miliolid group with an average contribution of ca. 53.4 %. The abundances of all important species are given in Figure 7. The opportunistic infaunal *Bulimina* spp. show maximum abundances during interglacial periods (ca. 18 %) and minimum abundances during glacial periods

(ca. 2 %; Fig. 7). *Uvigerina mediterranea* follows a similar distribution to Buliminids, with peak abundances corresponding to interglacial periods (Fig. 7). Relative to *Bulimina* spp., *U. mediterranea*, *G. subglobosa* and *B. spathulata*, the infaunal *T. angulosa* and the epifaunal *D. coronata* are the least abundant during the last two interglacials (between ca. 1 and 5 %), whilst they are the most abundant during glacial periods, with peak abundances reached during MIS 4 for *D. coronata* (ca. 30 %; Fig. 7). Abundances of Miliolids (5-22 %), *C.*

*lobatulus* (3-17 %) and *C. laevigata* (3-17 %) are relatively high throughout the entire core (Fig. 7); although Miliolids show higher abundances during glacials (ca. 20 %). The highest numbers of *C. laevigata* are recorded during glacial periods (ca. 20 %), whilst minimum abundances occur during interglacials (3 % during MIS 5). The epifaunal *G. praegeri* is homogeneously distributed, in contrast to *H. balthica* that first appears in the core at the onset of MIS 5, reaching maximum abundances during MIS 2 (ca. 11 %; Fig. 7).

### 4.5 Stable carbon isotopes


The range of $\delta^{13}$C values of the planktonic *G. bulloides* is between -2.2 ‰ at 12 cm and -0.5 ‰ at 292 cm, whereas that for the benthic *C. lobatulus* is between 0.9 ‰ at 872 cm and 1.8 ‰ at 362 cm (Fig. 4). The planktonic $\delta^{13}$C record has more variability compared to the benthic $\delta^{13}$C record (Fig. 4). During MIS 6, the benthic $\delta^{13}$C is relatively high (ca. 1.5 ‰), whilst the planktonic $\delta^{13}$C record fluctuates between -0.6 ‰ and -1.5 ‰. A decrease in the

planktonic $\delta^{13}$C record (from -0.7 to -1.5 ‰) marks the middle of MIS 5. In contrast, the benthic $\delta^{13}$C remains stable and low (ca. 1.2 ‰) throughout MIS 5 (Fig. 4). The passage from MIS 4 to MIS 3 is characterized by a shift from the low planktonic $\delta^{13}$C recorded during MIS 4 (-1.5 ‰) to higher planktonic $\delta^{13}$C (-0.5 ‰). Conversely, benthic $\delta^{13}$C values shift from high (1.8 ‰) to lower values (1.3 ‰). The passage from MIS 2 to MIS 1 is marked by a sharp decrease in planktonic and benthic $\delta^{13}$C (from -1.2 ‰ to -2.2 ‰ and from 1.8 ‰ to 1.0 ‰ respectively). The last two

glacial intervals, in particular MIS 4, are marked by a stronger difference between benthic and planktonic $\delta^{13}$C values (Fig. 4).

### 4.6 Elemental geochemistry

The Ti/Al and Si/Al ratios follow the same general trend. Variations in Ti/Al and Si/Al ratios are more marked during MIS 7 and MIS 3, in comparison with the more stable values recorded during other periods. Maximum

average Ti/Al and Si/Al values are reached during glacials, whereas interglacials record the lowest values (Fig. 8). The Zr/Al and Rb/Al ratios follow the same trend, whilst differing strongly from the Ti/Al and Si/Al records. The Zr/Al and Rb/Al ratios demonstrate overall low values throughout the core. However, higher Zr/Al and Rb/Al ratios are reached at the end of MIS 6, and during MIS 5 (ca. 400 cm) and MIS 3 (ca. 100 cm). In the same way as for



Ti/Al and Si/Al records, Zr/Al and Rb/Al ratios demonstrate an important variability during MIS 3, in comparison to
other periods where the records are comparatively stable (Fig. 8).

## 5. Discussion

### 5.1 Environmental controls on coral proliferation during interglacial periods

#### 5.1.1 Humid continental conditions, fluvial discharge and increased food availability

During interglacial periods, benthic foraminiferal assemblages are marked by high abundances of the infaunal
*Bulimina* spp., *U. mediterranea* and *B. spathulata*. Several authors describe *Bulimina* spp. as characteristic for
eutrophic and dysoxic environments (Phleger and Soutar, 1973; Lutze and Coulbourn, 1984; Jorissen, 1987;
Schmiedl et al., 2000). In the Mediterranean Sea, they are dominant in the vicinity of the Po river delta in the North
Adriatic Sea and close to the Rhône River delta (Jorissen, 1987; Mojtahid et al., 2009). The shallow infaunal *U.
mediterranea* and the opportunistic *B. spathulata* are known to demonstrate a positive correlation with organic
matter flux (De Rijk et al., 2000; Schmiedl et al., 2000; Fontanier et al., 2002; 2003; Drinia and Dermitzakis, 2010).
Moreover, *Bulimina* spp. and *U. mediterranea* are reported to be able to feed on fresh but also more refractory
organic matter (De Rijk et al., 2000; Koho et al., 2008; Dessandier et al., 2016). Based on these observations, the
benthic foraminiferal assemblage during interglacials would support a high organic matter export to the seafloor. The
overall higher TOC levels during interglacials confirm that the sediment during these periods was relatively enriched
in organic matter in comparison to glacial periods (Fig. 4). High abundance of the shallow infaunal *G. subglobosa*
has been linked to the deposition of fresh phytodetritus on the seafloor after bloom events (Gooday, 1993;
Fariduddin and Loubere, 1997; Suhr et al., 2003; Sun et al., 2006). It is typically found in high energy (e.g. steep
flanks, ridges) and well-oxygenated environments (Mackensen et al., 1995; Milker et al., 2009), and is a common
taxon of the Alboran Platform and of CWC environments (Margreth et al., 2009; Milker et al., 2009; Spezzaferri et
al., 2014). Mackensen et al. (1995) noted that *G. subglobosa* dominated in areas of the South Atlantic Ocean where
the organic carbon flux did not exceed 1 g.cm$^{-2}$yr$^{-1}$. In contrast, in the Mediterranean Sea, *B. marginata* is restricted
to sites with an organic carbon flux >2.5 g.cm$^{-2}$yr$^{-1}$, whilst *B. aculeata* is associated to a flux of 3 g.cm$^{-2}$yr$^{-1}$ (De Rijk
et al., 2000). The last two interglacials (MIS 7 and MIS 5) are marked by an increased abundance of *G. subglobosa*
at early stages followed by a general decline. Buliminids follow a converse trend, particularly during MIS 5, with
lower abundances at early stages (Fig. 7). This suggests that conditions during the later stages of interglacials became
increasingly eutrophic and in turn less oxygenated at the sediment/water interface, as the consumption of organic
matter led to oxygen depletion. These more environmentally stressful conditions resulted in decreased foraminiferal
diversity and a proliferation of opportunistic taxa (Fig. 7). Overall lower abundances of Miliolids, which are typically
found in well-oxygenated environments (Murray, 2006), further confirm eutrophication coupled to lower
oxygenation at the seafloor during interglacials, specifically towards the end of interglacials (Fig. 7).





Schmiedl et al. (2010) link the high abundance of *U. mediterranea* in the Aegean Sea to humid climatic conditions and increased river runoff. This observation is in agreement with overall increased fluvial and reduced aeolian input during interglacial periods at BRI, as evidenced by the Al-normalized elemental ratios (Fig. 8). Increased fluvial

input has been widely linked in the eastern Mediterranean to more humid continental conditions during interglacial times in response to a northern shift of the African monsoon (e.g. Gasse, 2000; Gasse and Roberts, 2005; Osborne et al., 2008; Coulthard et al., 2013). In contrast, the Alboran Sea lies below the maximum Inter-Tropical Convergence Zone northward position and is sheltered by the Atlas Mountain chain (Rohling et al., 2002; Tuenter et al., 2003; Lavaysse et al., 2009). Modern-day observations show that rainfall over the northwest Atlas Mountains is generally

associated to baroclinic activity over the North Atlantic (Knippertz et al., 2003; Braun et al., 2019). The south of the Atlas Mountains has one of the highest cyclonic activities in the Mediterranean borderlands, whilst the largest fraction of cyclones entering the Mediterranean Sea arrives from the Atlantic (Lionello et al., 2016). Pasquier et al. (2018) noticed that periods of increased input of organic matter from sediment-laden rivers occur during warm substages of the last 200 ky. These authors relate these pluvial events to negative North Atlantic Oscillation-like

conditions (Pasquier et al., 2018). The East Melilla Coral Province is located 50 km away from the mouth of the Moulouya River which takes its source in the High Atlas Mountains (Snousi, 2004; Emelyanov and Shimkus, 2012; Tekken and Kropp, 2012). The basin of the Moulouya River covers approximately 54,000 km², hence representing the largest river basin in Northwest Africa (Emelyanov and Shimkus, 2012; Tekken and Kropp, 2012). We propose that the influence of warm and moist Atlantic air masses during interglacial periods led to warmer and more humid

conditions over Northwest Africa and torrential rain fall. This would have led to a strengthening of the Moulouya River's flow rate, hence triggering episodes of important terrestrial organic matter input at BRI. These events may have in turn caused eutrophication and oxygen depletion at the seafloor, compatible with the benthic foraminiferal assemblages. Dysoxic conditions during interglacial periods would have hampered coral proliferation, as suggested by the low mound aggradation rates (Fig. 9). However, dysoxic conditions may have been limited to the sediment,

thus only affecting foraminiferal communities and not fully preventing colonial corals living above the sediment surface to develop. Such vertical decoupling between sediment and pelagic ecosystems has previously been observed in modern Norwegian CWC reefs (Wehrmann et al., 2009). Overall, high food availability triggered by increased fluvial discharge appears to be a decisive parameter governing coral proliferation at BRI.

**5.1.2 Enhanced surface and intermediate water mass mixing**

During interglacial periods, the high sea level and the increased evaporation in the Mediterranean leads to a more important inflow of low salinity MAW through the Strait of Gibraltar (Sierro et al., 2005). Thus, surface waters in the Alboran Sea are, in comparison to glacial periods, warmer and less dense. This is also noticed in the planktonic $\delta^{18}O$ record (Fig. 3). The enhanced MAW flow during interglacials triggers stronger Western and Eastern Alboran

Gyres, resulting in better mixing and downwelling. Knowing that the Banc des Provençaux and BRI are situated at relatively shallow water depths and in the path of the eastward circulating branch/Eastern Alboran Gyre (Lanoix, 1974; Viúdez and Tintoré, 1995; Fig. 10), and that mixing between surface and intermediate water masses is



documented to occur down to ca. 300 m water depth (Heburn and La Violette, 1990), it is conceivable that the corals living currently at 327 m depth were bathed by, or situated at the limit of mixing between surface and intermediate

water masses during interglacial periods. Higher input of MAW into the Alboran Sea would lead to an increased contribution of surface waters to intermediate water masses (ShW) and a deepening of the pycnocline. This would promote the formation of internal waves and increase turbulence at the seafloor of BRI, as suggested by the slightly higher $\overline{SS}$ values during interglacials (Fig. 4 and 5), and would have favoured coral proliferation by increasing lateral nutrient supply (Fig. 10). The slight offset between planktonic and benthic $\delta^{13}C$ records towards the end of MIS 7

and MIS 5 indicate that water masses were becoming more stratified towards the end of interglacials and that the contribution of MAW to intermediate water masses was hence possibly decreasing. Maximum *Bulimina* spp. abundance, minimum *G. subglobosa* abundance, and decreasing benthic foraminiferal diversity may suggest that reduced mixing, in concomitance with important fluvial discharge (section 6.1.1) led to oxygen depletion at the seafloor at the transition between interglacial and glacial periods. Severe oxygen depletion may explain the decline

of corals at the transition from interglacial to glacial periods.

### 5.1.3 Variability of cold-water coral mound formation between interglacial periods

Highest coral content is reached during MIS 5 and corresponds to a maximum in Buliminid abundance. The Al-normalized elemental ratios suggest that aeolian input during MIS 5 was relatively stable, whilst fluvial input would have increased throughout (Fig. 8). These stable conditions would have favoured a long-lasting coral proliferation

dominated by the scleractinian *D. pertusum* (Fig. 3). Marine Isotope Stage 9 and 7 are also dominated by *D. pertusum.* Although MIS 7 is poorly constrained, Al-normalized elemental ratios would indicate that this time period was more unstable than the previous interglacial period (Fig. 8). The late Holocene is marked by a decrease in coral abundance and a dominance of *M. oculata* over *D. pertusum*. The coral fragment at the top of core MD13-3462G has an age of 6.3 ka. Fink et al. (2013) obtained ages from surface coral fragments at BRI that were generally between

2.7 and 3.1 ka, whilst Stalder et al. (2015) reported an age of 5.4 ka for a surface coral fragment sampled at BRI. Similar ages of between 3.5 and 5.8 ka were obtained on surface coral fragments at the Western Melilla Coral Province (Wang et al. 2019). Dominance of the coral *M. oculata* during the Late Holocene was also observed at BRI by Stalder et al. (2015), whilst Wienberg (2019) reported that *M. oculata* already became the dominant coral species during the mid-Holocene. Previous observations suggest that *M. oculata* is more tolerant to environmental stress than

*D. pertusum* (e.g. Wienberg et al., 2009; Stalder et al., 2015). Thus, the dominance of *M. oculata* at the top of the core would indicate that conditions during the late Holocene were becoming increasingly unsuitable for coral proliferation, particularly for *D. pertusum*. This is consistent with modern-day seafloor observations that report a near-absence of CWCs at BRI (Hebbeln et al., 2019). These combined results point to unfavourable conditions for coral proliferation during the late Holocene, as suggested by Fink et al. (2013), Stalder et al. (2015; 2018) and Wang

et al. (2019). The recent decline of CWCs at the Eastern and Western Melilla Coral Provinces may be linked to the establishment of more arid conditions over North Africa ca. 4 ka ago (Gasse, 2000 and references therein; Shanahan et al., 2015). The fluctuations in coral and bryozoan abundances between the different interglacial periods may be caused by the influence of alternating dry and humid conditions.



### 5.2 Environmental conditions during glacial periods

**5.2.1 Arid continental conditions and reduced bottom currents**

At the exception of MIS 8, for which the boundaries are poorly defined, glacial periods are marked by a change in macrofaunal composition with lower coral and higher bryozoan content in comparison to interglacial periods. Higher bryozoan content during glacials at BRI is in tune with observations made at the Great Australian Bight, where lower temperatures, lower sea level stand, and increased upwelling probably promoted bryozoan proliferation during

glacial periods (James et al., 2000; Holbourn et al., 2002). Conversely, higher temperatures and downwelling during interglacials halted bryozoan extension at the Great Australian Bight (James et al., 2000; Holbourn et al., 2002). Rigid erect branching bryozoans such as *B. dichotoma* are known to be fragile, and hence to prefer low energy environments, being unable to withstand strong bottom currents and turbulence (Scholz and Hillmer, 1995; Bjerager and Surlyk, 2007). Eutrophic environments dominated by infaunal benthic foraminifera (e.g. *Bulimina* spp.) are

unfavourable for erect bryozoans, the high concentration of suspended food particles clogging up their feeding apparatus (Holbourn et al., 2002). Low $\overline{SS}$ values and reduced TOC content in the sediment confirm that glacial periods were marked by weak bottom current velocities and organic matter flux (Fig. 4 and 5). The presence of brachiopod/bivalve layers dominated by the brachiopod *Gryphus vitreus* also characterizes the glacial macrofauna (Fig. 3). This species is found between 160 and 250 m depth along the Mediterranean continental margin and thrives

in areas dominated by moderate bottom currents (Emig and Arnaud, 1988). Thus, this species' co-occurrence with bryozoans confirms that variations in sea level stand, hydrodynamics and trophic conditions govern the change in macrofaunal dominance at BRI. Low organic matter flux during glacial periods has been related to predominantly arid conditions over North Africa, in association with a weak North African monsoon (Gasse, 2000; Sierro et al., 2005). Such arid conditions led to the complete or severe desiccation of major African lakes during the last glacial,

such as Lake Victoria (Talbot and Livingstone, 1989; Johnson, 1996). High Ti/Al and Si/Al elemental ratios would indicate that aeolian input prevailed during glacial periods at BRI (at the exception of MIS 3, section 6.4), hence confirming that continental conditions were arid at these times (Fig. 8 and 9).

The reduced precipitation and retreat of vegetation would have led to a dwindling of fluvial discharge at BRI, as

evidenced by generally low Zr/Al and Rb/Al elemental ratios (Fig. 8). Glacial benthic foraminiferal assemblages are characterized by the dominance of large epibenthic suspension feeding foraminifera, such as *C. lobatulus* and *D. coronata*, together with the infaunal *C. laevigata* (Fig. 7). This follows observations made by Stalder et al. (2018) who noticed increased abundances of *Cibicides* spp., *D. coronata* and *C. laevigata* during glacial periods at BRI. These species share a preference for high quality fresh marine organic matter (De Rijk et al., 2000; Milker et al.,

2009, Stalder et al., 2018). *Cibicides lobatulus* and *D. coronata* prefer oxygen-rich bottom waters (Linke and Lutze; 1993; Margreth et al., 2009), whilst following Milker et al. (2009), high abundances of *C. laevigata* could be related to the presence of fine grained material in the western Mediterranean. Increased abundance of *C. laevigata* matches minimum $\overline{GS}$ values, thus confirming this species' affinity for fine grained glacial material (Fig. 4). In the Arctic basins and Norwegian-Greenland Sea, the dominance of the epibenthic *Cibicides wuellerstorfi* (a relative of *C.*





*lobatulus*) reflects a relative low flux of organic matter (Linke and Lutze; 1993). This species tolerates vertical flux rates <2 g.cm$^{-2}$.yr$^{-1}$ (Altenbach, 1989). The dominance of *C. lobatulus*, *D. coronata*, *C. laevigata* and Miliolids would thus indicate that the seafloor during glacial periods received less but higher quality organic matter and became more oxygenated in response to the stronger influence of intermediate and deep water masses (Fig. 10). These observations suggest that more arid conditions during glacial periods led to a shift from a more fluvial to a

more marine influenced environment (Fig. 10). We propose that weaker but comparatively fresher organic matter input favoured the development of the bryozoan *B. dichotoma*. This assumption is supported by experimental observations demonstrating how erect bryozoans feed essentially on diatoms and that suspension feeding foraminifera use the same food sources (Winston, 1977; 1981; Best and Thorpe, 1994; Goldstein, 1999). Lower nutrient input appears in contrast to have been detrimental for coral proliferation but would not have led to their

complete disappearance (Fig. 3 and 10). It can be hypothesized that there may exist a threshold in the quality and quantity of organic matter determining which of *D. pertusum* or *B. dichotoma* dominates the benthic environment at BRI.

### 5.2.2 Increased stratification and deep water overturning

As highlighted previously, the dominant macrofauna and low $\overline{SS}$ values (Fig. 3, 4 and 5) during glacial intervals at

BRI indicate weaker bottom currents. Wang et al. (2019) relate low off mound $\overline{GS}$ and high benthic foraminiferal $\delta^{13}$C values at BRI during glacials to a dominant influence of MAW coinciding with a low sea level stand. However, whilst the benthic foraminiferal $\delta^{13}$C values from core MD13-3462G are indeed relatively high during glacial periods, the planktonic foraminiferal $\delta^{13}$C values do not follow the same trend (Fig. 4). The decoupling between the planktonic and benthic $\delta^{13}$C records during the two last glacial periods, noticeably during MIS 4, suggests that water

mass stratification was greater than during interglacial periods and that the seafloor was not under the direct influence of surface MAW. During glacial periods, the flow of MAW was reduced due to lower sea level and the reduced evaporation over the Mediterranean (Sierro et al., 2005). This would have reduced the contribution of MAW to ShW and weakened Western and Eastern Alboran Gyres, which would have in turn led to less mixing between surface and intermediate water masses, whilst conversely increasing stratification.


Modern observations show that recently formed dense waters do not necessarily reach the deep western Mediterranean but may, in contrast, be located at intermediate water depths, above 1500 m depth (Sparnocchia et al., 1995; Millot, 1999; Ercilla et al., 2016). Ercilla et al. (2016) further exposed that WMDW can be identified at depths shallower than 500 m depth along the Moroccan margin and that it contributes to the overlying ShW, whilst deep

water overturning and ventilation peaked during MIS 2 (Cacho et al., 2006; Toucanne et al., 2012). Increased oxygenation of the seafloor, as evidenced by the benthic foraminiferal assemblage (Fig. 7), may suggest that the contribution of well-ventilated deep and intermediate water masses at BRI was more important during glacials than during interglacials (Fig. 10). The physical shape of BRI possibly plays a role in the rise of deep waters during glacial periods. In addition, the heavier benthic C-isotope record and the abundance of fresh organic matter feeding

foraminifera (*C. lobatulus* and *D. coronata*) during glacial periods could indicate that these waters were also





nutrient-rich. Although stratification between surface and intermediate water masses was greater during glacials, the stronger flow of well-ventilated WMDW at BRI would explain the higher oxygen availability at the seafloor. Overall during glacial periods, and in particular during the LGM, enhanced contribution of nutrient-rich and well-ventilated WDMW to overlying ShW, coupled to reduced fluvial input and turbulence, would have promoted bryozoan

proliferation (Fig. 10). However, such environmental conditions would be detrimental for coral proliferation (Fig. 10).

### 5.2.3 Fluctuating environmental conditions during the last glacial period

The benthic and planktonic foraminifera $\delta^{18}O$ and $\delta^{13}C$ values indicate that environmental conditions were particularly unstable during the last glacial period, as suggested by previous studies (Cacho et al., 2000; Martrat et

al., 2004; Pérez-Folgado et al., 2004; Cacho et al., 2006; Bout-Roumazeilles et al., 2007). The last glacial shows a strong variability in macrofaunal and benthic foraminiferal assemblages. Maximum coral content is reached during MIS 3 (Fig. 3). This increased coral content is associated to higher numbers of *G. subglobosa* and *C. laevigata*, together with phases of higher Zr/Al and Rb/Al elemental ratios (Fig. 7 and 8). These observations suggest that corals and the benthic foraminiferal community positively responded to short phases of increased surface

productivity related to important continental runoff during MIS 3. This is supported by observations made by Rogerson et al. (2018), who documented more humid conditions during MIS 3 in comparison to the more arid MIS 4 and 2. Humid conditions would hence have promoted coral proliferation through increased fluvial input at BRI, in the same way as during interglacial periods (section 6.1). Nevertheless, the dominance of *G. subglobosa* coupled to the absence of *Bulimina* spp. and *U. mediterranea* suggests that conditions were less eutrophic than during peak

interglacial periods and that the organic matter reaching the seafloor may have been less degraded.

At BRI, high planktonic foraminiferal $\delta^{18}O$ values during the last glacial are associated with increased Ti/Al and Si/Al elemental ratios (Fig. 8). There is evidence that during times of increased aridity, enhanced African winds blew north towards the Alboran Sea (Magri and Parra, 2002; Bout-Roumazeilles et al., 2007). During Heinrich Event 1,

the existence of a steppe/semi-desertic flora around the Alboran borderlands points to cold and dry climatic conditions (Combourieu Nebout et al. 2009). The association of high Ti/Al and Si/Al ratios with high planktonic foraminiferal $\delta^{18}O$ values confirms that increased aridity on land coupled to strong winds were concomitant with lower sea surface temperatures at BRI (Fig. 8). The arid continental conditions during these particularly cold spells would have led to reduced continental runoff. This could in turn explain the overall dwindling of coral communities

during these cold events (Fig. 3). Cacho et al. (1999) and Martrat et al. (2004) showed that sea surface temperature minima matched higher planktonic *G. bulloides* $\delta^{18}O$ values in the Alboran Sea during the last glacial. Moreover, these sea surface temperature minima are concurrent with North Atlantic Heinrich Events, i.e. the deposition of ice-rafted detritus from massive iceberg discharges during some of the colder stadials (Heinrich, 1988; Bond et al., 1992). Ice-rafted detritus layers were observed as far south as the Portuguese margin (Lebreiro et al., 1996; Bard et

al., 2000; Schönfeld and Zahn, 2000), the Gulf of Cádiz (Llave et al., 2006; Toucanne et al., 2007) and the Moroccan margin (Kudras and Thiede, 1970). Rapidly decreasing sea surface temperatures were also associated to North



Atlantic Heinrich events of the Bermuda Rise (Sachs and Lehman, 1999) and in the Alboran Sea (Cacho et al., 1999). Moreover, based on palynological and mineralogical evidence, Bout-Roumazeilles et al. (2007) revealed an intensification of wind, dust erosion and transport toward the Alboran Sea in provenance of western Morocco during

North Atlantic cold events. Based on these observations, we tentatively suggest that the dwindling of coral communities during the last glacial period may also be linked to the inflow of Atlantic glacial freshwater during Atlantic cold events. More precise investigations are however needed to assert this relationship.

**5.3 Interglacial-glacial transition phases**

The western Mediterranean is marked by abrupt interglacial-glacial transitions (Martrat et al. 2004). Benthic

foraminiferal assemblages and $\overline{SS}$ would confirm that the environment at BRI also experienced such abrupt transitions. Indeed, the interglacial-glacial transitions are characterized by increased $\overline{SS}$ values and *T. angulosa* abundances (Fig. 3 and 7). *Trifarina angulosa* is typical for current-swept areas and can withstand permanent winnowing (Mackensen et al., 1995; Schönfeld, 2002; Margreth et al., 2009). These results suggest that transition phases between interglacial and glacial periods were characterized by winnowing at the seafloor. In contrast, benthic

foraminiferal assemblages and $\overline{SS}$ would indicate that transition phases from glacial to interglacial periods were not marked by winnowing or erosional events. These observations differ from the ones drawn from Northeast Atlantic mounds, where winnowing and mass wasting are considered as precursor events for the re-initiation of coral proliferation during interglacials (Dorschel et al., 2005; Rüggeberg et al., 2007). Thus, the environmental mechanisms triggering the reset of coral proliferation at the onset of interglacials at BRI appear to be different from

the Northeast Atlantic. The re-establishment of coral proliferation during the last two interglacials at BRI is concomitant with an increase in Buliminid abundance. This increase in Buliminid abundance is coupled to higher Rb/Al values at the transition between MIS 6 and 5 (Fig. 7, 8). These observations confirm that the recovery of coral proliferation at BRI is tightly linked to an increase in river runoff, which in turn reflects more humid continental conditions. A similar process has been reported from the Viosca Knoll area, where the dispersal of terrestrial organic

matter by the Mississippi River triggers an increase in primary productivity, providing nutrients for coral communities (Mienis et al., 2012). As such, water mass rearrangements appear to be of secondary importance, whilst the rapid increase in fluvial discharge would be the primary factor triggering coral proliferation at BRI.

**5.4 Differences between Southeast Alboran and North Atlantic coral mound formation**

**5.4.1 Coral proliferation and environmental forcing**

In the Northeast and Northwest Atlantic, corals thrive during interglacial periods whilst their proliferation is halted during glacial periods (Dorschel et al., 2005; Rüggeberg et al., 2007; Frank et al., 2009; 2011; Matos et al., 2015; 2017). Coral proliferation at BRI does not follow the same pattern. Indeed corals also develop during interglacial periods, but also to a lesser extent during glacial periods (Fig. 3). Coral proliferation in the Northeast Atlantic is controlled by the northward advance of subtropical waters and of MOW (Henry et al., 2014; Boavida et al., 2019),

whereas corals at BRI are influenced by the interplay between inflowing MAW and outflowing LIW, ShW and



WMDW (Stalder et al., 2015; 2018; Wang et al., 2019; this study). Environmental control on coral development in both regions shares similarities but also shows differences. The positive response of corals to increased bottom current velocity is important in both regions. This follows the general consensus that strong bottom currents are decisive for the development of corals (e.g. White et al., 2005; Mienis et al., 2007; Roberts et al., 2009). The

topography of Brittlestar Ridge I may favour the formation of Taylor columns and the retention of organic matter, such as observed in the Rockall Trough (Northeast Atlantic, White et al., 2007). However, benthic foraminiferal assemblages associated to phases of coral proliferation in the Northeast Atlantic (Rüggeberg et al., 2007) and in the Southeast Alboran Sea (this study) differ. Benthic foraminiferal assemblages associated to phases of sustained coral proliferation at Propeller Mound (Northeast Atlantic) are essentially characterized by large epibenthic foraminifera

(*C. lobatulus*, *Cibicides refulgens*, *D. coronata*, and *Planulina ariminensis*) and the infaunal *Trifarina bradyi* (Rüggeberg et al., 2007). In contrast, at BRI, higher abundances of *C. lobatulus*, *D. coronata* and *T. angulosa* are associated to glacial periods or transition phases between interglacial and glacial periods with low coral abundance, while small infaunal foraminifera dominate phases of coral proliferation (Fig. 7). These contrasting observations suggest differences in food supply and bottom current regimes. Corals in the Northeast Atlantic thrive on fresh

marine-derived organic matter resulting from the North Atlantic blooms which are fuelled by upwelling (Dickinson et al., 1980). In contrast, corals at BRI are likely supplied by plankton blooms triggered by the input of degraded fluvial organic matter during interglacial times, whilst aeolian dust input allows corals to survive during glacial times by triggering local moderate bloom events in the area. In this regard, coral mounds situated in the Southeast Alboran Sea show more similarities to mounds located in the Viosca Knoll area or in the Gulf of Cadiz (Wienberg et al.,

2010; Mienis et al., 2012). The respective shallow location and proximity of BRI to the continent explains the higher influence of continental runoff on coral communities than in the deeper Northeast Atlantic sites. It can hence be expected that corals at BRI show higher sensibility to shifting continental climatic conditions.

### 5.4.2 Long-term coral mound build-up

Average mound aggradation rates for core MD13-3462G are particularly low in comparison to other CWC mound

provinces (Frank et al., 2011; Wienberg and Titschak, 2015). A maximum rate of ca. 9 cm.ky$^{-1}$ is reached during MIS 4, whilst rates do not exceed ca. 4 cm.ky$^{-1}$ during interglacial periods (Fig. 9). In comparison, mound aggradation rates during the Early Holocene reached up to 869 cm.ky$^{-1}$ at Stjernsund Fjord (Norwegian margin) and 290 cm.ky$^{-1}$ in the Porcupine Seabight (López Correa et al., 2012; Frank et al., 2009; Wienberg and Titschak, 2015). Moreover, mound aggradation rates during the Holocene for core MD13-3462G (Fig. 9) are extremely low in

comparison to other reported rates in the area. Indeed, Fink et al. (2013) and Stalder et al. (2015) report aggradation rates between ca. 260 and 290 cm.ky$^{-1}$ for the Early Holocene at the East Melilla Coral Province, whilst Wang et al. (2019) calculated rates between ca. 75 and 107 cm.ky$^{-1}$ for the West Melilla Coral Province. These combined observations suggest that coral mound formation demonstrates strong spatial and temporal variability at the East Melilla Coral Province, and more precisely during the Holocene at BRI.




Long-term coral mound formation at BRI and in the Porcupine Seabight do not show the same temporal distribution. Indeed, mound aggradation in the Porcupine Seabight is restricted to interglacial periods, whilst glacials are marked by winnowing and erosive events (Rüggeberg et al., 2007; Frank et al., 2011). Long-term coral mound formation at BRI took place during interglacial and glacial periods, though at much lower aggradation rates than in the Porcupine

Seabight (Fig. 9; Frank et al., 2011). Mound aggradation rates in core MD13-3462G are comparable to inactive or abandoned reefs in the Porcupine Seabight, i.e. <5 cm.ky$^{-1}$ (Frank et al., 2011), thus suggesting that CWCs did not thrive at the site of core MD13-3462G but rather developed under stressful, possibly dysoxic, environmental conditions. Average long-term mound aggradation rates at BRI show more similarities with mounds situated along the Mauritanian margin that developed during the last glacial (28-45 cm.ky$^{-1}$) but also during the last interglacial

period (16 cm.ky$^{-1}$; Wienberg et al., 2018; Wienberg and Titschak, 2015). In contrast with Atlantic CWC mounds, mounds from the East Melilla Coral Province show a high contribution of the erect cheleistome bryozoan *B. dichotoma*. Based on mound aggradation rates and macrofaunal content, we propose that *B. dichotoma* communities favoured mound formation at BRI, noticeably during glacial periods, by capturing fine-grained sediments in a similar way as CWCs do (Fig. 3 and 9). As such, mounds at BRI stand out and may be considered as mixed *B.*

*dichotoma*/CWC mounds, rather than CWC mounds per se.

**Conclusions**

The multiproxy study of core MD13-3462G provides information on the long-term development of a cold-water coral mound at Brittlestar Ridge I. Three important points can be concluded:

(1) Cold-water corals develop mainly during interglacial periods. Their growth is promoted by the combination of increased fluvial input and enhanced influence of Alboran Gyres. Increased fluvial organic matter inputs are driven by the increased impact of warm and moist Atlantic air masses with intensified Western and Eastern Alboran Gyres that lead to more important turnover between surface and intermediate water masses. This phenomenon is promoted by enhanced Modified Atlantic Water inflow at the Strait of Gibraltar. The seafloor was possibly depleted in oxygen

at the end of interglacial phases. These results demonstrate the paramount importance of enhanced fluvial input as a trigger for cold-water growth in the Southeastern Alboran Sea.

(2) Glacial periods are unfavourable for cold-water corals; in contrast the bryozoan *Buskea dichotoma* is more suited to glacial environmental conditions. The retreat of corals during glacial periods is triggered by arid continental

conditions that lead to reduced fluvial input and nutrient supply. Moreover, reduced inflow of Modified Atlantic Water at the Strait of Gibraltar results in a lower contribution of surface waters to intermediate waters. In contrast, the contribution of Western Mediterranean Deep Water to intermediate water masses increased. Weaker Alboran Gyres and increased contribution of well-ventilated deep waters at intermediate depths resulted in increased stratification. Lower input of organic matter, but less degraded, further characterizes glacial environmental



conditions. Aeolian dust was the main fertilizing influence and may have enabled corals to survive throughout glacial periods.

(3) Average coral mound aggradation rates are particularly low, varying between 1 and 9 cm.ky$^{-1}$. Mound formation takes place during glacial periods as well as during interglacial periods. Low mound aggradation rates during
interglacials and glacials suggest that corals did not thrive but rather developed under stressful environmental conditions at Brittlestar Ridge I. The erect cheleistome bryozoan *Buskea dichotoma* plays an important role in the long-term mound formation at Brittlestar Ridge I, noticeably during glacial periods. Overall, mound development at Brittlestar Ridge I is controlled by alternating aeolian and fluvial inputs, in response to North Atlantic climate dynamics.

From a wider perspective, this study demonstrates how cold-water coral environments can benefit from both fluvial and aeolian terrestrial input, during respectively interglacial and glacial periods. These results underline how cold-water coral systems are capable of withstanding important environmental changes and to survive and adapt to different climatic conditions.

**Data availability**

All the datasets used in this study are available at the open-access repository PANGEA (data awaiting DOI and link).

**Sample availability**

Archive halves of all core sections investigated for this study are available at the Department of Geosciences, University of Fribourg (Switzerland). The sediment residues and the splits of each sample analysed for benthic
foraminiferal assemblages are stored at the Department of Geosciences, University of Fribourg (Switzerland). Bryozoans identified in this study are available at the Palaeontological Museum of the University of Catania (Italy).

**Author contributions**

RF: writing (original draft), visualization, conceptualization, core sampling, investigation (benthic foraminiferal assemblages, main macrofaunal fragments, particle size analysis, stable isotope measurements assisted by TV and
radiocarbon dating assisted by IH). EF: conceptualization, writing (review and editing), XRF investigation (assisted by HV), preparation of samples for Uranium-series dating and RockEval6 pyrolysis. ARu: conceptualization, writing (review and editing), supervision. EH: investigation (CT analysis, macrofaunal quantification). VR: writing (review and editing), visualization. TV: writing (review and editing), investigation (stable isotope measurements), resources. IH: writing (review and editing), investigation (radiocarbon dating), resources. ARo: writing (review and editing),
investigation (bryozoan taxonomy). DVR: writing (review and editing), resources. TA: writing (review and editing),



investigation (RockEval6 pyrolysis), resources. HV: writing (review and editing), investigation (XRF), resources. NF & TK: writing (review and editing), investigation (Uranium-series dating). AF: investigation (core description, CT data analysis, XRF data analysis), conceptualization, writing (review and editing), project administration, funding acquisition, supervision.

**Competing interests**

The authors declare that they have no conflict of interest.

**Acknowledgements**

We thank the SNSF (Swiss National Science Foundation) projects 'Unconventional Carbonate Factories' and '4D-Diagenesis@Mound' (project numbers 200020_153125 and 200021_149247) for funding this research. We also are grateful for the ship time provided by IPEV on the R/V *Marion Dufresne II* within the framework of the EuroFLEETS GATEWAYS project (grant agreement 228344). We further thank Tim Collart for the help he provided with the *Rysgran* package for R and Marc Schori for his help with the *ArcGIS* software. We further acknowledge the help of Rene Eichstädter and Andrea Schröder-Ritzrau regarding Uranium-series dating and quality control. The DFG has provided funding for the Uranium-series dating of corals via the project FR1341/9-1.

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







**Figure 1.** Location of the study area. (A) General map of the Mediterranean Sea and location of the investigated region (B) Bathymetric map of the western Mediterranean Sea based on the GEBCO_2019 gridded bathymetric data. Abbreviations: EMCP: East Melilla Coral Province (red box); WAG: Western Alboran Gyre; EAG: Eastern Alboran Gyre. (C) Bathymetry and location of the Banc des Provençaux and Brittlestar Ridge I (BRI). The white dot indicates the location of the studied core MD13-3462G recovered during cruise "GATEWAY" No. 194 on board the research vessel *Marion Dufresne II* (Van Rooij et al, 2013), during which the multibeam data used was also acquired using the shipboard 12 kHz multibeam echosounder (Van Rooij et al., 2013).

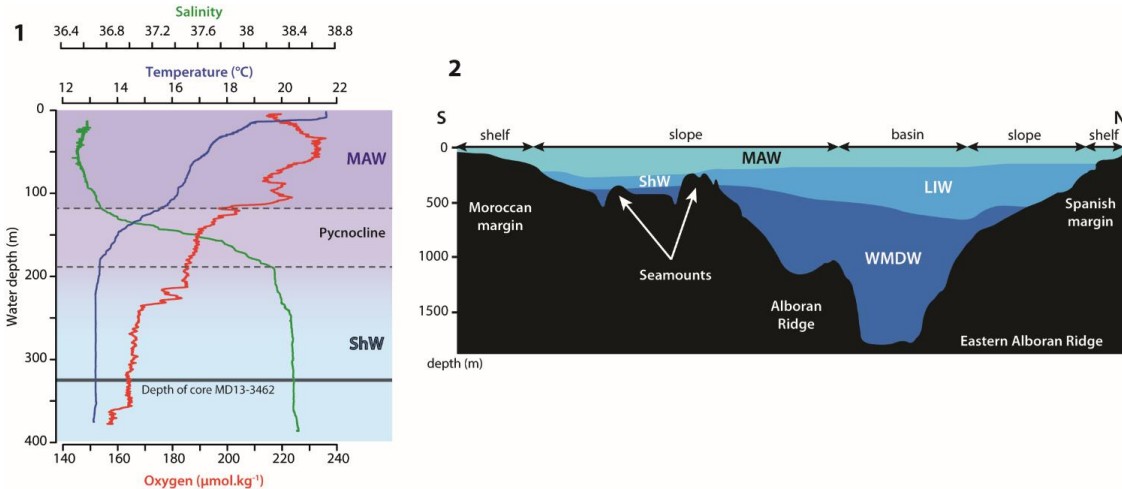

Figure 2. (1) CTD profile taken at the east of Brittlestar Ridge I (35°26,087'N; 2°30,100'W) during cruise "GATEWAY" (No. 194) on board the research vessel *Marion Dufresne II* (Van Rooij et al., 2013). Salinity, temperature (°C) and oxygen content (µmol.kg⁻¹) are indicated. The location of core MD13-3462G in relation to the profile is indicated by the black line. (2) North-South orientated bottom water profile of the East Alboran Sea modified from Ercilla et al. (2016). Abbreviations: MAW: Modified Atlantic Water, ShW: Shelf Water, LIW: Levantine Intermediate Water.



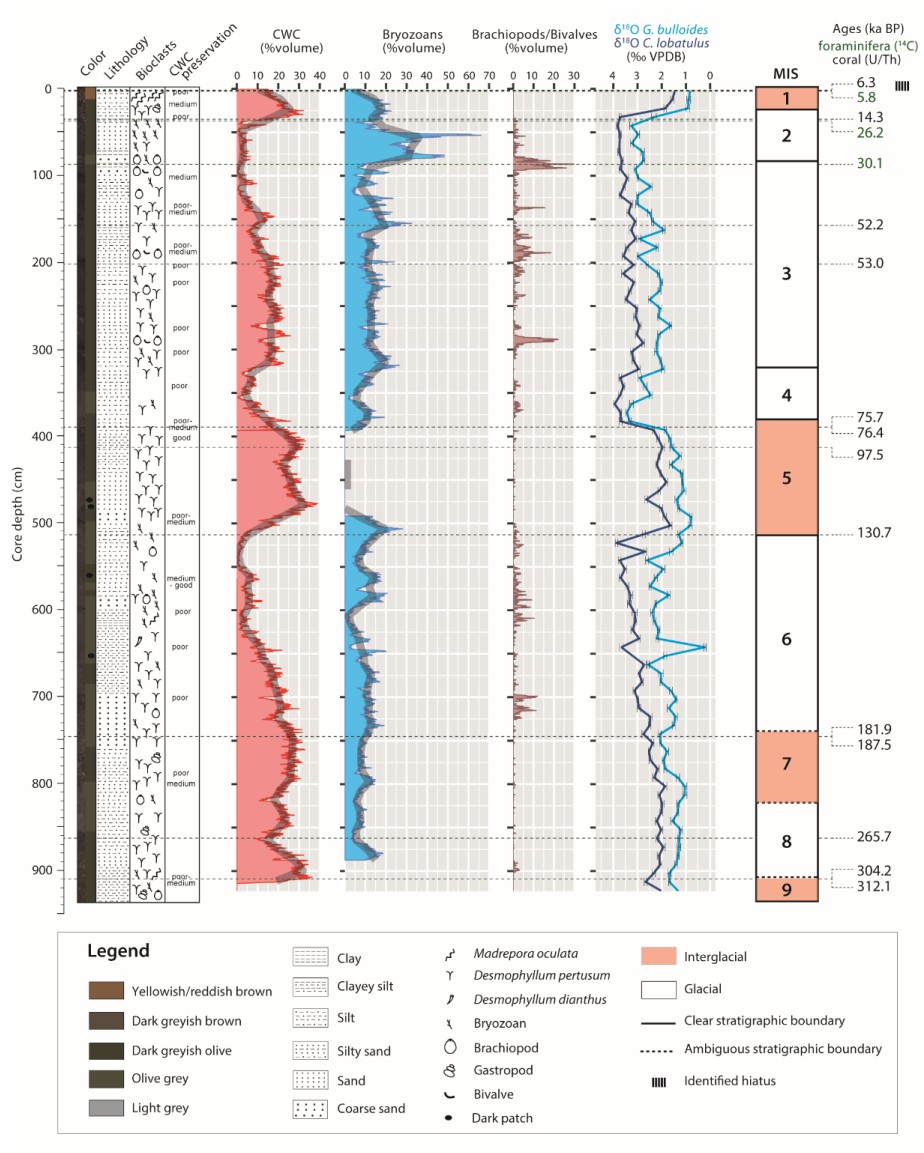





**Figure 3**. Core description, stratigraphy and macrofaunal composition of core MD13-3462G. Stratigraphy is based on the planktonic (*G. bulloides*) and benthic (*C. lobatulus*) δ$^{18}$O records (‰ VPDB), the Uranium-series ages of coral fragments and the epibenthic foraminiferal radiocarbon ages for the first meter of the core (far right). The quantification of the three main macrofaunal components (Cold-water corals: CWC, bryozoans and brachiopods/bivalves) performed through analysis of X-ray Computed Tomography (CT) images is given. Smoothed curves are indicated by the light grey shaded curves.






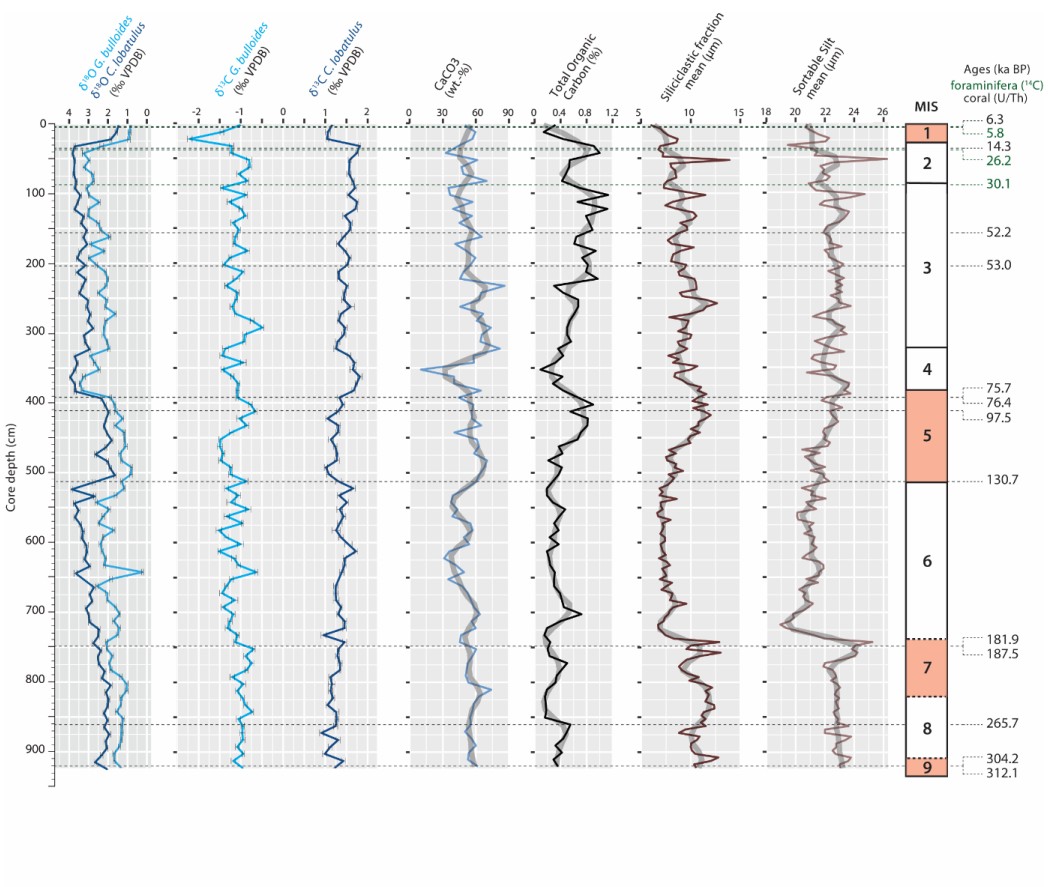


**Figure 4**. Planktonic (*G. bulloides*) and benthic (*C. lobatulus*) δ¹³C records, calcium carbonate (CaCO₃) content (expressed in weight percentage), Total Organic Carbon content (%), mean grain size of the siliciclastic fraction (μm) and mean grain size of the sortable silt fraction (the 10-63 μm grain size range, expressed in μm; McCave et al., 2006). Smoothed curves are indicated by the light grey shaded curves. The stratigraphy defined in Fig. 3 is given to the far right. The planktonic (*G. bulloides*) and benthic

(*C. lobatulus*) δ¹⁸O records (‰ VPDB) are provided as supporting information.



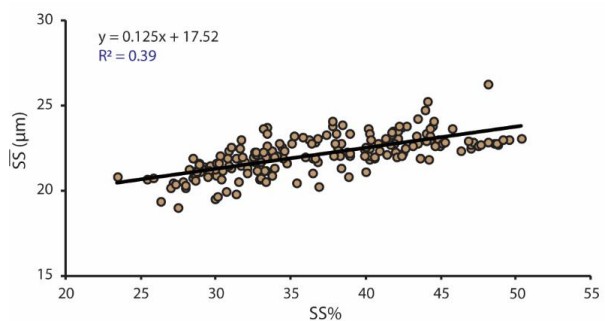

**Figure 5**. Dispersion plot of the sortable silt mean size (the 10-63 μm grain size range, expressed in μm) $\overline{SS}$ vs. the percentage of sortable silt (SS%). The slope of 0.125 μm and intercept at 0 % of 17.52 μm indicates a sorting process induced by bottom currents (McCave et al., 2006).




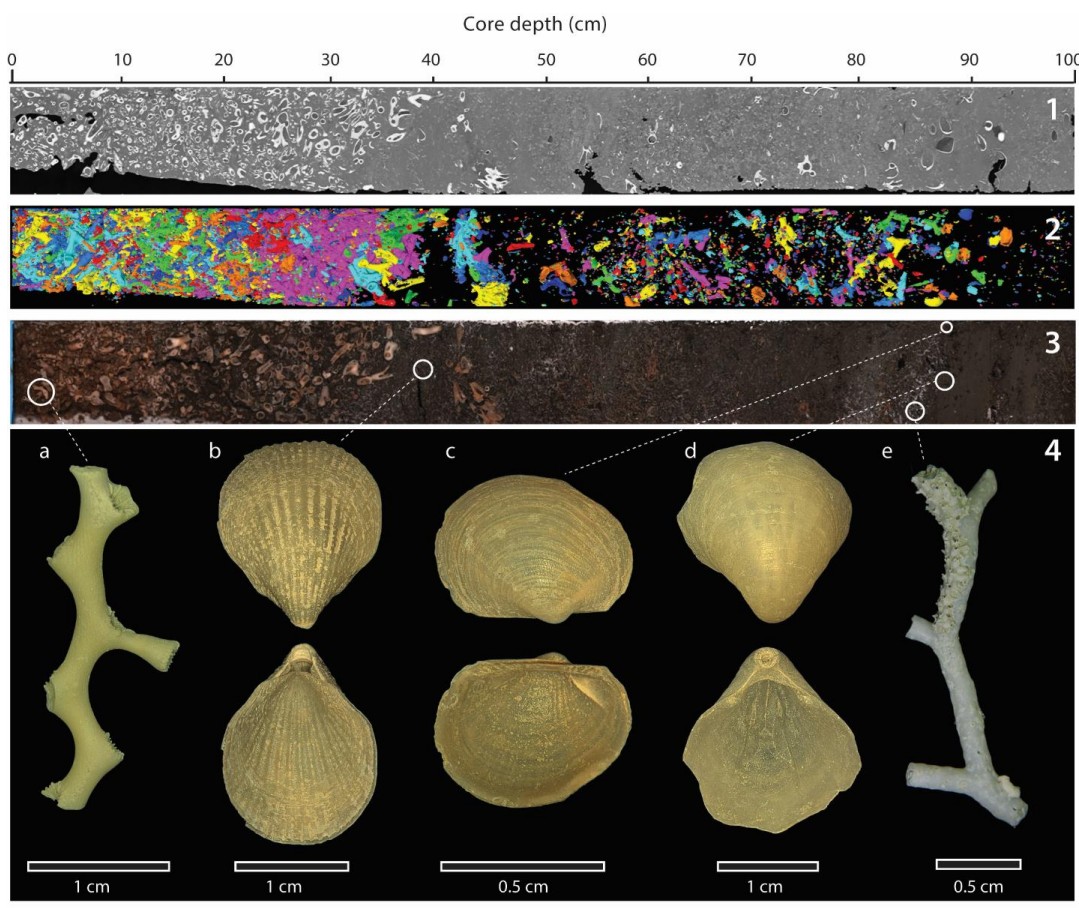


**Figure 6**. Example of a sediment core section showing the main macrofaunal components (section 1, 0-100 cm). (1) X-ray Computed Tomography imagery. (2) Three-dimensional reconstruction of coral fragments performed on X-ray Computed Tomography (CT) images. (3) Split-core high-resolution image. The white circles indicate the location of main macrofaunal components. (4) Main macrofaunal components: (a) the scleractinian coral *Madrepora oculata*, (b) the brachiopod *Terebratulina retusa*, (c) the bivalve *Bathyarca pectunculoides*, (d) the brachiopod *Gryphus vitreus*, (e) the bryozoan *Buskea dichotoma*.



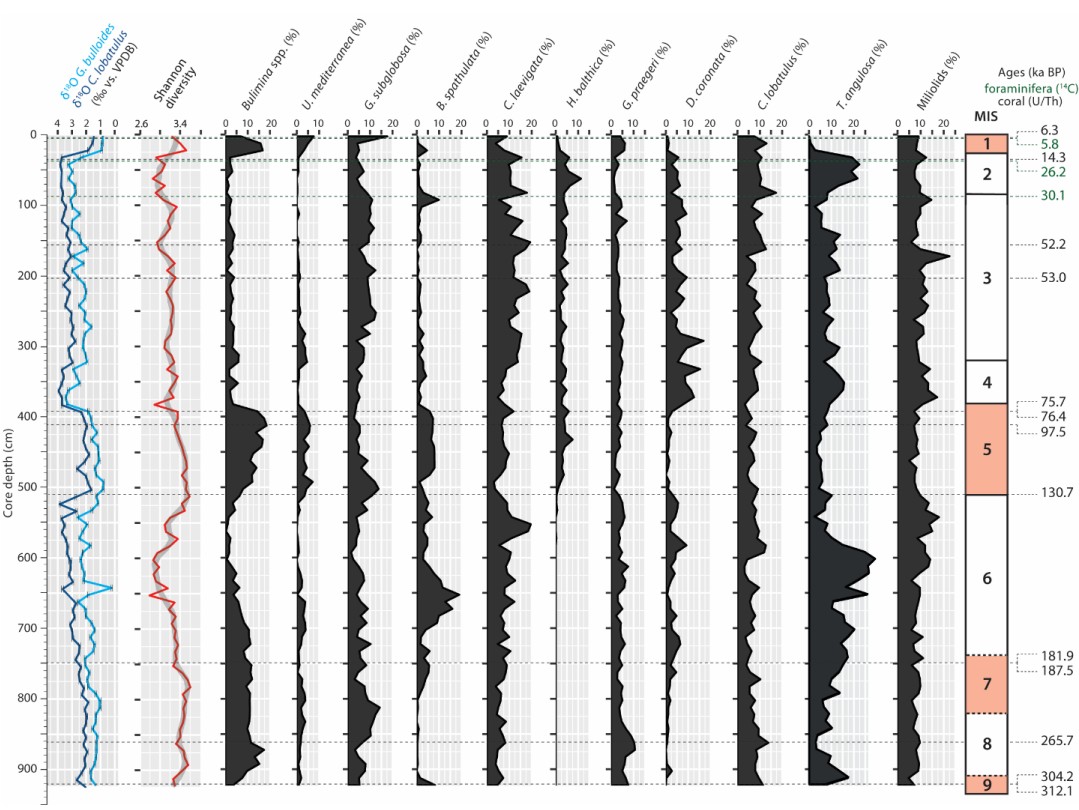

**Figure 7**. Distribution of main benthic foraminifera (expressed as the percentage of the total number of benthic foraminifera) and benthic foraminiferal Shannon diversity (the overlaid grey curve corresponds to the smoothed curve). The stratigraphy defined in Fig. 3 is given to the far right. The planktonic (*G. bulloides*) and benthic (*C. lobatulus*) δ18O records (‰, VPDB) are provided as supporting information.

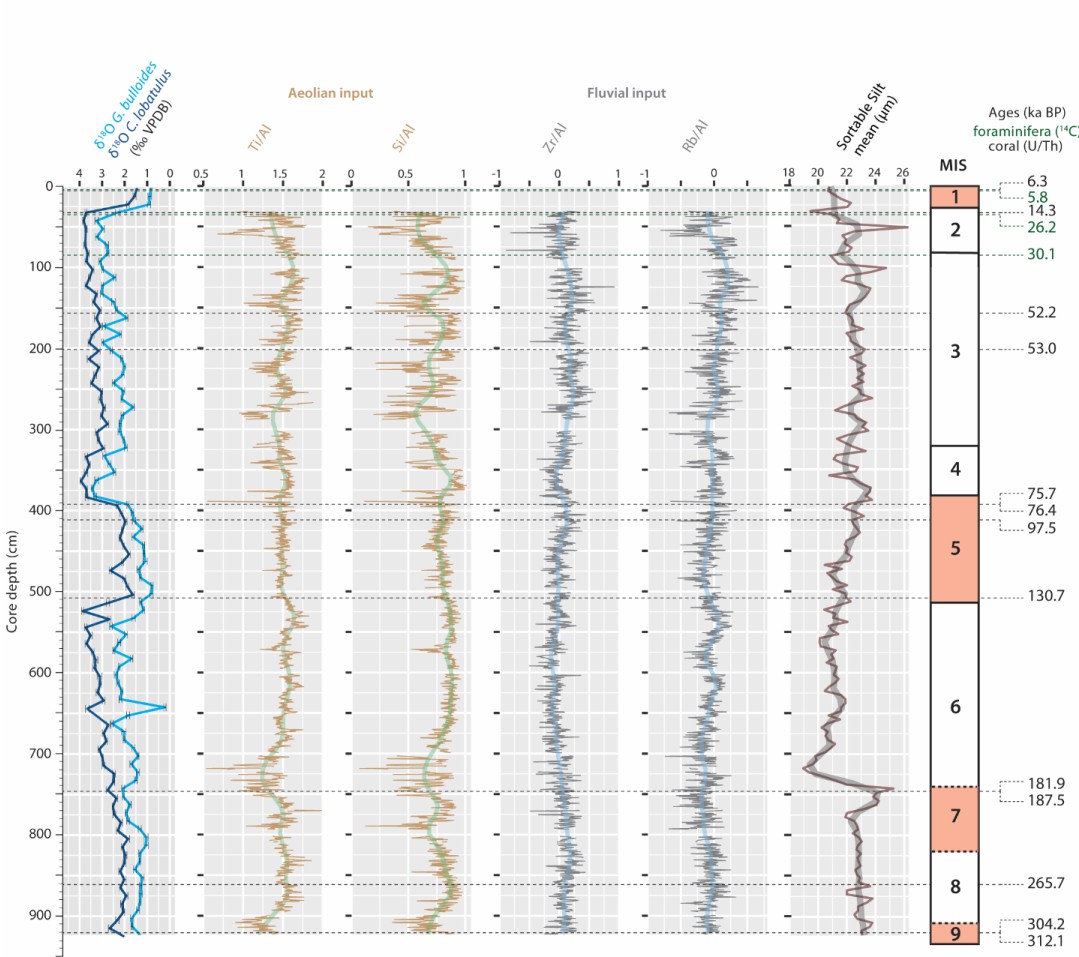


**Figure 8**. Log$_{10}$ titanium (Ti), silica (Si), zirconium (Zr) and rubidium (Rb) as aluminium (Al)-normalized ratios. These normalized elemental ratios are used as proxies for aeolian (Ti/Al and Si/Al) and fluvial input (Zr/Al and Rb/Al) at Brittlestar Ridge I. The stratigraphy defined in Fig. 3 is given to the far right. The planktonic (*G. bulloides*) and benthic (*C. lobatulus*) δ$^{18}$O

records (‰, VPDB), together with the mean size of the sortable silt fraction (the 10-63 μm grain size range, expressed in μm), are provided as supporting information.



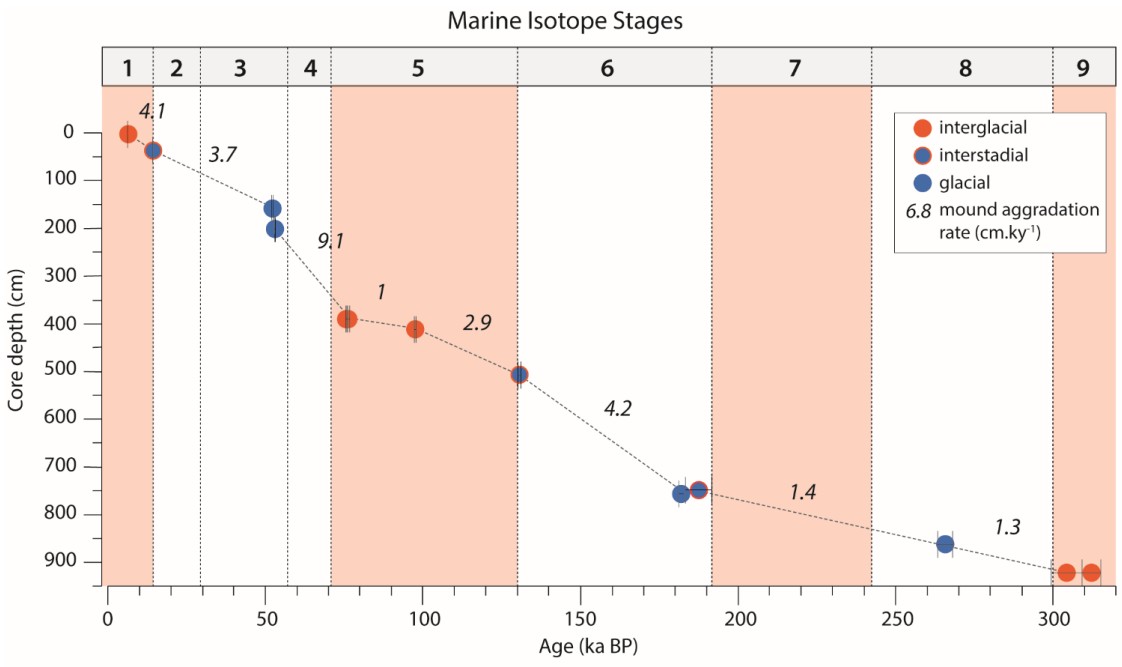

**Figure 9**. U-series coral ages (ka BP) vs. core depth (cm). Marine Isotope Stages (as defined by Lisiecki and Raymo, 2005) are
overlaid. All error bars are 2σ of the mean analytical uncertainty. The dashed lines between age-points (see legend) represent
average mound aggradation rates (in cm.ky⁻¹). Pink and white columns represent respectively interglacial and glacial periods.








**Figure 10**. Three dimensional diagrams and schematic models illustrating the differences between interglacial and glacial periods and the response of the benthic community at Brittlestar Ridge I. Water masses discussed in the text are illustrated (MAW: Modified Atlantic Water, LIW: Levantine Intermediate Water, ShW: Shelf Water; WMDW: Western Mediterranean Deep Water) as well as the Western Alboran Gyre (WAG) and Eastern Alboran Gyre (EAG). The flow strength of each water mass is depicted by the thickness of the arrows. The red star indicates the location of the East Melilla Coral Province. The position of the EAG and WAG is based on observations made by Lanoix (1974), La Violette (1983), and Viúdez and Tintoré (1995). Sea level of interglacial periods corresponds to the current sea level, whilst a 100 m lower sea level stand, following observations made by Rabineau et al. (2006), illustrates glacial periods. The LIW, ShW and WMDW flows follow the observations made by Ercilla et al. (2016). They have been simplified and thus do not represent their exact dynamics. The schematic models are not to scale, although relative depth limits between MAW and LIW have been respected. GEBCO_2019 gridded bathymetric data was used to construct the diagrams.



| LAB ID | Depth (cm) | $^{14}$C age (BP) | ±1σ | 2σ lower (cal years BP) | 2σ upper (cal years BP) | 2σ median (cal years BP) |
|---|---|---|---|---|---|---|
| ETH-87743 | 2 | 5777 | 25 | 5580 | 5920 | 5760 |
| ETH-87744 | 37 | 22811 | 78 | 25970 | 26530 | 26220 |
| ETH-87745 | 87 | 27587 | 124 | 30730 | 31160 | 30950 |


**Table 1.** Radiocarbon ages of epibenthic foraminifera (species selected: *Cibicides lobatulus*, *Cibicides refulgens* and *Discanomalina coronata*). Ages are corrected for a reservoir age of 390 ± 80 years (Siani et al., 2000).











| LAB ID | Depth (cm) | S[1] | Age (ka) | ± | Age[2] (ka) | ± | $^{238}$U (µg/g) | ± | $^{232}$Th (ng/g) | ± | $\delta^{234}$U (‰) | ± | $\delta^{234}$U$_i$ (‰) | ± |
|---|---|---|---|---|---|---|---|---|---|---|---|---|---|---|
| IUP- 8500 | 3 | M | 6.34 | 0.029 | 6.32 | 0.030 | 4.3377 | 0.00037 | 0.4311 | 0.00140 | 147.22 | 0.66 | 149.88 | 0.67 |
| IUP- 8501 | 36 | D | 14.31 | 0.047 | 14.30 | 0.049 | 3.4367 | 0.00012 | 0.3254 | 0.00084 | 145.33 | 0.64 | 151.33 | 0.67 |
| IUP- 8503 | 158 | D | 52.57 | 0.19 | 52.24 | 0.22 | 3.7330 | 0.00013 | 4.8320 | 0.01200 | 123.72 | 0.83 | 143.41 | 0.96 |
| IUP- 9310 | 201 | D | 53.07 | 0.12 | 53.04 | 0.13 | 2.6348 | 0.00008 | 0.3418 | 0.00059 | 126.01 | 0.45 | 146.39 | 0.53 |
| IUP- 8504 | 390 | D | 76.44 | 0.29 | 76.43 | 0.29 | 3.6896 | 0.00011 | 0.1328 | 0.00039 | 115.92 | 0.67 | 143.86 | 0.84 |
| IUP- 9183a | 390 | D | 75.66 | 0.20 | 75.65 | 0.17 | 3.7004 | 0.00016 | 0.1763 | 0.00046 | 117.75 | 0.49 | 145.83 | 0.61 |
| IUP- 9312 | 412 | D | 97.58 | 0.23 | 97.54 | 0.24 | 3.6265 | 0.00012 | 0.4572 | 0.00069 | 112.50 | 0.61 | 148.21 | 0.81 |
| IUP- 9313 | 507 | D | 130.7 | 0.45 | 130.7 | 0.46 | 3.4073 | 0.00015 | 0.3844 | 0.00072 | 105.96 | 0.85 | 153.30 | 1.25 |
| IUP- 8505 | 748 | D | 194.8 | 1.40 | 187.5 | 4.2 | 3.5659 | 0.00220 | 102.38[4] | 0.27000 | 95.01 | 0.84 | 161.40[3] | 2.40 |
| IUP- 9184b | 756 | D | 181.9 | 0.79 | 181.9 | 0.78 | 2.8694 | 0.00013 | 0.6018 | 0.00099 | 102.72 | 0.79 | 171.74[3] | 1.40 |
| IUP- 9314 | 862 | D | 265.7 | 2.10 | 265.7 | 2.4 | 3.4662 | 0.00018 | 0.6693 | 0.00150 | 70.40 | 1.10 | 149.10 | 2.60 |
| IUP- 8507 | 921 | D | 304.2 | 4.80 | 304.2 | 4.9 | 3.0370 | 0.00012 | 0.1176 | 0.00044 | 63.32 | 0.68 | 149.60 | 2.60 |
| IUP- 9185c | 921 | D | 312.1 | 3.40 | 312.1 | 3.0 | 3.3567 | 0.00016 | 0.2789 | 0.00061 | 58.58 | 0.77 | 141.50 | 2.20 |

**Table 2.** Details of Uranium-series isotope measurements (U/Th) carried out on 10 coral fragments. a) replicate of IUP-8504; b) replicate of IUP-8505; c) replicate of IUP-8507. Brackets denote activity ratios. All errors are 2σ of the mean analytical uncertainty. Ratios determined using a Th-U spike calibrated to a secular equilibrium reference material (HU-1 at the IUP). Uncorrected, closed-system age calculated using the decay constants of Jaffey et al. (1971) for $^{238}$U and Cheng et al. (2000) for $^{230}$Th and $^{234}$U. Ages are reported relative to the date of analysis, from year 2017 (IUP-8500 to IUP-8507) and year 2018 (other samples), and do not include uncertainties associated with decay constants. [1] Coral species: *M*: *Madrepora oculata*; *D*: *Desmophylum pertusum*. [2] Ages corrected for the contribution of initial $^{230}$Th based on an estimated seawater ($^{230}$Th/$^{232}$Th) activity ratio of 8 ± 4. [3] Significantly elevated $\delta^{234}$U$_i$ if compared to the present day seawater value of 146.8 ± 0.1 ‰, possibly indicative of U-series open system behaviour. [4] Samples containing strong residual amounts of non-carbonate contamination leading to high $^{232}$Th concentrations and thus age corrections.