# Peer review of "The influence of Atlantic climate variability on the long-term development of Mediterranean cold-water coral mounds (Alboran Sea, Melilla Mound Field)"

_Climate of the Past, 2020_

## Referee Comment (RC1) · Anonymous Referee #1 · 22 Jul 2020

In this manuscript, Fentimen et al present for the first time cold-water coral ages reaching beyond the last deglacial period. This is the first sediment core-based (i.e. +/- continuous) coral record from the Mediterranean reaching back to ∼300 kyr BP. These very interesting data set is compared to sedimentological, geochemical and micropaleontological data obtained from the matrix sediments of the same core in order to put the long-term development of the cold-water corals into a paleoenvironmental context. Unfortunately, this part has some serious problems (as outlined below) addressing the core stratigraphy and the interpretation of the applied proxies. There is a reason that

many other groups doing similar work try to use coral-barren cores taken nearby the coral mounds to reconstruct the paleoenvironment. As in most cases coral growth is intermittend and, thus, matrix sediments preserved among the corals only represent periods with coral growth. The often longer periods without coral growth are usually not preserved. In addition, some conceptional problems (e.g. linking highest coral percentages to best coral living conditions or explaining plankton blooms by enhanced fluvial input of degraded organic matter) and some "very brave" and/or selective interpretations of the data contribute to the perception of a paper that is far from being publishable. Thus, although I like the coral data set, I have to recommend to reject this manuscript. The most important step to solve the key problem would be to add considerable more coral ages.

Coral ages and stratigraphic interpretation In most cases, coral mound aggradation is intermittent, as also mentioned in this manuscript (lines 55 & 275, incl. some of the relevant references). That means, that short pulses of high mound aggradation (very vivid reef development) are interrupted by no growth periods (or maybe the presence of some individual corals, but no reefs) or even erosion, often resulting in a hiatus between core sections representing the vivid reef stages. This is also the case for Brittlestar Ridge (BR) 1, the study site for this manuscript, as has been shown for the last ∼14 kyr by several studies (Fink et al. 2013) including also work coming from the same group as this study (Stalder et al., 2015, 2018). Now, this common feature of coral mounds also applicable for – at least – the upper part of the BR 1 record, has been ignored in this study – it has not even been discussed with respect to the core presented here. In contrast, for core sections between individual coral ages, the stratigraphic interpretation is based on linear interpolation assuming that the core represents a continuous record. I strongly doubt the validity of this approach on this long coral mound core and to my knowledge there is no (or hardly any) long coral mound core reported that provides a continuous record. Actually, the interpretation provided here results in the observation that the by far highest observed mound aggradation rates in this record of 9.1 cm kyr-1 are indicated for MIS 4, which basically is in contrast to the main conclusion of the

authors that the corals preferentially lived during interglacials. On the other hand, this high aggradation rate for MIS 4 is not backed up by any dating referring to MIS 4; it is only based on linear interpolation. Earlier reported mound aggradation rates for BR 1 linked to well-established reefs in the Deglacial and the Holocene reach up to >400 cm kyr-1 (Fink et al., 2013; Stalder et al., 2015; Wienberg 2019). And even for periods with less well established reefs in the mid- and Late Holocene, mound aggradation rates are in the decimeter kyr-1 range. Thus, most likely, also the record presented here by Fentimen et al., would reveal a very different stratigraphic pattern with periods of high mound aggradation rates interrupted by hiatuses given that more effort would have been put into the dating of corals. This, definitely, would be needed, before this record is ready for publication.

Reading the coral mound record Fentimen et al. define the major coral build-up phases based on highest coral contents in their core. A detailed analyses of coral distribution as well as coral fragment orientation in a well-dated core from BR 1 revealed that highest mound aggradation rate ($\sim$400 cm kyr-1) coincides with rather low coral contents with coral fragments often preserved in an upright position (Titschack et al., 2016). Basically, this setting is interpreted as reflecting the partly preserved, fast growing reef being quickly filled up with sediments. In contrast, densely packed corals (usually flat laying) in a sediment core are often interpreted to reflect a coral rubble facies indicative of strongly reduced coral growth. In the core presented here, actually the highest aggradation rate in MIS 4 correlates with low coral contents ... Thus, the basic assumption used here (high coral content = best developed reef) is not valid.

Current reconstructions The authors use the sortable silt to infer past variations in current strength. This approach works very well in normal, current-controlled sediments. However, within a coral reef the current velocity is usually reduced compared to the coral-barren seabed. This effect is mentioned by the authors and their conclusion is that nevertheless relative variations in the sortable silt reflect relative variations in bottom current strength. However, this only would work out if the reef would be a constant

feature. But the authors also conclude that reef growth was quite variable through time. Consequently, the changing structure of the reef (from a large complex reef to few coral colonies) has a strong effect of the deceleration of the ambient bottom currents and, thus, on the sortable silt signal. Thus, only when the authors would have a good proxy for the state of the reef (and this cannot be the coral content) and if they could estimate the state-dependent effect of the reef on the bottom currents, finally an interpretation of the sortable silt data in respect to changing bottom currents might become possible. As yet, it is not possible. The authors added Fig. 5 to show the very good correlation between SSmean and SS% testifying the importance of the sorting process due to currents. This is not in contradiction to what has been said above: simply the reef state is another factor (in addition to the ambient bottom current strength) that has an effect of the actual current strength controlling sediment deposition within the reef. Consequently, the SSmean of the sediments deposited within the reef is not controlled by ambient bottom currents alone. Furthermore, when interpreting the data, the authors refer to a glacial/interglacial pattern with low glacial SSmean data. When looking at Fig. 4 I cannot see such a pattern. There are low SSmean values in MIS 6, but MIS 8 and 4 show rather high values and MIS 2 displays the full range of high and low values.

XRF data From the methodological point, it would be good to know, how the authors dealt with the effect of coral fragments on their element records. With a measurement taking every 5 mm, many of the individual measurements most likely will reflect the element composition within a single coral fragment. The authors refer to a post treatment of the data was carried out for data points affected by the uneven surface of the core, but what is with coral fragments being measured as part of the flat core surface? However, probably more importantly here is the interpretation of the data. To be honest, the ups and downs in the element ratio curves interpreted by the authors are not obvious to me. Instead, it reads as first there was the idea about the meaning of the data and then the data were interpreted accordingly. For instance, the authors refer to an overall increased fluvial and reduced aeolian input during interglacials (line 418) with lowest (highest) input of aeolian material during interglacials (glacials) (line

380). Looking at Fig. 8, it indicates (1) lowest but also highest Si/Al and Ti/Al ratios during glacials and (2) hardly any variability in the XRF data at all and MIS 5, 7, and 9 – the only interglacials covered by the XRF data do not show a clear trend as stated. Strongest variability is within MIS 3 with reaching highest and lowest values during this period. Actually, the strongest signal revealed by this data set is a decrease in aeolian AND fluvial input in MIS 2. The discussion, however, is oriented along the line either more fluvial and less aeolian or vice versa ... These data are used to back-up the conclusion that more humid conditions offer a better environment for the corals than more arid conditions. However, on a chronological much better resolved BR 1 record for the Early Holocene, Fink et al. (2013) exactly show the opposite with enhanced Si/Al ratio (more arid) corresponding to fastest mound aggradation (i.e. best living conditions for the corals). Without making any judgement, what is the right solution, I want to make the point here that the findings of the few papers dealing with cold-water corals in the region should be properly discussed.

TOC and productivity The TOC contents in the lower part (>250 cm) of this core range between 0.2% and 0.8% and get slightly higher in the upper part of the core reaching rarely above 1%. So overall, these variations are really minor. The increase towards the top, a feature common to very many marine TOC records, might reflect ongoing early diagenetic degradation of organic matter. In addition, the reported mound aggradation rates vary between 1 and 9.1 cm kyr – that is a factor of 9. Obviously, sedimentation rate has an effect of organic matter preservation and this might me important here seeing the range of aggradation rates. Furthermore, the authors invoke – partly severe – changes in bottom (and pore) water oxygenation – also this would affect organic matter preservation. So, using the only slightly varying TOC contents presented here as indicators for changing productivity (or organic matter flux), despite such other factors, is in my eyes over interpreting the data – unless the authors have good reasons to do so, but those are not presented. If the authors counted all the benthic foraminifera, why didn't they used the benthic foram accumulation rate as a productivity proxy? From line 397 onwards the link between TOC contents (the text partly refers to flux or export, however, no such information in terms of rates exists for this core) and benthic foram fauna composition leads to the conclusion that interglacials were more productive. However, TOC contents are highest in MIS 3 (and late MIS 5) and very low in MIS 7, 5e, and 1. As said before, it reads if first there was the interpretation and later on the data were analysed with the interpretation already preset. In line 644ff the authors refer to published knowledge that corals thrive on fresh organic matter. In the next sentence, the needed phytoplankton blooms in the study area are explained to be triggered by "input of degraded fluvial organic matter". Never heart about something like that. The river might bring (real) nutrients supporting the phytoplankton, but the phytoplankton cannot thrive on degraded organic matter. The link to the degraded OM is based on the statement of the authors that the OM in their sediment core is essentially of terrigenous origin (line 303). In a marine, productive setting like the Alboran Sea, this sounds rather unlikely . . .

Oxygen Line 438 ff refers to dysoxic conditions during interglacials that would have hampered coral proliferation as demonstrated by low mound aggradation rates. Well, the same group (and others) also published mound aggradation rates for BR 1 for the Early Holocene of >400 cm kyr-1 (Stalder et al., 2015) – that is 40 times higher as everything reported here. Obviously, corals can be very happy at BR 1 under such conditions . . . Furthermore, one of the main conclusion of the present manuscript is that the coral predominantly thrive under interglacial conditions . . . And, finally, later on it is argued that oxygen decreased at the transition from interglacials to glacials . . .

Specific comments Line 41: I would strongly suggest to differentiate between nutrients (nitrate, phosphate etc.) and food. In aphotic depths corals do not need any nutrients, but food. Later on in the text when you deal with river input, you really mean nutrients . . . Make a clear distinction between these terms. Line 73: see also Glogowski et al., 2015 Line 81: ref should be Lo Iacono et al. 2014 Line 96-104: not relevant here, skip Line 106: ref should be Fink et al. 2013 (first mention of BR) Line 134: "northwest" instead of northeast Line 141: "westward" instead of eastward Line 224: this means,

when a sample contained >300 specimen (e.g., 320) then it was split. In this case only 160 specimen were counted? Line 234: should read >2mm Line 303: you really think that the organic matter preserved in your core is of essential terrestrial origin? Later on, you use the TOC data as an indicator for productivity . . . Line 332ff: this is discussion, does not belong to results Line 390: the first sentence of the discussion refers to higher abundances of e.g. B. spathulata during interglacials. According to Fig. 7, their highest abundance is in MIS 6 and at the MIS 3/2 boundary . . . Line 451: "westward" instead of eastward Line 454ff: this is already documented by Wang et al. (2019) Line 455: How do you know? Any reference for this statement? Line 463: there is no section 6.1.1 Line 503: think about, if these mollusk layers may represent hiatuses . . . Line 511: there is no section 6.4 Line 527: how do you know about the quality of the organic matter? Line 563: Stronger contribution of nutrient-rich and well-ventilated West. Med Deep Water to the coral sites only can have supported bryozoan proliferation with respect to oxygen. Nutrients provided by the WMDW would be "real" nutrients such as nitrate, phosphate etc. which would be of no use for any organisms in these aphotic depths. Be more precise in using the terms nutrients and food! Line 568: cannot see "particularly unstable" isotope values during the last glacial in Fig. 4! Line 582ff: The link between high d18O and high Ti/Al and Si/Al ratios during the last glacial is not at all obvious, thus it cannot "confirm" (line 587) anything! Actually, between ∼100-200 cm you have high Si/Al ratios aligned with either high or low d18O values . . . Line 591-598: This Heinrich event discussion has no real relevance for this story . . . Line 600: where is the logic link?

---

## Author Comment (AC1) · 23 Aug 2020

Reply to Anonymous Referee #1

Manuscript title: The influence of Atlantic climate variability on the long-term development of Mediterranean cold-water coral mounds (Alboran Sea, Melilla Mound Field) submitted to Climate of the Past response by: Robin Fentimen et al.

In the following document, the responses to the comments made by Anonymous Referee #1 are addressed one by one.

[Figure]

Coral ages and stratigraphic interpretation

Comment Referee #1: In most cases, coral mound aggradation is intermittent, as also mentioned in this manuscript (lines 55 & 275, incl. some of the relevant references). That means, that short pulses of high mound aggradation (very vivid reef development) are interrupted by no growth periods (or maybe the presence of some individual corals, but no reefs) or even erosion, often resulting in a hiatus between core sections representing the vivid reef stages. This is also the case for Brittlestar Ridge (BR) 1, the study site for this manuscript, as has been shown for the last _14 kyr by several studies (Fink et al. 2013) including also work coming from the same group as this study (Stalder et al., 2015, 2018). Now, this common feature of coral mounds also applicable for – at least – the upper part of the BR 1 record, has been ignored in this study – it has not even been discussed with respect to the core presented here. In contrast, for core sections between individual coral ages, the stratigraphic interpretation is based on linear interpolation assuming that the core represents a continuous record. I strongly doubt the validity of this approach on this long coral mound core and to my knowledge there is no (or hardly any) long coral mound core reported that provides a continuous record.

Response: We are well aware that coral mound aggradation at Brittlestar Ridge 1 is intermittent, as supported by the publications mentioned by Reviewer #1. We also fully agree that there are very few, if any, continuous records provided by coral mounds (especially covering 300 ky). However, it is not correct to state that we have ignored this common feature, as written at Line 275 and 276: "In contrast, constructing a continuous age model based on stable isotope records is generally considered untrustworthy for cores collected from coral mounds since sedimentation is intermittent (Dorschel et al., 2005)". We have, at no point in this study, considered or hinted that the core represents a continuous record. We are well aware that the record is discontinuous, hence the decision to plot the different sedimentological, micropaleontological and geochemical records against depth, and not against an age model (see for example Fig. 3). As

described in section 4.1 (Lines 264 to 292), corals and foraminifera were selected at major sedimentological boundaries (clear facies changes, e.g transition from a Lophelia pertusa (coral) to a Buskea dichotoma (bryozoan) horizon). In conjunction with the stable oxygen isotope record, we then defined the chronology and limited important time intervals (interglacial vs. glacial periods). We do not assume that the record is continuous between two coral ages; we simply isolate and define key time intervals corresponding to Marine Isotope Stages in order to understand the major environmental changes having affected Brittlestar Ridge 1 (this thanks to the wide variety of proxies used). It has to be pointed out here that the core is only 926 cm long, and covers ca. 300 ky. Thus, it presents a very condensed record, nothing like the cores described in the studies by Fink et al. (2013), Stalder et al. (2015, 2018) or Fentimen et al. (2020) which cover only the last 15 ky (for core lengths ranging from 350 to 490 cm). In comparison, the record presented here is, in this sense, more comparable to core GeoB 6730-1 taken from the Propeller Mound region (Northeast Atlantic) which is 360 cm long and covers ca. 207 ky (Dorschel et al., 2005; Rüggeberg et al., 2007). Thus, we followed a similar approach in order to define the stratigraphy as the one used by Rüggeberg et al. (2007), i.e. plotting the different datasets against depth and not age, and being aware of the fact that the core section can be discontinuous. We can add a sentence to state more clearly that we do not consider the record to be continuous.

Comment Referee #1: Actually, the interpretation provided here results in the observation that the by far highest observed mound aggradation rates in this record of 9.1 cm kyr-1 are indicated for MIS 4, which basically is in contrast to the main conclusion of the authors that the corals preferentially lived during interglacials.

Response: We do not agree with Reviewer #1 on this point. Highest mound aggradation rates (which are, as correctly pointed by Reviewer #1, still comparatively very low) during MIS 4 are not incompatible with the preference of corals for interglacial periods. In core MD13-3462G, the bryozoan B. dichotoma appears to play an equally important role on mound build-up as cold-water corals (see macrofaunal quantifications, Figure

3). This is developed in section 5.4.2 (Lines 654 to 680). This is one main conclusion of this study, that the bryozoan B. dichotoma plays an important role on mound development (when considering mound development over the last 300 ky, not only the last deglacial). Thus, at site MD13-3462G, mound aggradation does not go hand in hand with coral reef aggradation.

Comment Referee #1: On the other hand, this high aggradation rate for MIS 4 is not backed up by any dating referring to MIS 4; it is only based on linear interpolation. Earlier reported mound aggradation rates for BR 1 linked to well-established reefs in the Deglacial and the Holocene reach up to >400 cm kyr-1 (Fink et al., 2013; Stalder et al., 2015; Wienberg 2019). And - even for periods with less well established reefs in the mid- and Late Holocene, mound aggradation rates are in the decimeter kyr-1 range. Thus, most likely, also the record presented here by Fentimen et al., would reveal a very different stratigraphic pattern with periods of high mound aggradation rates interrupted by hiatuses given that more effort would have been put into the dating of corals. This, definitely, would be needed, before this record is ready for publication.

Response: Indeed, only the beginning of MIS 4 is backed up by coral dating. The end of MIS 4 is identified thanks to the stable oxygen isotope record (transition from high to low values, see Figure 3 and section 4.1). As pointed out in section 4.1, coral ages at the upper and lower boundaries of coral build-up phases in core MD13 3462G (e.g. at 390 and 507 cm depth) correspond to changes in the stable oxygen isotope records, which in turn match the changes between Marine Isotope Stages (Lisiecki and Raymo, 2005). Thus, the significant change in the stable oxygen isotope record at 320 cm can be attributed to an important change, which we interpret as the end of MIS 4. This change is also observed in other proxies, such as benthic foraminiferal assemblages and the macrofaunal distribution. It is the combination of all these proxies, in addition to the stable oxygen isotope records, that allows us to interpret this boundary as the end of MIS 4. We believe that combining different proxies (stable oxygen isotope record, facies analyses, foraminiferal asssemblages, macrofaunal abundances) proves to be

stronger than basing interpretations on one single dating (although we recognize that these are also necessary)

We agree that the addition of supplementary coral ages would reveal more and shorter periods of higher mound aggradation rates interrupted by hiatuses. However, the average mound aggradation rate for a given longer time period (for example MIS 5) would not be any different (it would just show a higher variability). Again, we believe that the time scale considered in this study (300 ky for a 920 cm long core), does not allow for the precise study of mound aggradation rates that Reviewer #1 suggests. This is why this manuscript concentrates rather on characterizing changes between interglacial and glacial periods and describing average mound aggradation rates for those longer time periods. Moreover, the Holocene and last interglacial are very reduced in core MD13-3462G compared to other published records (Fink et al., 2013; Stalder et al., 2015, 2018). This is a peculiarity of core MD13-3462G in comparison to other records that we are aware of (thus the decision to focus on older parts of the core).

We insist that calculating mound aggradation rates as precisely as suggested by Reviewer #1, and as done by previous studies focusing only on the last Deglacial and the Holocene (Fink et al., 2013; Stalder et al., 2015), is - in this case - impossible. However, the coral ages presented in this work, are sufficient to delimit interglacial and glacial periods, and hence to characterize the environment at Brittlestar Ridge I during these periods (which is the goal of the study).

Comment Referee #1: Reading the coral mound record Fentimen et al. define the major coral build-up phases based on highest coral contents in their core. A detailed analyses of coral distribution as well as coral fragment orientation in a well-dated core from BR 1 revealed that highest mound aggradation rate (_400 cm kyr-1) coincides with rather low coral contents with coral fragments often preserved in an upright position (Titschack et al., 2016). Basically, this setting is interpreted as reflecting the partly preserved, fast growing reef being quickly filled up with sediments. In contrast, densely packed corals (usually flat laying) in a sediment core are often interpreted to reflect a

coral rubble facies indicative of strongly reduced coral growth. In the core presented here, actually the highest aggradation rate in MIS 4 correlates with low coral contents . . . Thus, the basic assumption used here (high coral content = best developed reef) is not valid.

Response: To discuss the orientation of coral fragments (cfr. upright position or coral rubble) and to avoid any discussion in terms of coral rubble versus in-situ coral framework, we are adding the visualization of the CT-fragments bigger than 2 cm to the manuscript. We are aware of the fact that the highest cold-water coral content cannot be always interpreted as the best-developed reef. The reviewer wrongly states that the authors are making this assumption. Coral content has been compared with other microfaunal observations. Although coral content is varying in the studied core between glacials and interglacials, the overall mound aggradation rates are low (seeing the mound as a system, cfr. sediments + microfauna + faunal content). Moreover, mound aggradation rates are similar for glacials and interglacials (= one of the conclusions of the manuscript). So – indeed – mound aggradation rates and coral content are not comparable, as correctly stated by the reviewer and by Titschack et al., 2016.

This being said, the statement that the work of Titschack et al. (2016) evidences that highest mound aggradation rates coincide with rather low coral contents is disputable and might be dependent upon regional and temporal differences. The unit with the highest mean aggradation rate at BR 1 in the study of Titschack et al. (530 ky.cm-1; Unit B; Titschack et al., 2016) demonstrates in some parts of the section a coral content of up to 30 % (average ca. 10 %). In this study (core MD13-3462G), a coral content of up to 30 % is considerably higher to what is observed during MIS 4 or MIS 2 (maximum 5 to 10 %) and is actually close to the maximum coral content documented in our study during interglacials (Fig. 3). When coral content is as low as 5 % and when bryozoan content reaches over 60 % (for example during MIS 2), we believe that it is correct to say that bryozoans are thriving and corals are not.

Moreover, we believe that the comparison to the work of Titschack et al. (2016) is

hindered by the time interval considered in their study in comparison to the one considered in this manuscript. Titschack et al. (2016) considered a 447 cm long core, covering the time interval between 11.2 and 9.8 ky, while this study is focusing on a 926 cm long, covering 300 ky and studying mainly the time interval between 14.3 and 180 ka.

Current reconstructions

Comment Referee #1: The authors use the sortable silt to infer past variations in current strength. This approach works very well in normal, current-controlled sediments. However, within a coral reef the current velocity is usually reduced compared to the coral-barren seabed. This effect is mentioned by the authors and their conclusion is that nevertheless relative variations in the sortable silt reflect relative variations in bottom current strength. However, this only would work out if the reef would be a constant feature. But the authors also conclude that reef growth was quite variable through time. Consequently, the changing structure of the reef (from a large complex reef to few coral colonies) has a strong effect of the deceleration of the ambient bottom currents and, thus, on the sortable silt signal. Thus, only when the authors would have a good proxy for the state of the reef (and this cannot be the coral content) and if they could estimate the state-dependent effect of the reef on the bottom currents, finally an interpretation of the sortable silt data in respect to changing bottom currents might become possible. As yet, it is not possible. The authors added Fig. 5 to show the very good correlation between SSmean and SS% testifying the importance of the sorting process due to currents. This is not in contradiction to what has been said above: simply the reef state is another factor (in addition to the ambient bottom current strength) that has an effect of the actual current strength controlling sediment deposition within the reef. Consequently, the SSmean of the sediments deposited within the reef is not controlled by ambient bottom currents alone. Furthermore, when interpreting the data, the authors refer to a glacial/interglacial pattern with low glacial SSmean data. When looking at Fig. 4 I cannot see such a pattern. There are low SSmean values in MIS 6, but MIS 8

and 4 show rather high values and MIS 2 displays the full range of high and low values.

Response: We agree and are well aware that the SS mean of the sediments is not controlled by ambient currents alone (see section 3.4, Lines 205 to 211). Numerous studies have shown that the coral framework results in a local reduction of current velocity (as stated by Reviewer #1). Thus, the current velocity calculated thanks to the SS mean is actually an underestimation of the ambient bottom current (this is shown by a number of studies also, e.g. Huvenne et al., 2009; Titschack et al., 2009; Fentimen et al., 2020). Taking this into account, it is possible to compare relatively reef build-up phases to phases without any or very few cold-water corals (for example between MIS 4 and 5 or MIS 2 and MIS 1, see Figure 3). We agree with Reviewer #1 that using the SS mean alone and in a core with little coral content variations (e.g. between 20 and 30 %) would require a good proxy for the state of the reef and the effect of the reef on bottom currents. However, in this manuscript we aim to compare coral build-up phases (for example MIS 5) with intervals when corals are absent or near to absent (e.g. MIS 4 and 2).

Moreover, in this study the SS mean is used as supporting information for interpretations essentially based on benthic foraminiferal assemblages and the macrofauna. Indeed, the benthic foraminiferal assemblages and the macrofauna are also a valuable proxy when reconstructing bottom current velocities (this is stated in section 3.4). For example, we utilize the abundance of the benthic foraminifera species Trifarina angulosa (Lines 604 to 611), Cassidulina laevigata (Lines 517 to 523) and the abundance of the bryozoan Buskea dichotoma and the brachiopod Gryphus vitreus (Lines 502 to 504, Line 539) to identify respectively increasing and decreasing bottom currents. Indeed, the infaunal dwelling Trifarina angulosa is widely documented to live in areas dominated by strong currents and to resist winnowing (Mackensen et al., 1995; Schönfeld, 2002; Margreth et al., 2009, references in the manuscript). The SS mean is not used alone and is in support of the other proxies used to reconstruct bottom currents. It is important to point out that the interpretations made in this manuscript are

dominantly built around the information gathered thanks to the benthic foraminiferal assemblages (see Discussion section, e.g. Lines 389 to 444, 514 to 536). However, Reviewer #1 made no remarks or comments concerning interpretations related to benthic foraminiferal assemblages, although it is the most widely used proxy in this work and a solid dataset within a coral mound environment. We believe that this lack of consideration of benthic foraminiferal assemblages may have led to a wrong understanding of certain interpretations made in this manuscript. This manuscript presents for the first time a high resolution assessment of benthic foraminiferal assemblages at Brittlestar Ridge 1 (total number of samples: 92). In comparison, previously published work in the area only presented a reduced dataset (Stalder et al., 2015: 29 samples, Stalder et al., 2018: 38).

XRF data

Comment Referee #1: From the methodological point, it would be good to know, how the authors dealt with the effect of coral fragments on their element records. With a measurement taking every 5 mm, many of the individual measurements most likely will reflect the element composition within a single coral fragment. The authors refer to a post treatment of the data was carried out for data points affected by the uneven surface of the core, but what is with coral fragments being measured as part of the flat core surface?

Response: It is indeed not detailed in the manuscript how we dealt with the effect of coral fragments on elemental records. However, the authors are fully aware of this effect. The presence of aragonite and calcite is diluting the background sediments. Using ratios of element intensities instead of intensities of single elements, may account partly for those so-called dilution effects (f.e. amongst others Weltje et al., 2015; Weltje and Tallingii, 2008). Spot analyses on coral-skeletons are – indeed – explaining the much higher small-scaled peaks – especially - in coral-rich units. For this reason, authors were calculating the running mean (using the Loess smoothing in R) on the datasets reflecting the overall trends. To read more about the Loess smoothing in R,

please read for examples the following document: "W. S. Cleveland, E. Grosse and W. M. Shyu (1992): Local regression models. Chapter 8 of Statistical Models in S eds. J.M. Chambers and T.J. Hastie, Wadsworth & Brooks/Cole". For the background sediments, normalization with Al has been performed because it behaves conservatively.

Comment Referee #1: However, probably more importantly here is the interpretation of the data. To be honest, the ups and downs in the element ratio curves interpreted by the authors are not obvious to me. Instead, it reads as first there was the idea about the meaning of the data and then the data were interpreted accordingly. For instance, the authors refer to an overall increased fluvial and reduced aeolian input during interglacials (line 418) with lowest (highest) input of aeolian material during interglacials (glacials) (line 380). Looking at Fig. 8, it indicates (1) lowest but also highest Si/Al and Ti/Al ratios during glacials and (2) hardly any variability in the XRF data at all and MIS 5, 7, and 9 – the only interglacials covered by the XRF data do not show a clear trend as stated. Strongest variability is within MIS 3 with reaching highest and lowest values during this period. Actually, the strongest signal revealed by this data set is a decrease in aeolian AND fluvial input in MIS 2. The discussion, however, is oriented along the line either more fluvial and less aeolian or vice versa . . . These data are used to back-up the conclusion that more humid conditions offer a better environment for the corals than more arid conditions. However, on a chronological much better resolved BR 1 record for the Early Holocene, Fink et al. (2013) exactly show the opposite with enhanced Si/Al ratio (more arid) corresponding to fastest mound aggradation (i.e. best living conditions for the corals). Without making any judgement, what is the right solution, I want to make the point here that the findings of the few papers dealing with cold-water corals in the region should be properly discussed.

Response: We agree with Reviewer #1 that this part needs to be reworked. Indeed, highest but also lowest Si/Al and Ti/Al ratios are recorded during glacial periods. This will be corrected in the revised version of the manuscript. Furthermore, the decrease in aeolian but also fluvial input during MIS 2 needs more attention. However, some points

addressed are wrong: at no point do we state that interglacials show a clear trend (the word "clear" is not used). The observation that strongest variability within the XRF data is found during MIS 3 is correct and stated twice in the manuscript: Line 383 to 385: "In the same way as for Ti/Al and Si/Al records, Zr/Al and Rb/Al ratios demonstrate an important variability during MIS 3, in comparison to other periods where the records are comparatively stable" and Line 511, and is further discussed in section 5.2.3. The conclusion that more humid conditions (increased fluvial input) during interglacials are a more suitable environment for corals is essentially based on the benthic foraminiferal assemblage composition. It is noticeably based on the high abundances of the infaunal Buliminids, Uvigerina mediterranea and Bolivina spathulata during interglacials (Figure 7). These species are, in the Mediterranean, dominant in the vicinity of the Po and Rhone river deltas (Jorissen, 1987; Mojtahid et al., 2009). Nevertheless, we do agree that the XRF records need to be described and interpreted with more reserve. Thus, also following the specific comment below (i.e."the link between high d18O and high Ti/Al and Si/Al ratios during the last glacial is not at all obvious"), discussion linked to the XRF records has been reduced and reworked.

TOC and productivity

Comment Referee #1: The TOC contents in the lower part (>250 cm) of this core range between 0.2% and 0.8% and get slightly higher in the upper part of the core reaching rarely above 1%. So overall, these variations are really minor. The increase towards the top, a feature common to very many marine TOC records, might reflect on-going early diagenetic degradation of organic matter. In addition, the reported mound aggradation rates vary between 1 and 9.1 cm kyr – that is a factor of 9. Obviously, sedimentation rate has an effect of organic matter preservation and this might me important here seeing the range of aggradation rates. Furthermore, the authors invoke – partly severe – changes in bottom (and pore) water oxygenation – also this would affect organic matter preservation. So, using the only slightly varying TOC contents presented here as indicators for changing productivity (or organic matter flux), despite

such other factors, is in my eyes over interpreting the data – unless the authors have good reasons to do so, but those are not presented.

Response: We understand the interrogations addressed by Reviewer #1 concerning the low TOC content throughout the core and the potential effects of mound aggradation and water oxygenation. These interrogations have been discussed and pointed out for some time (Doyle and Garrels, 1985). However, we had to our knowledge, no means to counteract and take into account these potential effects. This is why the TOC contents are in the manuscript only scarcely used and only in support of interpretations made thanks to benthic foraminiferal assemblages and macrofauna (TOC contents are only mentioned twice in the entire discussion, and always in support of other datasets such as foraminiferal assemblages: Lines 399 and 501). The abundance of infaunal benthic foraminifera, e.g. Uvigerina mediterranea and Bulimina spp., are indeed a more trustworthy proxy for increased productivity than TOC contents. Nonetheless, the combined use of foraminifera, macrofauna and TOC content, gives a good indication of productivity. Low TOC contents are, to our knowledge, common in coral mound records. If Reviewer #1 prefers, it is possible not to address TOC content or to indicate more clearly that this proxy is considered untrustworthy and is only used in support to interpretations made thanks to benthic foraminiferal assemblages and macrofauna.

Comment Referee #1: If the authors counted all the benthic foraminifera, why didn't they used the benthic foram accumulation rate as a productivity proxy?

Response: We did not use the benthic foraminiferal accumulation because we consider it as an untrustworthy productivity proxy. A number of micropaleontological studies have pointed this out (see the review Jorissen et al., 2007). Noticeably, Naidu and Malmgren (1995) showed that in low oxygen environments, BFAR (benthic foram accumulation rate) does not reflect surface-water productivity. Since we suspect that the seafloor at BR1 was at times depleted in oxygen, we further avoided to use the BFAR as a productivity proxy. Moreover, taphonomic processes, which directly impact BFAR, are not well constrained (see for example Murray, 2006; Fentimen et al., 2020).

Overall, since benthic foraminifera were identified at species level throughout the core, we prefer to consider benthic foraminiferal assemblages rather than the BFAR (which ignores species composition).

Comment Referee #1: From line 397 onwards the link between TOC contents (the text partly refers to flux or ex- port, however, no such information in terms of rates exists for this core) and benthic foram fauna composition leads to the conclusion that interglacials were more productive. However, TOC contents are highest in MIS 3 (and late MIS 5) and very low in MIS 7, 5e, and 1. As said before, it reads if first there was the interpretation and later on the data were analysed with the interpretation already preset. In line 644ff the authors refer to published knowledge that corals thrive on fresh organic matter. In the next sentence, the needed phytoplankton blooms in the study area are explained to be triggered by "input of degraded fluvial organic matter". Never heart about something like that. The river might bring (real) nutrients supporting the phytoplankton, but the phytoplankton cannot thrive on degraded organic matter. The link to the degraded OM is based on the statement of the authors that the OM in their sediment core is essentially of terrigenous origin (line 303). In a marine, productive setting like the Alboran Sea, this sounds rather unlikely . . .

Response: There is no mention in the manuscript of MIS 5e, we do not believe that the stratigraphy presented in this manuscript allows to identify sub-stages. On average, the TOC content during MIS 5 is indeed higher than during MIS 6 or MIS 4 (see Figure 4). So this statement is not incorrect.

We agree that such a pattern is not as clear for MIS 7, although an increase can be observed. Corrections will be made to indicate that this pattern is not applicable for MIS 7 and 1 (possibly due to the biases indicated above). Details will be added to section 5.2.3 mentioning that MIS 3 shows high TOC content, this actually matches the conclusions made in the first paragraph of section 5.2.3 that highlight that corals and the benthic foraminiferal community positively responded to short phases of increased surface productivity related to important continental runoff during MIS 3.

We agree that the phrasing at line 644 is awkward, this will be corrected. Correction: "In contrast, corals at BRI are likely supplied by plankton blooms triggered by river-transported nutrients during interglacial times". The statement that organic matter in the sediment core is of terrigenous origin is based on the high oxygen index (OI) values. This can be observed in the Supplementary data (this may not have been added to the online submission, if this is the case we apologize for the inconvenience, Lines 301 to 303). Plotting a Van Krevelen index (see for example, Espitalié et al., 1985), i.e. the Hydrogen Index (HI) against the Oxygen Index (OI), demonstrates that the organic matter is indeed of terrestrial origin and well oxidized (see figures below). Plotting a pseudo Van Krevelen index (i.e. OI vs. Tmax) also indicates that the organic matter is of terrestrial deltaic origin (see below).

Figure 1. Van Krevelen diagram

Figure 2. Pseudo Van Krevelen index

Oxygen

Comment Reviewer #1: Line 438 ff refers to dysoxic conditions during interglacials that would have hampered coral proliferation as demonstrated by low mound aggradation rates. Well, the same group (and others) also published mound aggradation rates for BR 1 for the Early Holocene of >400 cm kyr-1 (Stalder et al., 2015) – that is 40 times higher as everything reported here. Obviously, corals can be very happy at BR 1 under such conditions . . .

Response: Indeed, Stalder et al. (2015) and Fink et al. (2013) published mound aggradation rates for BR 1 of over > 400 cm.ky-1, respectively 457 cm.ky-1 (between 13023 and 12717 ka) and 416 cm.ky-1 (between 12874 and 11240). So these rates are not for the Early Holocene if we accept that the Holocene Epoch started at 11.7 b2k (see Walker et al., 2018). This stratigraphy has been accepted by the International Commission on Stratigraphy, and formally ratified by the Executive Committee of the International Union of Geological Sciences on 14th June 2018 (Walker et al., 2018).

Cold-water coral communities are nowadays rare at BR 1 (Hebbeln et al., 2019).

Moreover, we believe again that the time scale considered in this study (300 ky for a 920 cm long core) allows to identify more long-term environmental changes than those from Fink et al. (2013), Stalder et al. (2015) or Fentimen et al. (2020). The time-scale covered by these studies allow to identify precisely short but rapid periods of mound aggradation. This study does not aim to do this, but rather to look at the wider picture (see previous paragraphs). Again, the mound aggradation rates presented are average values. We believe that a core covering the last two interglacials allows to draw more solid conclusions about the impact of environmental changes on mound development than a precise study of the last 15 ky.

We would like to add that is the opinion of the authors that, besides general trends, very local environmental variations at BR 1 may account for important differences between cores recovered in the area. A manuscript discussing such local differences is in preparation at the moment.

Comment Reviewer #1: Furthermore, one of the main conclusion of the present manuscript is that the coral predominantly thrive under interglacial conditions . . . And, finally, later on it is argued that oxygen decreased at the transition from interglacials to glacials . . .

Response: This is not a correct citation of the manuscript. For example, conclusion section (Line 705) "(. . .) corals did not thrive but rather developed under stressful environmental conditions at Brittlestar Ridge I". The term "thrive" is not associated to coral development at BR 1 in the manuscript. Or Line 671: "CWCs did not thrive at the site of core MD13-3462G but rather developed under stressful, possibly dysoxic, environmental conditions".

It is not written that oxygen decreased at the transition from interglacials to glacials, but rather that "These results suggest that transition phases between interglacial and glacial periods were characterized by winnowing at the seafloor (Line 609)" and that

"The seafloor was possibly depleted in oxygen at the end of interglacial phases (Line 690)". However, this sentence is possibly awkward and can be reworked to insist that the end of interglacial periods were possibly marked by oxygen depletion, whilst the beginnings of glacial periods were rather marked by increased bottom currents.

Specific comments

Reviewer #1: Line 41: I would strongly suggest to differentiate between nutrients (nitrate, phosphate etc.) and food. In aphotic depths corals do not need any nutrients, but food. Later on in the text when you deal with river input, you really mean nutrients ... Make a clear distinction between these terms. Line 73: see also Glogowski et al., 2015

Response: Corrections made

Reviewer #1: Line 81: ref should be Lo Iacono et al. 2014

Response: Reference to Lo Iacono et al. 2014 added

Reviewer #1: Line 96-104: not relevant here, skip

Response: This part introduces the overall geological setting of the study area. It explains how such ridges as BR 1 /the study site) were shaped. Although possibly not of interest and redundant for readers acquainted with the area, we believe that this information is interesting for readers not acquainted with the geology of the Western Mediterranean (especially since this work is directed at an international audience, not exclusively European researchers). Thus, we would prefer to keep this as such (especially since this part is short, i.e. 8 lines).

Reviewer #1: Line 106: ref should be Fink et al. 2013 (first mention of BR)

Response: Correction made and reference added. Although we agree that Brittlestar Ridges were mentioned and named by Fink et al. 2013, Comas et al. (2009) do also mention ridges in the area: "On the seafloor, mounds appear as ridge-like buildups

100–250 m wide, 2-6 km long, and 20–60 m (up to 100 m) high above the seabed". Thus this reference should still be mentioned. We also noticed that the reference to Comas et al. (2009) was missing in the reference list. This has been added.

Reviewer #1: Line 134: "northwest" instead of northeast

Response: Correction made.

Reviewer #1: Line 141: "westward" instead of eastward

Response: correction made "westward circulating branch of the Eastern Alboran Gyre"

Reviewer #1: Line 224: this means, when a sample contained >300 specimen (e.g., 320) then it was split. In this case only 160 specimen were counted?

Response: Indeed this sentence is wrong and needs to be corrected. It now reads: "If the residue contained more than 600 specimens, it was split using a dry microsplitter." Samples were split if the residue contained over 600 foraminifera (targeted number of 300 individuals).

Reviewer #1: Line 234: should read >2mm

Response: correction made

Reviewer #1: Line 303: you really think that the organic matter preserved in your core is of essential terrestrial origin? Later on, you use the TOC data as an indicator for productivity . . .

Response: Yes the RockEval results support this, see response page 7 (and attached Van Krevelen diagram). At no point in the manuscript is TOC data used as an indicator for productivity. In the discussion section, the mention to TOC can be found twice: Line 399: "The overall higher TOC levels during interglacials confirm that the sediment during these periods was relatively enriched in organic matter in comparison to glacial periods", and Line 501: "Low SSmean values and reduced TOC content in the sediment confirm that glacial periods were marked by weak bottom current velocities and

organic matter flux".

Reviewer #1: Line 332: this is discussion, does not belong to results

Response: Which sentence does Reviewer #1 refer to? We do not think that either of the two sentences are out of place in the results section.

Reviewer #1: Line 390: the first sentence of the discussion refers to higher abundances of e.g. B. spathulata during interglacials. According to Fig. 7, their highest abundance is in MIS 6 and at the MIS 3/2 boundary . . .

Response: We agree that B. spathulata shows lower abundances than Buliminids and U. mediterranea (see Figure 7). However, B. spathulata do increase during MIS 5, 7 and 9 (when compared to values at the onset and end of these stages, see Figure 7). Thus, the sentence has been reworked, it now reads: "During interglacial periods, benthic foraminiferal assemblages are marked by high abundances of the infaunal Bulimina spp., U. mediterranea and to a lesser extent B. spathulata". It is correct that B. spathulata reaches highest abundance during MIS 6, a mention to this is now added in the results section. We took the decision not to discuss this point, since the discussion is already long and we had to focus on main trends. Other benthic foraminiferal species show interesting abundance patterns but are not graphically represented here (166 species were in all identified). The species graphically represented here were chosen because they are good discriminating species, they represent dominant species in the core, and their ecology is well constrained (which is not the case for all species, making their use as an environmental proxy more complex and limited). However, the complete benthic foraminiferal dataset is available as supplementary information.

Reviewer #1: Line 451: "westward" instead of eastward

Response: correction made "westward circulating branch of the Eastern Alboran Gyre"

Reviewer #1: Line 454: this is already documented by Wang et al. (2019)

Response: We agree that this follows what has been documented by Wang et al.

(2019) for the Bolling-Allerod and Early Holocene, although our interpretations are here based on a broader time scale. A reference to this has been added.

Reviewer #1: Line 455: How do you know? Any reference for this statement?

Response: Knowing that BR 1 is situated in the path of the Eastern Alboran Gyre and that and that mixing between surface and intermediate water masses is documented to occur down to 300 m (e.g. Heburn and La Violette, 1990), it is conceivable that corals profited from this oceanographic setup. This interpretation is noticeably based on benthic foraminiferal assemblage composition and the foraminferal stable isotope (O and C) values (see discussion). This is an interpretation of the dataset (thus discussion), and as all interpretations (especially when paleoenvironments) it is tentative and may be revised if new information and knowledge is gathered and fuels the debate. We do not "know", we are proposing an interpretation, which with the information at hand, seems the most plausible.

Reviewer #1: Line 463: there is no section 6.1.1

Response: corrected, section 5.1.1

Reviewer #1: Line 503: think about, if these mollusk layers may represent hiatuses . . .

Response: this interpretation was considered. However, considering that most of the shells were intact or near-to intact (Figure 6), and that they are quite fragile (and brittle), we do not think that these layers represent hiatuses. Furthermore, new radiocarbon datings (performed at ETH Zürich in collaboration with Dr. Irka Hajdas) from the first meter of core MD13-3462G confirm that these layers do not correspond to hiatuses. These radiocarbon datings are part of a manuscript that is currently being prepared.

Reviewer #1: Line 511: there is no section 6.4

Response: correction made, section 5.2.3

Reviewer #1: Line 527: how do you know about the quality of the organic matter?

Response: The benthic foraminiferal assemblage is used as a proxy for organic matter quality. This is developed prior to the sentence Line 527 (see lines 515 to 528). Noticeably, Cibicides spp., D. coronata and C. laevigata share a preference for high quality fresh marine organic matter (e.g. De Rijk et al., 2000; Milker et al., 2009, Stalder et al., 2018).

Reviewer #1: Line 563: Stronger contribution of nutrient-rich and well ventilated West. Med Deep Water to the coral sites only can have supported bryozoan proliferation with respect to oxygen. Nutrients provided by the WMDW would be "real" nutrients such as nitrate, phosphate etc. which would be of no use for any organisms in these aphotic depths. Be more precise in using the terms nutrients and food!

Response: correction made, "food" instead of "nutrients"

Reviewer #1: Line568: cannot see "particularly unstable" isotope values during the last glacial in Fig. 4!

Response: Indeed, the sentence is not correct it has been taken out.

Reviewer #1: Line 582ff: The link between high d18O and high Ti/Al and Si/Al ratios during the last glacial is not at all obvious, thus it cannot "confirm" (line 587) anything! Actually, between _100-200 cm you have high Si/Al ratios aligned with either high or low d18O values . . .

Response: We agree that this statement is not supported by the data. This whole section has been deleted (Lines 581 to 602).

Reviewer #1: Line 591-598: This Heinrich event discussion has no real relevance for this story . . .

Response: Section has been deleted (see previous comment)

Reviewer #1: Line 600: where is the logic link?

Response: Section has been deleted (see previous comment)

[Figure]

[Figure]

**Fig. 1.**

[Figure]

Origin of Organic matter

I - Lacustrine
IIa - Marine
IIb - Mixed marine-terrestrial
III - Terrestrial deltaic
IV - Terrestrial, high rank

**Fig. 2.**

---

## Referee Comment (RC2) · Naoufel Haddam (Referee) · 27 Aug 2020

General comments: Fentimen et al. provide interesting and new data on cold-water corals for the Alboran Sea covering the last 300 ky. They present a multiproxy comparison, using lithological properties and benthic foraminifera faunal assemblages to assess the environmental changes in the area. These new data are valuable, well presented in a well written manuscript, and I strongly advocate for their publication. However, the interpretation of the X-ray fluorescence (XRF) is a bit problematic for the production of decisive environmental interpretations, considering that the principal aim

of this study is to define the climatic processes susceptible of affecting the coral mound formation. I suggest the addition of some figures and more importantly, to either tone-down some of these interpretations, or if possible, to add (or compare to) more records that support the environmental reconstructions presented in this article.

Specific comments: In the introduction it might widen the scope of the manuscript to add that the selected coral mound is (strategically) located at the interface between different water masses as showed in fig. 2.2. It is later an important aspect of the discussion. The Oceanography section could be improved by describing water masses from shallower to deeper, if possible. More Calls to the figures in this section would be helpful. It would be helpful to add in fig.2 the water masses flowing directions (crosses and dots, in and out of the paper for example). Could you please elaborate whether Alboran Sea gyres strengths and structures display seasonal changes, as you mention that they are non-permanent? Concerning the benthic foraminifera faunal assemblages in section 3.5, I was wondering whether you checked the 63 -150 $\mu$m fraction? By doing so, you could (qualitatively at least) assess if there is a bias on small species (eg. the opportunistic specie Epistominella exigua)? Also, you did not mention on which literature you based your foraminifera species identifications? This should be added in the material and methods. I am aware of the difficulties due to the inconstant depositional processes in this area (which is also a problem in this study but I won't insist on it as you dealt with it fairly in the manuscript), but did you estimate the accumulation rates of benthic foraminifera (BFAR), and compared it to the foraminiferal density and TOC? It would be very helpful to add pictures of the benthic foraminifera cited in the text, especially those selected for any geochemical measurements. Providing pictures should be generalized in the litterature as confusions persist. I don't have access to the Annex, but you could add there a reference list to the original description of each foraminifera specie, at least for those cited in the main text. It is hard to get decisive and conclusive interpretation from the TOC, according to the curve presented in fig. 4. It would be interesting to compare this curve with the BFAR. In any case, the interpretation should be toned down. If available, other proxies of primary productivity changes

Interactive
comment

would be appreciable. If the sieved samples are still available, the fastest (but not the best) way to try check this would be to count the planktonic foraminifera and calculate their accumulation rates? Section 4.3 You only show the section 1 in the figure, will the other sections be added in supplementary? This would be interesting to illustrate your description line 320. I agree that Zr/Al and Rb/Al differ from Ti/Al and Si/Al but mostly in the fact that the first two are harder to interpret than the later. Unfortunately, I am not sure that any conclusive strong interpretation can be extracted from these curves. In the discussion it would be very helpful to have a figure showing TOC + benthic forams assemblages (at least the species that have a "significant ecological meaning") + $\delta$13C, especially to illustrate the discussion lines 555 - 560. Line 412-413 You only described B. spathulata as opportunistic (line 394), but you used the plural form "taxa". Do you consider all the fresh organic matter-feeding species as opportunistic? Also, it has been suggested that relatively small shelled species that rely on fresh organic matter show a faster (/stronger?) response to seasonality changes (Fontanier et al., 2006). Are there any signs of E. exigua in the samples? Since we are in the seasonality topic, are there any past and present evidences of seasonal changes affecting the surface stratification in the area and also the primary productivity? This would be the right place in the article to discuss this topic, and eventually support the benthic foraminifera findings (especially for what is observed at MIS 5 and 7). This addition would be a solution to balance the ""weakness"" of the XRF trends. You could also add a figure showing XRF + benthic forams for the runoff hypothesis. Is there any occurrence of deep infaunal and/or dysoxic species such as Globobulimina spp, Chilostomella, etc ? Section 5.1.2 It is very difficult to see systematic interglacial SS increases supporting seafloor turbulence. Also, the abundances of T. angulosa which is often associated with strong bottom water energy do not support this here. The steady fluvial increase during MIS 5 suggested by SS figure 8 might be plausible plausible, but it is still hard to observe a systematic strong glacial/interglacial signal. The second half of section 5.1.1 is well argumented. But I still have questions about the fate of these runoffs. How can we be sure that this "material" is not displaced laterally by the currents, away from

the studied area? Line 621 I am not sure that there is enough arguments in the discussion to dismiss water mass rearrangements yet. Line 646 What about seasonality changes? For the conclusion and also as a conclusive remark for all the manuscript; I suggest toning down the terrestrial inputs implications as the XRF data far from being clear enough to give decisive interpretations. I also suggest keeping the door open to other processes such as water mass reorganization or maybe the impact of primary productivity changes due to seasonality variations (affecting the gyres?).

Other suggestions: - Lines 24 – 36 I would suggest a reorganization of the second half of the abstract, as it feels that information are randomly presented, which might confuse some readers. - Line 134 isn't it northwest instead of northeast that the MAW enters the Alboran Sea? - Line 226 I think that it is important to mention in section 3.5 the error on the benthic foraminifera relative percentages. With ∼300 specimens counted, variations of less than ∼5 % are not very trustworthy. For more statistics of the sorts you can check and cite Patterson and Fishbein, 1989 and Fatela and Taborda, 2002. - Lines 300 – 304 are a bit too interpretative and should be placed and developed in the discussion. - Line 309 The sentence "This trend is mirrored in GS (Fig. 4)" is not useful as you are describing both SS and GS starting at line 306. - I think that you could place figure 5 in supplementary information, and put the sentences lines 310-313; "The percentage of sortable silt (SS%) increases with…and SS% is indicative of a sorting process induced by bottom currents (Fig. 5)", in the discussion, to support your (toned-down) interpretations. - Line 323, is it possible to indicate quantitavely the dominance of B. dichotoma ? - Could sentences lines 327-329 be simplified by just saying that bryozan and coral content is generally inversely correlated? - Lines 346 onward, it would be helpful to display the mean percentages of each species within the Bulimina grouping. Just out of curiosity, is the offset between the 3 species relatively stable down-core? I would also displace the diversity sentences (lines 343 – 345) to the end of the 4.4 section. - Line 359 I think you meant T. angulosa which is the one showing a ∼30 % abundance during MIS 6. I can't see 30 % for D. coronata during MIS 4 in figure 7. I advise to recheck the description of this figure in general. - Section

4.5 The Holocene is not mentioned, yet it is among the periods showing the most changes. - Line 398 I would replace "support" by "suggest". As mentioned before it would be better to tone down the interpretation. It is also a good spot in the text to put the origin of the TOC. - Line 428 could you please precise where these inputs occur?

---

## Short Comment (SC1) · 5 Sep 2020

I am posting this comment even if it may not be considered by the Editor as I am from the same Institute where the first authors was affiliated until February 2020 and other authors are presently affiliated (and co-authors of some other publications on the topic together with me).

However I cannot leave without commenting on such a manuscript and leave it to be published at least for the foraminiferal part that is claimed to be robust and presents

major weakness, as well as the paloeceanographic part.

In general, the interpretation is forced, giving CWC foraminifera a "clear, fixed and not questionable" significance, which may not be the case, especially for these ecosystems that are not completely well understood. These organisms can easily adapt and change their ecological preferences according to geographical location, oceanographic parameters, e.g., water depth, substratum, salinity, temperature, etc. . . . . . .(e.g., the same species can live in relatively shallower or deeper water according to the type of substratum, the same applies for all other parameters, e.g., salinity, nutrient and oxygen availability). In the manuscript all the discussion is based on given and fixed foraminifera ecological preference taken from the literature and in different geographical setting, instead of starting from establish proxies (e.g. TOC) and then interpret foraminiferal data.

Every situation must be evaluated case by case and anyhow a complete dataset including fractions smaller than 125 $\mu$m should be presented.Explainig everything with displacement is not a real reason. The same applies to the counting of the plankton, is more a problem of time consuming than scientific. To demonstrate that it is a scientific reason, data sghould be presented first and tehn excluded. The >125 $\mu$m can be useful when making taxonomic work e.g., taxonomic atlases and guides with plates (e.g., Milker and Schmiedl, 2012) but not for ecological purposes, in this case the 40 $\mu$m frection counts should be presented and eventually afterward not included in the discussion.

It is not clear how he density of benthic foraminifera has been calculated. The method used should be better explained and should be specified the reason for the choice. The method used in the manuscript does not correspond to any of the generally used in micropaleontology.

Line 226-227: "The benthic foraminiferal density was calculated by dividing the total number of foraminifera of a given sample by the sample fraction's weight".

RC: However, only 300 specimens per split were counted (line 223-224). How the splits represent "the fraction's weight"?

Usually density is calculated: - Number of foraminifera x gram of sediment (when sediments are generally homogeneous, not containing macrofaunal that can overestimate the weigh). It is calculated using number of foraminifera per single split, number of splits and weight of dry bulk sediments, and not fractions deriving from washing. (E.g., Moura et al. 2017 among many others)

- Number of foraminifera per volume (the most used for living assemblages and suggested in standard protocols). This method can be used also for fossil assemblages. In the article by Schönfeld et al., (2012) it is additionally and clearly stated that the 63 $\mu$m size fraction should be investigated when the environment is expected be more euthrophic. Or to show variations in organic matter content.

- Species percentages over the total specimens counted (in use for fossil foraminifera, especially planktonics). The first to use this metod were Haq et al. (1977) and successively Premoli Silva and Boersma (1989). Followed by many others.

I would like also to comment on Figure 10, which looks very fancy but presents a few problems. First of all it is upside down (even if the cardinal points are marked), we conventionally (and geographically) see the African margin at South and European Margin at North. Not the vice versa. This confuses the reader. As commented above during glacials the thermocline and pycnocline should be very shallow favoring water and nutrient mixing. In Figure 10 glacials are on the contrary described as stratified, the explanation for this is based only on comparison with modern times, it is generally confused and/or based on assumptions and circular reasoning. No clear evidence is presented.

On the contrary interglacial are represented as are the typical models for high latitudes/glacial times e.g., with strong mixing of water mass and nutrients. I First of all in the Mediterranean this cannot be possible, even in the past, also considering the

temperate latitude and seasonality. Additionally, if during interglacials fluvial input increased, then the fresh water plume arriving into the sea must have produced a clear separation of water masses (fluids with different densities) and not mixing. The closest large river is only at 50 km (Mouloya) , if the fluvial input was so massive to trigger coral growth, then also the fresh water plume must have been significant enough to produce stratification not mixing. Other alternative processes must be discussed?

If responsible for stratification in glacials are the stronger ShW then it must be demonstrated that they are indeed stronger (what ever "stronger" means: denser? colder?) and remarkably colder than at the surface to justify such a stratification acting a physical barrier between the sea floor and the surface. And this is not possible with the present data. At least an intermediate water species should have been analyzed for oxygen isotopes and not only at the BRI site but also in the Atlantic waters, e.g., Cadiz to have the ShW signature, as these are the waters that are supposed to influence the Alboran Sea (e.g, as in the title). Only Atlantic or Mediterranean waters are marked in the figures. If there were rivers they have to be documented as they are not only today but also how they were in the past 400.000 years, according to geological information.

Last but not least and for respect to the funding agency the first author Robin Fentimen should also acknowledge the Swiss National Science Foundation Project Ref. 200020_153125 "Faunal assemblages from active, declining and buried cold-water coral ecosystems" that payed his salary for 3 years over the 4 years of his PhD, and that has co-funded with the amount of 54.000 Euro the cruise Eurofleets GATEWAY, MD194 during which the cores investigated in this research were retrieved.

---

## Author Comment (AC2) · 5 Sep 2020

Reply to Referee #2

Manuscript title: The influence of Atlantic climate variability on the long-term development of Mediterranean cold-water coral mounds (Alboran Sea, Melilla Mound Field)

submitted to Climate of the Past

response by: Robin Fentimen et al.

[Figure]

In the following document, the responses to the comments made by Referee #2 are addressed one by one.

General comment

Comment Referee #2: the interpretation of the X-ray fluorescence (XRF) is a bit problematic for the production of decisive environmental interpretations, considering that the principal aim of this study is to define the climatic processes susceptible of affecting the coral mound formation. I suggest the addition of some figures and more importantly, to either tonedown some of these interpretations, or if possible, to add (or compare to) more records that support the environmental reconstructions presented in this article.

Response: We agree with Referee #2 that interpretations linked to the XRF records need to toned down. This is also in agreement with the points raised by Reviewer #1 (see comment and reply). Following these comments the section from Lines 581 to 602 has been deleted in the new version of the manuscript since the XRF data did not sufficiently support the interpretations made. Furthermore, we agree that down toning interpretations linked to the XRF in other parts of the manuscript is needed.

Regarding additional figures, the manuscript is already figure-rich. In addition, the comments and suggestions of Reviewer #1 (and also Reviewer #2, e.g. foraminiferal plates) will require the addition of extra three figures: 1. adding the CT visualization of fragments bigger than 2 cm; 2. adding the Van Krevelen diagram (see reply to Referee #1; 3. adding a plate illustrating the most important foraminifera species. As such, we consider that the manuscript will hopefully be adequately illustrated (see below for specific comments).

Specific comments

Reviewer comment: In the introduction it might widen the scope of the manuscript to add that the selected coral mound is (strategically) located at the interface between

different water masses as showed in fig. 2.2. It is later an important aspect of the discussion.

Response: Comment integrated in the revised version of the manuscript.

Reviewer comment: The Oceanography section could be improved by describing water masses from shallower to deeper, if possible. More Calls to the figures in this section would be helpful. It would be helpful to add in fig.2 the water masses flowing directions (crosses and dots, in and out of the paper for example). Could you please elaborate whether Alboran Sea gyres strengths and structures display seasonal changes, as you mention that they are non-permanent?

Response: Modifications made in the revised version of the manuscript. Details concerning the seasonal changes to the Alboran Sea gyres have been added.

Reviewer comment: Concerning the benthic foraminifera faunal assemblages in section 3.5, I was wondering whether you checked the 63-150 $\mu$m fraction? By doing so, you could (qualitatively at least) assess if there is a bias on small species (e.g. the opportunistic specie Epistominella exigua)?

Response: The fraction 63-125 $\mu$m (we used a 125 $\mu$m mesh) was not investigated in this study. It was intended to exclude the smaller forms which are more likely to be displaced by bottom currents (e.g. Lutze and Colbourn, 1984) which govern cold-water coral environments. Moreover, the inclusion of the finer fraction would make the data less comparable to other important benthic foraminiferal studies in the area (e.g. Schönfeld, 2002; Milker & Schmiedl, 2012; Stalder et al., 2015; 2018; Fentimen et al., 2020). However, we do agree that this approach has its drawbacks, noticeably the underestimation of smaller opportunistic species, and we strongly advocate for the investigation of the finer fraction in areas with weaker bottom currents and for the living assemblages. The authors have considered this methodological point in a study of the Moira Mounds - NE Atlantic CWC mounds (Fentimen et al., 2020 Marine Micropal). Taking into account the pros and cons of integrating the finer fraction (63-125 $\mu$m), we

decided to only focus on the larger fraction for this precise setting (this approach was also used in a high energy setting by Schönfeld (1997).

Reviewer comment: Also, you did not mention on which literature you based your foraminifera species identifications? This should be added in the material and methods.

Response: The identifications were based on a selection of benthic foraminiferal atlases (thus cross-referencing), essentially: Jones (1994), Murray (2003), Margreth (2010) and Milker and Schmiedl (2012). This has been added to the Material and Methods. A full list of the literature used for identifying foraminifera species (+ the original description of the given species) has been added as Annex 2 (see attached file).

Reviewer comment: I am aware of the difficulties due to the inconstant depositional processes in this area (which is also a problem in this study but I won't insist on it as you dealt with it fairly in the manuscript), but did you estimate the accumulation rates of benthic foraminifera (BFAR), and compared it to the foraminiferal density and TOC?

Response: We did estimate the BFAR (see attached Annex 1 and Supp. Figure 1) but decided to not include it in the manuscript. We decided to avoid using the BFAR since in such environments, which show intermittent sedimentation and erosive events, we believe it to be an untrustworthy proxy (see response to reviewer 1).

Reviewer comment: It would be very helpful to add pictures of the benthic foraminifera cited in the text, especially those selected for any geochemical measurements. Providing pictures should be generalized in the literature as confusions persist.

Response: We agree with the Reviewer that adding a plate with the most abundant foraminifera species would avoid potential confusions and allow comparison to other studies (and identifications, since these may indeed vary slightly from one person to another). Our identifications are in agreement with Jones (1994), Murray (2003), Margreth (2010) and Milker and Schmiedl (2012) (the literature used; see response to the

comment below and the newly added Annex 2), that clearly illustrate these species. For this reason, and since the manuscript is already figure-rich, we had first decided not to add a plate. However we agree that adding a plate is important to document our identifications so this will be done in the revised version.

Reviewer comment: I don't have access to the Annex, but you could add there a reference list to the original description of each foraminifera specie, at least for those cited in the main text.

Response: This has been added in Annex 2 (in the same table as the list of literature used for identifications). Annex 1 (foram counts) has also been attached to the reply as it was previously missing, we apologize for the inconvenience.

Reviewer comment: It is hard to get decisive and conclusive interpretation from the TOC, according to the curve presented in fig. 4. It would be interesting to compare this curve with the BFAR. In any case, the interpretation should be toned down.

Response: We agree with the Reviewer that these interpretations need to be toned down (also in agreement with Reviewer 1's comments). A curve comparing TOC to the BFAR is attached "Fig. BFAR _ TOC"). This can be added as Supp. Material to the manuscript if wished. However, BFAR in cold-water coral environments is biased by bottom current dynamics and may rather reflect hiatuses. Also, as mentioned in the reply to Reviewer 1, a number of micropaleontological studies have pointed out the reasons why the BFAR is potentially biased (see the review Jorissen et al., 2007). Noticeably, Naidu and Malmgren (1995) showed that in low oxygen environments, BFAR does not reflect surface-water productivity. Since we suspect that the seafloor at BR1 was at times depleted in oxygen, we further avoided to use the BFAR as a productivity proxy. Moreover, taphonomic processes, which directly impact BFAR, are not well constrained (see for example Murray, 2006; Stefanoudis et al., 2017; Capotondi et al., 2020; Fentimen et al., 2020).

Reviewer comment: If available, other proxies of primary productivity changes would

be appreciable. If the sieved samples are still available, the fastest (but not the best) way to try check this would be to count the planktonic foraminifera and calculate their accumulation rates?

Response: This approach would indeed be the fastest but as mentioned, not the best. Indeed, the accumulation rate of planktonic foraminifera is considerably biased in such settings by the strong currents affecting the area (see also response to comment above). We believe that the accumulation rate of planktonic foraminifera would probably reflect sorting by bottom currents rather than productivity (see for example, Fentimen et al., 2020, Marine Micropal) and as such should be avoided. For this study we had started assessing planktonic foraminiferal assemblages but decided against further investigations for the following reasons: (1) planktonic foraminifera are more likely to be allochtonous than benthic foraminifera, especially considering the setting, and (2) planktonic foraminifera from the study site were concentrated essentially within the smaller sized material (63-150 $\mu$m), thus further increasing the probability of a high contribution of allochtonous foraminifera.

Reviewer comment: Section 4.3: You only show the section 1 in the figure, will the other sections be added in supplementary? This would be interesting to illustrate your description line 320.

Response: Other sections will be added to the manuscript (also in agreement with the comment by Reviewer 1) together with the CT visualization of fragments bigger than 2 cm. We are currently working on producing this.

Reviewer comment: I agree that Zr/Al and Rb/Al differ from Ti/Al and Si/Al but mostly in the fact that the first two are harder to interpret than the later. Unfortunately, I am not sure that any conclusive strong interpretation can be extracted from these curves.

Response: The interpretations linked to the XRF will be toned down, as previously suggested. We aim to use this dataset rather as supporting information for the macrofaunal and microfaunal assemblages. We will make this clearer in the revised version

of the manuscript.

Reviewer comment: In the discussion it would be very helpful to have a figure showing TOC + benthic forams assemblages (at least the species that have a "significant ecological meaning") + $\delta$13C, especially to illustrate the discussion lines 555 - 560.

Response: We agree with this, however this was not done to avoid overloading Figure 7 which already contains a lot of information. Thus we would prefer not to combine or present a new figure, especially since the manuscript already contains quite some figures (+ the ones that will be added following both reviewers' comments: see reply general comment and reply to reviewer 1).

Reviewer comment: Line 412-413: You only described B. spathulata as opportunistic (line 394), but you used the plural form "taxa". Do you consider all the fresh organic matter-feeding species as opportunistic?

Response: Indeed this is not clearly stated in the sentence, precision is added in the revised version. Indeed, we consider B. spathulata as opportunistic but also Bulimina spp., following the observations made by Eichler et al. (2014) or Lutze and Coulbourn (1984).

Reviewer comment: Also, it has been suggested that relatively small shelled species that rely on fresh organic matter show a faster (/stronger?) response to seasonality changes (Fontanier et al., 2006). Are there any signs of E. exigua in the samples?

Response: There was no E. exigua in the samples. There were very scarce occurrences of Alabaminella weddellensis (a species sharing the same ecology, i.e. responding rapidly to periods of increased phytodetritus input). See also attached Annex 1. We agree that E. exigua (and other opportunistic phytodetritus-feeding species) are generally small and are essentially found in the smaller fraction (63-125 $\mu$m), so variations in this species abundance are possibly missed. Other studies at BR 1 did not either report the presence of Epistominella exigua and E. vitrea (Stalder et al., 2015;

2018). This question and response go together with the discussion to integrate or not the finer fraction (63 - 125 $\mu$m) - see above, previous comment.

Reviewer comment: Since we are in the seasonality topic, are there any past and present evidences of seasonal changes affecting the surface stratification in the area and also the primary productivity? This would be the right place in the article to discuss this topic, and eventually support the benthic foraminifera findings (especially for what is observed at MIS 5 and 7). This addition would be a solution to balance the ""weakness"" of the XRF trends. You could also add a figure showing XRF + benthic forams for the runoff hypothesis.

Response: Primary productivity in the Alboran Sea is controlled by a number of variables: the formation of Western Mediterranean Deep Water in the Gulf of Lions which would itself be influenced by varying atmospheric conditions (for studies on the topic, see for example Ausin et al. (2014; 2015) and references therein). Moreover, the influence of entering Atlantic Water (which enters as a jet at the Strait of Gibraltar) on primary productivity is also important and is subject to seasonal changes (the strength of the jet at the Strait of Gibraltar will have an effect on the strength of both Western and Eastern Alboran Gyres). For literature, see for example: Heburn and La Violette (1990), Oguz et al. (2014). So seasonal changes do indeed affect surface stratification at the study site. However, considering the location of BR 1, the benthic foraminiferal assemblages, TOC (see Van Krevelen diagram attached) and at a lesser extent XRF results, we believe that BR 1 is essentially impacted by variations in terrestrial input, and secondly by water mass rearrangements (see discussion section 5.1.2). Again we would prefer to avoid adding an extra figure for the reasons already mentioned above (i.e. high number of figures already presented in the manuscript).

Reviewer comment: Is there any occurrence of deep infaunal and/or dysoxic species such as Globobulimina spp, Chilostomella, etc ?

Response: Deep infaunal species can be considered rare. The most abundant deep

infaunal species is Chilostomella oolina (max. abundance 5%, av. abundance ca. 1 to 2 %). Globobulimids are even less abundant (max. abundance 2 %, av. abundance approximately 0.5 %). See Annex 1 (attached). In order to estimate oxygen content variation, we used the formula proposed by Schmiedl et al. (2003): (OH/ (OH + LO) + Div) * 0.5, with OH = relative abundance of high oxygen indicators (e.g. Cibicides pachyderma, Gyroidina orbicularis, Hanzawaia boueana, Lenticulina spp., Pyrgo spp., Quinqueloculina spp., and Sigmoilopsis schlumbergeri), LO = relative abundance of low oxygen indicators (Bolivina spp., Bulimina spp., Cassidulina carinata, Chilostomella oolina, Globobulimina spp., Melonis barleeanus, Nonionella turgida, Praeglobobulimina ovata, Trifarina spp., and Uvigerina spp.) and Div = normalized benthic foraminifera diversity. We decided however not to include this in the manuscript since it essentially reflects the abundance of buliminids, and was hence redundant.

Reviewer comment: Section 5.1.2 It is very difficult to see systematic interglacial SS increases supporting seafloor turbulence. Also, the abundances of T. angulosa which is often associated with strong bottom water energy do not support this here. The steady fluvial increase during MIS 5 suggested by SS figure 8 might be plausible, but it is still hard to observe a systematic strong glacial/interglacial signal.

Response: We agree with the Reviewer that this statement is misleading and does not match with the the foraminiferal assemblages. The sentence has thus been reworked in the revised version of the manuscript. The sentence now reads: "This would promote the formation of internal waves and would have favoured coral proliferation by increasing lateral food availability (Fig. 10)". This is also in better agreement with the title of the section.

Reviewer comment: The second half of section 5.1.1 is well argumented. But I still have questions about the fate of these runoffs. How can we be sure that this "material" is not displaced laterally by the currents, away from the studied area? Line 621: I am not sure that there is enough arguments in the discussion to dismiss water mass rearrangements yet.

Response: The results of the RockEval analyses indicate that the origin of the organic matter preserved in the sediment at Brittlestar Ridge 1 is of terrestrial origin. The newly added figure (Oxygen Index vs. Hydrogen Index diagram, see attached) demonstrates this. If the material resulting from terrestrial run offs were to be displaced laterally by currents, one would expect the signal of the organic matter preserved at BR 1 to be rather marine in origin. This is not the case. The statement Line 621 is not intended to dismiss or exclude water mass rearrangements, it rather suggests that they are of secondary importance at BR 1 when compared to fluvial input. This is especially true in comparison to other CWC environments, for example the extensively studied CWC mounds in the Northeast Atlantic (Irish margin), where water mass rearrangements are believed to drive almost exclusively cold-water coral growth dynamics.

Reviewer comment: Line 646: What about seasonality changes?

Response: No seasonality changes were documented in this study, so we cannot make any conclusions about these (nor can we confirm anything).

Reviewer comment: For the conclusion and also as a conclusive remark for all the manuscript; I suggest toning down the terrestrial inputs implications as the XRF data far from being clear enough to give decisive interpretations. I also suggest keeping the door open to other processes such as water mass reorganization or maybe the impact of primary productivity changes due to seasonality variations (affecting the gyres?).

Response: We agree that the XRF data needs to be toned down (as mentioned previously and in the reply to Reviewer 1). However the conclusion that fluvial input plays a decisive role in coral development is especially based is also and especially supported by benthic foraminiferal assemblages (see section 5.1.1). The influence of water mass rearrangements is also highlighted in the conclusion (e.g. "Increased fluvial organic matter inputs are driven by the increased impact of warm and moist Atlantic air masses with intensified Western and Eastern Alboran Gyres that lead to more important turnover between surface and intermediate water masses. This phenomenon

is promoted by enhanced Modified Atlantic Water inflow at the Strait of Gibraltar"). We do not exclude in this final section the effect of water mass rearrangements and do not develop the impact of seasonality variations since we have little indications about this in this study. Hence we would prefer to keep the conclusion brief and as such.

Other suggestions

Reviewer comment: Lines 24 – 36 I would suggest a reorganization of the second half of the abstract, as it feels that information are randomly presented, which might confuse some readers.

Response: This has been reworked in the revised version of the manuscript.

Reviewer comment: Line 134 isn't it northwest instead of northeast that the MAW enters the Alboran Sea?

Response: Indeed, this has been corrected.

Reviewer comment: Line 226 I think that it is important to mention in section 3.5 the error on the benthic foraminifera relative percentages. WithâĹij300 specimens counted, variations of less thanâĹij5 % are not very trustworthy. For more statistics of the sorts you can check and cite Patterson and Fishbein, 1989 and Fatela and Taborda, 2002.

Response: This has been mentioned in the latest version of the manuscript.

Reviewer comment: Lines 300 – 304 are a bit too interpretative and should be placed and developed in the discussion.

Response: We agree with the Reviewer. This has been placed and further developed in the discussion (in addition to the OI vs. HI diagram which illustrates this statement, see attached figure).

Reviewer comment: Line 309 The sentence "This trend is mirrored in GS (Fig. 4)" is not useful as you are describing both SS and GS starting at line 306. - I think that you could place figure 5 in supplementary information, and put the sentences lines 310-

313; "The percentage of sortable silt (SS%) increases with...and SS% is indicative of a sorting process induced by bottom currents (Fig. 5)", in the discussion, to support your (toned-down) interpretations.

Response: We agree with the Reviewer on both theses points. Figure 5 has been moved to the Annexes (making room for other figures, as stated in the general comment). Reviewer comment: Line 323, is it possible to indicate quantitavely the dominance of B. dichotoma ?

Response: Yes this is possible and has been added in the revised version of the manuscript. B. dichotoma makes up for over 95% of all counted bryozoans.

Reviewer comment: Could sentences lines 327-329 be simplified by just saying that bryozoan and coral content is generally inversely correlated?

Response: This could be done but we believe that it would possibly be a case of over simplification. Coral and bryozoan content are indeed anti-correlated during MIS 5 and MIS 2 but the distribution pattern of both organisms does not always follow such a pattern (see for example MIS 6).

Reviewer comment: Lines 346onward, it would be helpful to display the mean percentages of each species within the Bulimina grouping. Just out of curiosity, is the offset between the 3 species relatively stable down-core? I would also displace the diversity sentences (lines 343 – 345) to the end of the 4.4 section.

Response: The mean percentages of Buliminid species can be found in Annex 1. The most abundant species is B. marginata, followed by B. striata and then B. aculeatea. B. aculeata shows the strongest offset with the two other species (noticeably during the last glacial). Moving lines 343 - 345 to end of the section has been done in the revised version of the manuscript.

Reviewer comment: Line 359 I think you meant T. angulosa which is the one showing aâĹij30 % abundance during MIS 6. I can't see 30 % for D. coronata during MIS 4 in

figure 7. I advise to recheck the description of this figure in general.

Response: This sentence needs indeed to be corrected. D. coronata reaches ca. 20 % during MIS 4. Corrections have been made.

Reviewer comment: Section 4.5 The Holocene is not mentioned, yet it is among the periods showing the most changes.

Response: We rather chose to highlight the changes at the transition between the last glacial and the Holocene: "The passage from MIS 2 to MIS 1 is marked by a sharp decrease in planktonic and benthic $\delta$13C (from -1.2 ‰ to -2.2 ‰ and from 1.8 ‰ to 1.0 ‰ respectively)". We chose not to insist too much on the Holocene in this study as this core is not the best suited to study this time interval at BR 1 (cores studied by Fink et al., 2013; Stalder et al., 2015; 2018 are better examples).

Reviewer comment: Line 398 I would replace "support" by "suggest". As mentioned before it would be better to tone down the interpretation. It is also a good spot in the text to put the origin of the TOC.

Response: We agree with the Reviewer. This needs to be toned down and the origin of the organic matter (OI vs. HI diagram) integrated at this stage of the discussion.

Reviewer comment: Line 428 could you please precise where these inputs occur?

Response: Precision added.
* * *
Species list and quantitative data of benthic foraminifera from core MD13-3462G.

| Depth (cm) | 2 | 12 | 22 | 32 | 42 | 52 | 62 | 72 | 82 | 92 | 102 | 112 | 122 | 132 | 142 | 152 |
|---|---|---|---|---|---|---|---|---|---|---|---|---|---|---|---|---|
| Sum counts | 582 | 438 | 413 | 620 | 616 | 639 | 434 | 394 | 341 | 374 | 315 | 282 | 301 | 332 | 314 | 341 |
| Split | 16 | 4 | 4 | 2 | 8 | 16 | 4 | 16 | 32 | 16 | 64 | 128 | 64 | 128 | 128 | 64 |
| Total (Sum counts × Split) | 9312 | 1752 | 1652 | 1240 | 4928 | 10224 | 1736 | 6304 | 10912 | 5984 | 20160 | 36096 | 19264 | 42496 | 40192 | 21824 |
| Fraction weight (g) | 0.8 | 1.07 | 0.86 | 0.32 | 0.76 | 2.39 | 0.62 | 1.69 | 1.46 | 1.01 | 5.01 | 6.47 | 3.04 | 10.85 | 10.38 | 10.32 |
| Foraminifera/g | 11640 | 1637 | 1921 | 3875 | 6484 | 4278 | 2800 | 3730 | 7474 | 5925 | 4024 | 5579 | 6337 | 3917 | 3872 | 2115 |
| *Adelosina laevigata* | 0 | 0 | 0 | 0 | 0 | 0 | 0 | 0 | 0 | 0 | 0 | 0 | 0 | 0 | 0 | 0 |
| *Alabaminella weddellensis* | 0 | 0 | 0 | 0 | 0 | 0 | 0 | 0 | 0 | 0 | 0 | 0 | 0 | 0 | 0 | 0 |
| *Ammonia beccarii* | 0 | 0 | 0 | 0 | 0 | 0 | 0 | 0 | 0 | 0 | 0 | 1 | 0 | 0 | 0 | 0 |
| *Amphistegina lessonii* | 0 | 0 | 0 | 0 | 0 | 0 | 0 | 0 | 0 | 0 | 0 | 0 | 0 | 0 | 0 | 0 |
| *Amphycorina scalaris* | 6 | 14 | 8 | 0 | 1 | 0 | 0 | 0 | 0 | 0 | 0 | 1 | 0 | 0 | 0 | 0 |
| *Anomalinoides globulosus* | 0 | 0 | 0 | 0 | 0 | 0 | 1 | 0 | 0 | 0 | 0 | 1 | 0 | 1 | 1 | 1 |
| *Astrononion antarcticus* | 0 | 0 | 1 | 2 | 0 | 0 | 0 | 0 | 0 | 0 | 0 | 1 | 0 | 0 | 0 | 0 |
| *Astrononion gallowayi* | 0 | 0 | 0 | 3 | 0 | 0 | 2 | 0 | 0 | 0 | 0 | 0 | 0 | 0 | 0 | 0 |
| *Astrononion stelligerum* | 0 | 0 | 0 | 22 | 0 | 7 | 19 | 29 | 6 | 0 | 17 | 15 | 1 | 2 | 12 | 0 |
| *Bigenerina nodosaria* | 1 | 3 | 0 | 0 | 0 | 0 | 0 | 0 | 0 | 0 | 1 | 0 | 0 | 0 | 0 | 0 |
| *Biloculinella depressa* | 0 | 0 | 0 | 0 | 0 | 0 | 0 | 0 | 0 | 0 | 0 | 0 | 0 | 0 | 0 | 0 |
| *Biloculinella inflata* | 0 | 0 | 0 | 0 | 0 | 0 | 0 | 0 | 0 | 0 | 0 | 0 | 0 | 1 | 0 | 0 |
| *Biloculinella labiata* | 0 | 1 | 1 | 1 | 0 | 1 | 0 | 0 | 1 | 0 | 0 | 0 | 0 | 0 | 0 | 0 |
| *Bolivina alata* | 5 | 2 | 2 | 1 | 5 | 2 | 0 | 0 | 0 | 0 | 4 | 0 | 0 | 2 | 3 | 3 |
| *Bolivina difformis* | 2 | 2 | 1 | 0 | 0 | 0 | 0 | 0 | 0 | 0 | 0 | 0 | 0 | 0 | 0 | 0 |
| *Bolivina pseudoplicata* | 0 | 0 | 2 | 1 | 3 | 0 | 1 | 2 | 1 | 1 | 3 | 1 | 0 | 0 | 0 | 0 |
| *Bolivina spathulata* | 1 | 3 | 19 | 0 | 9 | 4 | 4 | 4 | 10 | 37 | 7 | 3 | 4 | 2 | 6 | 6 |
| *Bolivina spinescens* | 0 | 0 | 1 | 0 | 0 | 0 | 0 | 0 | 0 | 0 | 0 | 0 | 0 | 0 | 0 | 0 |
| *Bolivina striatula* | 2 | 2 | 3 | 6 | 9 | 11 | 12 | 13 | 3 | 0 | 8 | 2 | 13 | 12 | 5 | 4 |
| *Bolivina subspinescens* | 0 | 0 | 0 | 0 | 0 | 0 | 0 | 0 | 0 | 0 | 0 | 0 | 0 | 0 | 0 | 0 |
| *Bolivina variabilis* | 0 | 0 | 0 | 0 | 0 | 0 | 0 | 0 | 0 | 0 | 0 | 0 | 0 | 0 | 0 | 0 |
| *Bulimina aculeata* | 10 | 5 | 11 | 7 | 8 | 16 | 1 | 4 | 0 | 2 | 3 | 4 | 4 | 7 | 12 | 10 |
| *Bulimina marginata* | 23 | 44 | 31 | 3 | 5 | 4 | 2 | 2 | 8 | 3 | 2 | 0 | 2 | 1 | 1 |
| *Bulimina striata* | 7 | 20 | 27 | 3 | 1 | 0 | 0 | 0 | 0 | 0 | 0 | 0 | 0 | 0 | 0 | 0 |
| *Cancris auricula* | 0 | 0 | 0 | 0 | 0 | 0 | 0 | 0 | 0 | 0 | 0 | 0 | 0 | 0 | 0 | 0 |
| *Cassidulina carinata* | 0 | 0 | 0 | 0 | 0 | 0 | 0 | 0 | 0 | 0 | 1 | 0 | 0 | 0 | 0 | 0 |
| *Cassidulina crassa* | 6 | 18 | 10 | 2 | 22 | 8 | 2 | 2 | 9 | 2 | 3 | 4 | 4 | 8 | 3 | 3 |
| *Cassidulina laevigata* | 52 | 17 | 32 | 96 | 60 | 68 | 45 | 42 | 61 | 20 | 34 | 23 | 48 | 43 | 33 | 66 |
| *Cassidulina reniforme* | 0 | 3 | 0 | 2 | 4 | 0 | 0 | 4 | 4 | 1 | 0 | 0 | 0 | 0 | 0 | 0 |
| *Cassidulinoides bradyi* | 2 | 1 | 2 | 0 | 0 | 2 | 0 | 0 | 0 | 0 | 1 | 1 | 0 | 1 | 0 | 0 |
| *Chilostomella oolina* | 2 | 8 | 6 | 0 | 0 | 0 | 0 | 0 | 0 | 0 | 0 | 0 | 0 | 0 | 0 | 0 |
| *Cibicides aravaensis* | 3 | 17 | 10 | 1 | 10 | 6 | 1 | 2 | 6 | 2 | 0 | 1 | 0 | 2 | 2 | 0 |

**Fig. 1.**

CPD
Annex 1. List of all benthic foraminfera species identified in this thesis, together with the references used for identification (see below for reference list).

| Species | Original name | References used for identification |
|---|---|---|
| *Adelosina laevigata* d'Orbigny, 1826 | *Adelosina laevigata* d'Orbigny, 1826 | Milker and Schmiedl, 2012 (Fig. 12. 18-19) |
| *Alabaminella weddellensis*(Earland, 1936) | *Eponides weddellensis* Earland, 1936 | Erdem and Schönfeld, 2017 (Fig. 8. 24); Setoyama and Kaminski, 2015 (Fig. 5. 3) |
| *Ammonia beccarii* (Linnaeus, 1758) | *Nautilus beccarii* Linnaeus, 1758 | Milker and Schmiedl, 2012 (Fig. 27. 1-2) |
| *Amphistegina lessonii* d'Orbigny, 1826 | *Amphistegina lessonii* d'Orbigny, 1826 | Hottinger, 1993 (Pl. 184, Fig. 1-11) |
| *Amphicoryna scalaris* (Batsch, 1791) | *Nautilus scalaris* Batsch, 1791 | Murray, 2003 (Fig. 5. 1); Milker and Schmiedl, 2012 (Fig. 18. 22-25) |
| *Anomalinoides globulosus* (Chapman and Parr, 1937) | *Anomalina globulosa* Chapman and Parr, 1937 | Margreth, 2010 (Pl. 39, Fig. 1) |
| *Astrononion antarcticus* Parr, 1950 | *Astrononion antarcticus* Parr, 1950 | Margreth, 2010 (Pl. 37, Fig. 4) |
| *Astrononion gallowayi* Loeblich and Tappan, 1953 | *Astrononion gallowayi* Loeblich and Tappan, 1953 | Margreth, 2010 (Pl. 37, Fig. 3) |
| *Astrononion stelligerum* (d'Orbigny, 1839) | *Nonionina stelligera* d'Orbigny, 1839 | Cimerman and Langer, 1991 (Pl. 84, Fig. 13-15) |
| *Bigenerina nodosaria* d'Orbigny, 1826 | *Bigenerina nodosaria* d'Orbigny, 1826 | Margreth, 2010 (Pl. 5, Fig. 5); Milker and Schmiedl, 2012 (Fig. 10. 10-11) |
| *Biloculinella depressa* (d'Orbigny, 1826) | *Biloculina depressa* d'Orbigny, 1826 | Margreth, 2010 (Pl. 8, Fig. 3); Murray, 2003 (Fig. 4. 2-3) |
| *Biloculinella globulus* (Bornemann, 1855) | *Biloculina globulus* Bornemann, 1855 | Margreth, 2010 (Pl. 8, Fig. 2); Milker and Schmiedl, 2012 (Fig. 16. 19) |
| *Biloculinella inflata* (Wright, 1902) | *Biloculina inflata* Wright, 1902 | Milker and Schmiedl, 2012 (Fig. 16. 20) |
| *Biloculinella labiata* (Schlumberger, 1891) | *Biloculina labiata* Schlumberger, 1891 | Milker and Schmiedl, 2012 (Fig. 16. 21-22) |
| *Bolivina alata* (Seguenza, 1862) | *Vulvulina alata* Seguenza, 1862 | Margreth, 2010 (Pl. 24, Fig. 1) |
| *Bolivina difformis* (Williamson, 1858) | *Textularia variabilis var. difformis* Williamson, 1858 | Margreth, 2010 (Pl. 24, Fig. 6); Milker and Schmiedl, 2012 (Fig. 19. 28-29) |
| *Bolivina pseudoplicata* Heron-Allen an Earland, 1930 | *Bolivina pseudoplicata* Heron-Allen and Earland, 1930 | Milker and Schmiedl, 2012 (Fig- 19. 22-23); Murray, 2003 (Fig. 5. 17) |
| *Bolivina spathulata* (Williamson, 1858) | *Textularia variabilis var. spathulata* Williamson, 1858 | Milker and Schmiedl, 2012 (Fig. 20. 1-2) |
| *Bolivina spinescens* Cushman, 1911 | *Bolivina spinescens* Cushman, 1911 | Margreth, 2010 (Pl. 24, Fig. 7) |
| *Bolivina striatula* Cushman, 1922 | *Bolivina striatula* Cushman, 1922 | Margreth, 2010 (Pl. 24, Fig. 5); Milker and Schmiedl, 2012 (Fig. 20. 3) |
| *Bolivina subspinescens* Cushman, 1922 | *Bolivina subspinescens* Cushman, 1922 | Margreth, 2010 (Pl. 24, Fig. 8); Milker and Schmiedl, 2012 (Fig. 19. 24) |
| *Bolivina variabilis* (Williamson, 1858) | *Textularia variabilis* Williamson, 1858 | Milker and Schmiedl, 2012 (Fig. 19. 25-26) |
| *Bulimina aculeata* d'Orbigny, 1826 | *Bulimina aculeata* d'Orbigny, 1826 | Margreth, 2010 (Pl. 27, Fig. 8); Milker and Schmiedl, 2012 (Fig. 20. 19) |
| *Bulimina marginata* d'Orbigny, 1826 | *Bulimina marginata* d'Orbigny, 1826 | Milker and Schmiedl, 2012 (Fig. 20. 23); Murray, 2003 (Fig. 6. 4-5) |
| *Bulimina striata* d'Orbigny, 1826 | *Bulimina striata* d'Orbigny, 1826 | Frontalini et al., 2014 (Fig. 6. 5); Margreth, 2010 (Pl. 27, Fig. 10) |
| *Cancris auricula* (Fichtel and Moll, 1798) | *Nautilus auricula* Fichtel and Moll, 1798 | Milker and Schmiedl, 2012 (Fig. 21. 14-15); Murray, 2003 (Fig. 6. 6-7) |
| *Cassidulina carinata* (Silvestri, 1896) | *Cassidulina laevigata var. carinata* Silvestri, 1896 | Margreth, 2010 (Pl. 25, Fig. 5); Milker and Schmiedl, 2012 (Fig. 20. 5) |
| *Cassidulina crassa* d'Orbigny, 1839 | *Cassidulina crassa* d'Orbigny, 1839 | Jones, 1994 (Pl. 54, Fig. 4); Margreth, 2010 (Pl. 26, Fig. 3) |
| *Cassidulina laevigata* d'Orbigny, 1826 | *Cassidulina laevigata* d'Orbigny, 1826 | Margreth, 2010 (Pl. 25, Fig. 4); Murray, 2003 (Fig. 6. 8-10) |

**Fig. 2.**

[Figure]

**Fig. 3.**

[Figure]

**Fig. 4.**

---

## Author Comment (AC3) · 5 Sep 2020

The comment was uploaded in the form of a supplement:
https://cp.copernicus.org/preprints/cp-2020-82/cp-2020-82-AC3-supplement.pdf

---

## Author Comment (AC4) · 5 Sep 2020

**Annex 2**. List of all benthic foraminfera species identified in this thesis, together with the references used for identification (see below for reference list).

| Species | Original name | References used for identification |
|---|---|---|
| *Adelosina laevigata* d'Orbigny, 1826 | *Adelosina laevigata* d'Orbigny, 1826 | Milker and Schmiedl, 2012 (Fig. 12. 18-19) |
| *Alabaminella weddellensis*(Earland, 1936) | *Eponides weddellensis* Earland, 1936 | Erdem and Schönfeld, 2017 (Fig. 8. 24); Setoyama and Kaminski, 2015 (Fig. 5. 3) |
| *Ammonia beccarii* (Linnaeus, 1758) | *Nautilus beccarii* Linnaeus, 1758 | Milker and Schmiedl, 2012 (Fig. 27. 1-2) |
| *Amphistegina lessonii* d'Orbigny, 1826 | *Amphistegina lessonii* d'Orbigny, 1826 | Hottinger, 1993 (Pl. 184, Fig. 1-11) |
| *Amphicoryna scalaris* (Batsch, 1791) | *Nautilus scalaris* Batsch, 1791 | Murray, 2003 (Fig. 5. 1); Milker and Schmiedl, 2012 (Fig. 18. 22-25) |
| *Anomalinoides globulosus* (Chapman and Parr, 1937) | *Anomalina globulosa* Chapman and Parr, 1937 | Margreth, 2010 (Pl. 39, Fig. 1) |
| *Astrononion antarcticus* Parr, 1950 | *Astrononion antarcticus* Parr, 1950 | Margreth, 2010 (Pl. 37, Fig. 4) |
| *Astrononion gallowayi* Loeblich and Tappan, 1953 | *Astrononion gallowayi* Loeblich and Tappan, 1953 | Margreth, 2010 (Pl. 37, Fig. 3) |
| *Astrononion stelligerum* (d'Orbigny, 1839) | *Nonionina stelligera* d'Orbigny, 1839 | Cimerman and Langer, 1991 (Pl. 84, Fig. 13-15) |
| *Bigenerina nodosaria* d'Orbigny, 1826 | *Bigenerina nodosaria* d'Orbigny, 1826 | Margreth, 2010 (Pl. 5, Fig. 5); Milker and Schmiedl, 2012 (Fig. 10. 10-11) |
| *Biloculinella depressa* (d'Orbigny, 1826) | *Biloculina depressa* d'Orbigny, 1826 | Margreth, 2010 (Pl. 8, Fig. 3); Murray, 2003 (Fig. 4. 2-3) |
| *Biloculinella globulus* (Bornemann, 1855) | *Biloculina globulus* Bornemann, 1855 | Margreth, 2010 (Pl. 8, Fig. 2); Milker and Schmiedl, 2012 (Fig. 16. 19) |
| *Biloculinella inflata* (Wright, 1902) | *Biloculina inflata* Wright, 1902 | Milker and Schmiedl, 2012 (Fig. 16. 20) |
| *Biloculinella labiata* (Schlumberger, 1891) | *Biloculina labiata* Schlumberger, 1891 | Milker and Schmiedl, 2012 (Fig. 16. 21-22) |
| *Bolivina alata* (Seguenza, 1862) | *Vulvulina alata* Seguenza, 1862 | Margreth, 2010 (Pl. 24, Fig. 1) |
| *Bolivina difformis* (Williamson, 1858) | *Textularia variabilis var. difformis* Williamson, 1858 | Margreth, 2010 (Pl. 24, Fig. 6); Milker and Schmiedl, 2012 (Fig. 19. 28-29) |
| *Bolivina pseudoplicata* Heron-Allen an Earland, 1930 | *Bolivina pseudoplicata* Heron-Allen an Earland, 1930 | Milker and Schmiedl, 2012 (Fig- 19. 22-23); Murray, 2003 (Fig. 5. 17) |
| *Bolivina spathulata* (Williamson, 1858) | *Textularia variabilis var. spathulata* Williamson, 1858 | Milker and Schmiedl, 2012 (Fig. 20. 1-2) |
| *Bolivina spinescens* Cushman, 1911 | *Bolivina spinescens* Cushman, 1911 | Margreth, 2010 (Pl. 24, Fig. 7) |
| *Bolivina striatula* Cushman, 1922 | *Bolivina striatula* Cushman, 1922 | Margreth, 2010 (Pl. 24, Fig. 5); Milker and Schmiedl, 2012 (Fig. 20. 3) |
| *Bolivina subspinescens* Cushman, 1922 | *Bolivina subspinescens* Cushman, 1922 | Margreth, 2010 (Pl. 24, Fig. 8); Milker and Schmiedl, 2012 (Fig. 19. 24) |
| *Bolivina variabilis* (Williamson, 1858) | *Textularia variabilis* Williamson, 1858 | Milker and Schmiedl, 2012 (Fig. 19. 25-26) |
| *Bulimina aculeata* d'Orbigny, 1826 | *Bulimina aculeata* d'Orbigny, 1826 | Margreth, 2010 (Pl. 27, Fig. 8); Milker and Schmiedl, 2012 (Fig. 20. 19) |
| *Bulimina marginata* d'Orbigny, 1826 | *Bulimina marginata* d'Orbigny, 1826 | Milker and Schmiedl, 2012 (Fig. 20. 23); Murray, 2003 (Fig. 6. 4-5) |
| *Bulimina striata* d'Orbigny, 1826 | *Bulimina striata* d'Orbigny, 1826 | Frontalini et al., 2014 (Fig. 6. 5); Margreth, 2010 (Pl. 27, Fig. 10) |
| *Cancris auricula* (Fichtel and Moll, 1798) | *Nautilus auricula* Fichtel and Moll, 1798 | Milker and Schmiedl, 2012 (Fig. 21. 14-15); Murray, 2003 (Fig. 6. 6-7) |
| *Cassidulina carinata* (Silvestri, 1896) | *Cassidulina laevigata var. carinata* Silvestri, 1896 | Margreth, 2010 (Pl. 25, Fig. 5); Milker and Schmiedl, 2012 (Fig. 20. 5) |
| *Cassidulina crassa* d'Orbigny, 1839 | *Cassidulina crassa* d'Orbigny, 1839 | Jones, 1994 (Pl. 54, Fig. 4); Margreth, 2010 (Pl. 26, Fig. 3) |
| *Cassidulina laevigata* d'Orbigny, 1826 | *Cassidulina laevigata* d'Orbigny, 1826 | Margreth, 2010 (Pl. 25, Fig. 4); Murray, 2003 (Fig. 6. 8-10) |

**Annex 2**. continuation

| Species | Original name | References used for identification |
|---|---|---|
| *Cassidulina reniforme* (Nørvang, 1945) | *Cassidulina crassa var. reniforme* Nørvang, 1945 | Margreth, 2010 (Pl. 25, Fig. 3) |
| *Cassidulinoides bradyi* (Norman, 1881) | *Cassidulina bradyi* Norman, 1881 | Margreth, 2010 (Pl. 26, Fig. 6); Milker and Schmiedl, 2012 (Fig. 20. 9) |
| *Chilostomella oolina* Schwager, 1878 | *Chilostomella oolina* Schwager, 1878 | Jones, 1994 (Pl. 55, Fig. 12-14, 17-18); Margreth, 2010 (Pl. 38, Fig. 5) |
| *Cibicides aravaensis* Perelis and Reiss, 1975 | *Cibicides aravaensis* Perelis and Reiss, 1975 | Margreth, 2010 (Pl. 34, Fig. 5) |
| *Cibicides lobatulus* (Walker and Jacob, 1798) | *Echinus lobatulus* Walker and Jacob, 1798 | Holbourn and Henderson, 2002 (Fig. 3. 1-3); Murray, 2003 (Fig. 6. 13-15) |
| *Cibicides mundulus* (Brady, Parker and Jones, 1888) | *Truncatulina mundula* Brady, Parker and Jones, 1888 | Holbourn and Henderson, 2002 (Fig. 4. 1-9); Margreth, 2010 (Pl. 33, Fig. 1) |
| *Cibicides refulgens* Montfort, 1808 | *Cibicides refulgens* Montfort, 1808 | Margreth, 2010 (Pl. 34, Fig. 4); Murray, 2003 (Fig. 7. 1-2) |
| *Cibicides ungerianus* (d'Orbigny, 1846) | *Rotalina ungeriana* d'Orbigny, 1846 | Margreth, 2010 (Pl. 34, Fig. 3) |
| *Cibicides wuellestorfi* (Schwager, 1866) | *Anomalina wuellestorfi* Schwager, 1866 | Holbourn and Henderson, 2002 (Fig. 5. 6-8); Margreth, 2010 (Pl. 35, Fig. 2) |
| *Clavulina cylindrica* d'Orbigny, 1826 | *Clavulina cylindrica* d'Orbigny, 1826 | Milker and Schmiedl, 2012 (Fig. 11. 7) |
| *Cornuspira foliacea* (Philippi, 1844) | *Orbis foliaceus* Philippi, 1844 | Milker and Schmiedl, 2012 (Fig. 11. 24-25) |
| *Cycloforina stalkeri* (Loeblich and Tappan, 1953) | *Quinqueloculina stalkeri* Loeblich and Tappan, 1953 | Margreth, 2010 (Pl. 7, Fig. 6) |
| *Dentalina advena* (Cushman, 1923) | *Nodosaria advena* Cushman, 1923 | Jones, 1994 (Pl. 63, Fig. 1) |
| *Dentalina bradyensis* (Dervieux, 1894) | *Laevidentalina haueri* Neugeboren, 1856 | Jones, 1994 (Pl. 62, Fig. 19-20) |
| *Dentalina guttifera* d'Orbigny, 1846 | *Dentalina guttifera* d'Orbigny, 1846 | Milker and Schmiedl, 2012 (Fig. 18. 13) |
| *Discanomalina coronata* (Parker and Jones, 1865) | *Anomalina coronata* Parker and Jones, 1865 | Margreth, 2010 (Pl. 39, Fig. 2) |
| *Discanomalina japonica* Asano, 1951 | *Discanomalina japonica* Asano, 1951 | Margreth, 2010 (Pl. 39, Fig. 3) |
| *Discanomalina semipunctata* (Bailey, 1851) | *Rotalina semipunctata* Bailey, 1851 | Milker and Schmiedl, 2012 (Fig. 26. 24-25) |
| *Discanomalina vermiculata* (d'Orbigny, 1839) | *Truncatulina vermiculata* d'Orbigny, 1839 | Jones, 1994 (Pl. 97, Fig. 7) |
| *Discorbinella bertheloti* (d'Orbigny, 1839) | *Rosalina bertheloti* d'Orbigny, 1839 | Kaminski et al., 2002 (Pl- 5, Fig. 1-2); Milker and Schmiedl, 2012 (Fig. 23. 29-30) |
| *Eggerella bradyi* (Cushman, 1911) | *Verneuilina bradyi* Cushman, 1911 | Holbourn and Henderson, 2002 (Fig. 1. 12-13) |
| *Eggerella humboldti* Todd and Brönniman, 1957 | *Eggerella humboldti* Todd and Brönniman, 1957 | Margreth, 2010 (Pl. 5, Fig. 2) |
| *Elphidium aculeatum* (d'Orbigny, 1846) | *Polystomella aculeata* d'Orbigny, 1846 | Milker and Schmiedl, 2012 (Fig. 27. 5-6) |
| *Elphidium crispum* (Linnaeus, 1758) | *Nautilus crispus* Linnaeus, 1758 | Cimerman and Langer, 1991 (Pl. 90, Fig. 1-6) |
| *Elphidium excavatum* (Terquem, 1875) | *Polystomella excavata* Terquem, 1875 | Darling et al., 2016 (Fig. 3, F) |
| *Favulina hexagona* (Williamson, 1848) | *Entosolenia squamosa var. hexagona* Williamson, 1848 | Margerel, 2016 (Fig. 2, A-I); Milker and Schmiedl, 2012 (Fig. 19. 4) |
| *Favulina squamosa* (Montagu, 1803) | *Vermiculum squamosum* Montagu, 1803 | Margerel, 2016 (Fig. 3, A-B); Margreth, 2010 (Pl. 14, Fig. 5) |
| *Fissurina alveolata* (Brady, 1884) | *Lagena alveolata* Brady, 1884 | Jones, 1994 (Pl. 60, Fig. 30-32) |

| Species | Original name | References used for identification |
|---|---|---|
| *Fissurina eburnea* (Buchner, 1940) | *Lagena eburnea* Buchner, 1940 | Margreth, 2010 (Pl. 16, Fig. 2) |
| *Fissurina fasciata* (Egger, 1857) | *Oolina fasciata* Egger, 1857 | Milker and Schmiedl, 2012 (Fig. 19. 8-9) |
| *Fissurina lacunata* (Burows and Holland, 1895) | *Lagena lacunata* Burrows and Holland, 1895 | Margreth, 2010 (Pl. 17, Fig. 6); Milker and Schmiedl, 2012 (Fig. 19. 10-11) |
| *Fissurina orbignyana* Seguenza, 1862 | *Fissurina orbignyana* Seguenza, 1862 | Milker and Schmiedl, 2012 (Fig. 19. 13); Murray, 2003 (Fig. 5. 5-6) |
| *Fursenkoina acuta* (d'Orbigny, 1846) | *Polymorphina acuta* d'Orbigny, 1846 | Milker and Schmiedl, 2012 (Fig. 21. 10-11) |
| *Gaudryina pseudotrochus* (Cushman 1922) | *Textularia pseudotrochus* Cushman, 1922 | Margreth, 2010 (Pl. 5, Fig. 1) |
| *Gaudryina rudis* Wright, 1900 | *Gaudryina rugosa* Wright, 1900 | Milker and Schmiedl, 2012 (Fig. 10. 6); Murray, 2003 (Fig. 2. 12-13) |
| *Gavelinopsis praegeri* (Heron-Allen and Earland, 1913) | *Discorbina praegeri* Heron-Allen and Earland, 1913 | Margreth, 2010 (Pl. 31, Fig. 2); Murray, 2003 (Fig. 8. 5-6) |
| *Glabratella patelliformis* (Brady, 1884) | *Discorbina patelliformis* Brady, 1884 | Margreth, 2010 (Pl. 32, Fig. 3); Milker and Schmiedl, 2012 (Fig. 23. 16-17) |
| *Glandulina ovula* d'Orbigny, 1846 | *Glandulina ovula* d'Orbigny, 1846 | Jones, 1994 (Pl. 63, Fig. 6) |
| *Globobulimina affinis* (d'Orbigny, 1839) | *Bulimina affinis* d'Orbigny, 1839 | Margreth, 2010 (Pl. 28, Fig. 2-3); Milker and Schmiedl, 2012 (Fig. 20. 24) |
| *Globobulimina doliolum* (Terquem and Terquem, 1886) | *Bulimina doliolum* Terquem and Terquem, 1886 | Margreth, 2010 (Pl. 28, Fig. 5) |
| *Globobulimina ovula* (d'Orbigny, 1839) | *Bulimina ovula* d'Orbigny, 1839 | Loeblich and Tappan, 1994 (Pl. 244, Fig. 15-16) |
| *Globobulimina pyrula* (d'Orbigny, 1846) | *Bulimina pyrula* d'Orbigny, 1846 | Fontanier et al., 2013 (Fig. 3. 30) |
| *Globobulimina turgida* (Bailey, 1851) | *Bulimina turgida* Bailey, 1851 | Margreth, 2010 (Pl. 28, Fig. 1) |
| *Globocassidulina subglobosa* (Brady, 1881) | *Cassidulina subglobosa* Brady, 1881 | Milker and Schmiedl, 2012 (Fig. 20. 13-14); Murray, 2003 (Fig. 8. 7) |
| *Glomospira charoides* (Jones and Parker, 1860) | *Trochammina squamata var. charoides* Jones and Parker, 1860 | Margreth, 2010 (Pl. 2, Fig. 5) |
| *Grigelis orectus* Loeblich and Tappan, 1994 | *Grigelis orectus* Loeblich and Tappan, 1994 | Margreth, 2010 (Pl. 12, Fig. 3) |
| *Gyroidina altiformis* Stewart and Stewart, 1930 | *Gyroidina altiformis* Stewart and Stewart, 1930 | Margreth, 2010 (Pl. 40, Fig. 3) |
| *Gyroidina lamarckiana* (d'Orbigny, 1839) | *Rotalina lamarckiana* d'Orbigny, 1839 | Kaminski et al., 2002 (Pl. 4, Fig. 11-12); Margreth, 2010 (Pl. 39, Fig. 5) |
| *Gyroidina soldanii* d'Orbigny, 1826 | *Gyroidina soldanii* d'Orbigny, 1826 | Margreth, 2010 (Pl. 40, Fig. 1) |
| *Homalohedra eucostata* (McCulloch, 1977) | *Oolina eucostata* McCulloch, 1977 | Margreth, 2010 (Pl. 15, Fig. 2) |
| *Homalohedra williamsoni* (Alcock, 1865) | *Entosolenia williamsoni* Alcock, 1865 | Margreth, 2010 (Pl. 14, Fig. 8) |
| *Hoeglundina elegans* (d'Orbigny, 1826) | *Rotalia (Turbinuline) elegans* d'Orbigny, 1826 | Margreth, 2010 (Pl. 18, Fig. 3); Milker and Schmiedl, 2012 (Fig. 19. 15-16) |
| *Hyalinea balthica* (Schröter in Gmelin, 1791) | *Nautilus balthicus* Schröter, 1791 | Milker and Schmiedl, 2012 (Fig. 24. 1-2); Murray, 2003 (Fig. 8. 8-10) |
| *Hyalinonetrion gracillimum* (Seguenza, 1862) | *Amphorina gracillima* Seguenza, 1862 | Margreth, 2010 (Pl. 14, Fig. 1); Milker and Schmiedl, 2012 (Fig. 18. 30) |
| *Hyrrokkin sarcophaga* Cedhagen, 1994 | *Hyrrokkin sarcophaga* Cedhagen, 1994 | Margreth, 2010 (Pl. 30, Fig. 5) |
| *Karreriella bradyi* (Cushman, 1911) | *Gaudryina bradyi* Cushman, 1911 | Holbourn and Henderson, 2002 (Fig. 2.1. 4-5); Margreth, 2010 (Pl. 5, Fig. 3) |

| Species | Original name | References used for identification |
|---|---|---|
| *Lachlanella bradyana* Cushman, 1917 | *Lachlanella bradyana* Cushman, 1917 | Milker and Schmiedl, 2012 (Figure 14. 19-21) |
| *Lagena doveyensis* Haynes, 1973 | *Lagena doveyensis* Haynes, 1973 | Milker and Schmiedl, 2012 (Figure 18. 31) |
| *Lenticulina calcar* (Linnaeus, 1758) | *Nautilus calcar* Linnaeus, 1758 | Margreth, 2010 (Pl. 12, Fig. 5); Milker and Schmiedl, 2012 (Fig. 18. 17-18) |
| *Lenticulina gibba* (d'Orbigny, 1839) | *Cristellaria gibba* d'Orbigny, 1839 | Margreth, 2010 (Pl. 12, Fig. 7) |
| *Lenticulina orbicularis* (d'Orbigny, 1826) | *Robulina orbicularis* d'Orbigny, 1826 | Margreth, 2010 (Pl. 12, Fig. 8); Milker and Schmiedl, 2012 (Fig. 18. 19-20) |
| *Lenticulina vortex* (Fichtel and Mol, 1798) | *Nautilus vortex* Fichtel and Moll, 1798 | Margreth, 2010 (Pl. 12, Fig. 6) |
| *Marginulina obesa* (Cushman, 1923) | *Marginulina glabra var. obesa* Terquem, 1866 | Jones, 1994 (Pl. 65, Fig. 5-6) |
| *Melonis barleeanum* (Williamson, 1858) | *Nonionina barleeana* Williamson, 1858 | Milker and Schmiedl, 2012 (Fig. 26. 11-12); Murray, 2003 (Fig. 8. 11, 14) |
| *Melonis pompiloides* (Fichtel and Moll, 1798) | *Nautilus pompilioides* Fichtel and Moll, 1798 | Margreth, 2010 (Pl. 37, Fig. 5) |
| *Miliolinella elongata* (Kruit, 1955) | *Miolinella circularis var. elongata* Kruit, 1955 | Margreth, 2010 (Pl. 9, Fig. 1); Milker and Schmiedl, 2012 (Fig. 16. 23-24) |
| *Miliolinella subrotunda* (Montagu, 1803) | *Vermiculum subrotundum* Montagu, 1803 | Milker and Schmiedl, 2012 (Fig. 16. 31-32); Murray, 2003 (Fig. 4. 6) |
| *Neoconorbina terquemi* (Rzehak, 1888) | *Discorbina terquemi* Rzehak, 1888 | Milker and Schmiedl, 2012 (Fig. 22. 5-6) |
| *Neolenticulina variabilis* (Reuss, 1850) | *Cristellaria variabilis* Reuss, 1850 | Jones, 1994 (Pl. 68, Fig. 11-16) |
| *Nodosaria lamnulifera* Boomgaart, 1950 | *Nodosaria lamnulifera* Boomgaart, 1950 | Jones, 1994 (Pl. 64, Fig. 6-10) |
| *Nonion fabum* (Fichtel and Moll, 1798) | *Nautilus faba* Fichtel and Moll, 1798 | Margreth, 2010 (Pl. 36, Fig. 2); Milker and Schmiedl, 2012 (Fig. 25. 22-24) |
| *Nonionella turgida* (Williamson, 1858) | *Rotalina turgida* Williamson, 1858 | Milker and Schmiedl, 2012 (Fig. 26. 1-6); Murray, 2003 (Fig. 9. 4-5) |
| *Nuttallides umbonifera* (Cushman, 1933) | *Pulvinulina umbonifera* Cushman, 1933 | Margreth, 2010 (Pl. 35, Fig. 4) |
| *Oolina globosa* (Montagu, 1803) | *Vermiculum globosum* Montagu, 1803 | Margreth, 2010 (Pl. 15, Fig. 5) |
| *Oolina lineata* (Williamson, 1848) | *Entosolenia lineata* Williamson, 1848 | Margreth, 2010 (Pl. 15, Fig. 3) |
| *Oolina melo* d'Orbigny, 1839 | *Oolina melo* d'Orbigny, 1839 | Margreth, 2010 (Pl. 14, Fig. 6) |
| *Parabrizalina porrecta* (Brady, 1881) | *Bulimina porrecta* Brady, 1881 | Margreth, 2010 (Pl. 25, Fig. 2) |
| *Parafissurina lateralis* (Cushman, 1913) | *Lagena lateralis* Cushman, 1913 | Jones, 1994 (Pl. 56, Fig. 17-18); Milker and Schmiedl, 2012 (Fig. 19. 14) |
| *Patellina corrugata* Williamson, 1858 | *Patellina corrugata* Williamson, 1858 | Milker and Schmiedl, 2012 (Fig. 11. 21-23); Murray, 2003 (Fig. 9. 7) |
| *Planispirinoides bucculentus* (Brady, 1884) | *Miliolina bucculenta* Brady, 1884 | Jones, 1994 (Pl. 114, Fig. 3) |
| *Planodiscorbis rarescens* (Brady, 1884) | *Discorbina rarescens* Brady, 1884 | Milker and Schmiedl, 2012 (Fig. 22. 7-8) |
| *Planulina ariminensis* d'Orbigny, 1826 | *Planulina ariminensis* d'Orbigny, 1826 | Margreth, 2010 (Pl. 34, Fig. 1); Milker and Schmiedl, 2012 (Fig. 24. 3-4) |
| *Planulina mediterranensis* d'Orbigny, 1826 | *Planulina mediterranensis* d'Orbigny, 1826 | Milker and Schmiedl, 2012 (Fig. 24. 21-24) |
| *Pullenia subcarinata* (d'Orbigny, 1839) | *Nonionina subcarinata* d'Orbigny, 1839 | Margreth, 2010 (Pl. 38, Fig. 2) |
| *Pygmaeoseistron laevis ovalis* Williamson, 1848 | *Pygmaeoseistron laevis ovalis* Williamson, 1848 | Margreth, 2010 (Pl. 13, Fig. 10) |

**Annex 2**. continuation

| Species | Original name | References used for identification |
|---|---|---|
| *Pyrgo comata* (Brady, 1881) | *Biloculina comata* Brady, 1881 | Jones, 1994 (Pl. 3, Fig. 9); Margreth, 2010 (Pl. 9, Fig. 3) |
| *Pyrgo depressa* (d'Orbigny, 1826) | *Biloculina depressa* d'Orbigny, 1826 | Milker and Schmiedl, 2012 (Fig. 17. 10-11) |
| *Pyrgo elongata* (d'Orbigny, 1826) | *Biloculina elongata* d'Orbigny, 1826 | Margreth, 2010 (Pl. 10, Fig. 5); Milker and Schmiedl, 2012 (Fig. 17. 12) |
| *Pyrgo sarsi* (Schlumberger, 1891) | *Biloculina sarsi* Schlumberger, 1891 | Margreth, 2010 (Pl. 10, Fig. 1-2) |
| *Pyrgoella sphaera* (d'Orbigny, 1839) | *Biloculina sphaera* d'Orbigny, 1839 | Milker and Schmiedl, 2012 (Fig. 17. 7) |
| *Quinqueloculina berthelotiana* d'Orbigny, 1839 | *Quinqueloculina berthelotiana* d'Orbigny, 1839 | Milker and Schmiedl, 2012 (Fig. 15. 3-6) |
| *Quinqueloculina laevigata* d'Orbigny, 1839 | *Quinqueloculina laevigata* d'Orbigny, 1839 | Milker and Schmiedl, 2012 (Fig. 15. 13-15) |
| *Quinqueloculina neapolitana* Sgarrella and Moncharmont Zei, 1993 | *Quinqueloculina neapolitana* Sgarrella and Moncharmont Zei, 1993 | Milker and Schmiedl, 2012 (Fig. 15. 20-22) |
| *Quinqueloculina parvula* Schlumberger, 1894 | *Quinqueloculina parvula* Schlumberger, 1894 | Milker and Schmiedl, 2012 (Fig. 15. 25-27) |
| *Quinqueloculina semiluna* Terquem, 1758 | *Quinqueloculina semiluna* Terquem, 1758 | Margreth, 2010 (Pl. 7, Fig. 8); Milker and Schmiedl, 2012 (Fig. 15. 30-31) |
| *Quinqueloculina stelligera* Schlumberger, 1893 | *Quinqueloculina stelligera* Schlumberger, 1893 | Milker and Schmiedl, 2012 (Fig. 16. 1-4) |
| *Quinqueloculina viennensis* Le Calvez and Le Calvez, 1958 | *Quinqueloculina viennensis* Le Calvez and Le Calvez, 1958 | Margreth, 2010 (Pl. 7, Fig. 7); Milker and Schmiedl, 2012 (Fig. 16. 5-7) |
| *Reussella spinulosa* (Reuss, 1850) | *Verneuilina spinulosa* Reuss, 1850 | Milker and Schmiedl, 2012 (Fig. 21. 6-7) |
| *Robertinoides bradyi* (Cushman and Parker, 1936) | *Robertina bradyi* Cushman and Parker, 1936 | Jones, 1994 (Pl. 50, Fig. 18); Margreth, 2010 (Pl. 18, Fig. 7) |
| *Rosalina anomala* Terquem, 1875 | *Rosalina anomala* Terquem, 1875 | Murray, 2003 (Fig. 9. 9-10) |
| *Rosalina bradyi* (Cushman, 1915) | *Discorbis globularis var. bradyi* Cushman, 1915 | Margreth, 2010 (Pl. 31, Fig. 5); Milker and Schmiedl, 2012 (Fig. 22. 11-14) |
| *Sigmoilopsis schlumbergeri* (Silvestri, 1904) | *Sigmoilina schlumbergeri* Silvestri, 1904 | Jones, 1994 (Pl. 8, Fig. 1-4); Milker and Schmiedl, 2012 (Fig. 18. 7-8) |
| *Siphonotextularia fretensis* Vella, 1957 | *Siphonotextularia fretensis* Vella, 1957 | Loeblich and Tappan, 1994 (Pl. 41, Fig. 1-4) |
| *Sphaeroidina bulloides* d'Orbigny, 1828 | *Sphaeroidina bulloides* d'Orbigny, 1828 | Margreth, 2010 (Pl. 32, Fig. 2); Milker and Schmiedl, 2012 (Fig. 23. 3-4) |
| *Spirillina limbata* Brady, 1879 | *Spirillina limbata* Brady, 1879 | Milker and Schmiedl, 2012 (Fig. 11. 13-14) |
| *Spirillina vivipara* Ehrenberg, 1843 | *Spirillina vivipara* Ehrenberg, 1843 | Milker and Schmiedl, 2012 (Fig. 11. 15-16); Murray, 2003 (Fig. 4. 1) |
| *Spiroloculina asperula* Karrer, 1868 | *Spiroloculina asperula* Karrer, 1868 | Jones, 1994 (Pl. 8, Fig. 14) |
| *Spiroloculina bradyi* Barker, 1960 | *Spiroloculina bradyi* Barker, 1960 | Jones, 1994 (Pl. 10, Fig. 1-2) |
| *Spiroloculina excavata* d'Orbigny, 1846 | *Spiroloculina excavata* d'Orbigny, 1846 | Milker and Schmiedl, 2012 (Fig. 13. 3-4); Murray, 2003 (Fig. 4. 13-14) |
| *Spiroloculina rostrata* Reuss, 1850 | *Spiroloculina rostrata* Reuss, 1850 | Milker and Schmiedl, 2012 (Fig. 13. 5-6) |
| *Spiroloculina tenuissima* Reuss, 1867 | *Spiroloculina tenuissima* Reuss, 1867 | Jones, 1994 (Pl. 10, Fig. 9-10) |
| *Spiroplectammina expansa* (Plummer, 1927) | *Textularia carinata var. expansa* Plummer, 1927 | Plummer, 1926 (Pl. 3, Fig. 3) |
| *Spliroplectammina wrightii* (Silvestri, 1903) | *Spiroplecta wrightii* Silvestri, 1903 | Jones, 1994 (Pl. 42, Fig. 17-18); Margreth, 2010 (Pl. 3, Fig. 6) |
| *Stomatorbina concentrica* (Parker and Jones, 1864) | *Pulvinulina concentrica* Parker and Jones, 1864 | Margreth, 2010 (Pl. 30, Fig. 3); Milker and Schmiedl, 2012 (Fig. 21. 24-25) |

| Species | Original name | References used for identification |
|---|---|---|
| *Textularia agglutinans* d'Orbigny, 1839 | *Textularia agglutinans* d'Orbigny, 1839 | Milker and Schmiedl, 2012 (Fig. 10. 15-16) |
| *Textularia calva* Lalicker, 1935 | *Textularia calva* Lalicker, 1935 | Milker and Schmiedl, 2012 (Fig. 10. 17) |
| *Textularia conica* d'Orbigny, 1839 | *Textularia conica* d'Orbigny, 1839 | Milker and Schmiedl, 2012 (Fig. 10. 18) |
| *Textularia gramen* d'Orbigny, 1846 | *Textularia gramen* d'Orbigny, 1846 | Milker and Schmiedl, 2012 (Fig. 10. 19-20) |
| *Textularia pala* Cžjžek, 1848 | *Textularia pala* Cžjžek, 1848 | Frontalini et al., 2014 (Fig. 4. 3); Milker and Schmiedl, 2012 (Fig. 10. 21-22) |
| *Textularia pseudorugosa* Lacroix, 1931 | *Textularia pseudorugosa* Lacroix, 1931 | Milker and Schmiedl, 2012 (Fig. 10. 23-24) |
| *Textularia truncata* Höglund, 1947 | *Textularia truncata* Höglund, 1947 | Margreth, 2010 (Pl. 6, Fig. 1); Murray, 2003 (Fig. 3. 17-18) |
| *Trifarina angulosa* (Williamson, 1858) | *Uvigerina angulosa* Williamson, 1858 | Margreth, 2010 (Pl. 29, Fig. 6); Murray, 2003 (Fig. 10. 5) |
| *Trifarina bradyi* Cushman, 1923 | *Trifarina bradyi* Cushman, 1923 | Margreth, 2010 (Pl. 29, Fig. 7) |
| *Triloculina marioni* Schlumberger, 1893 | *Triloculina marioni* Schlumberger, 1893 | Margreth, 2010 (Pl. 11, Fig. 1); Milker and Schmiedl, 2012 (Fig. 17. 17-18) |
| *Triloculina tricarinata* d'Orbigny in Deshayes, 1832 | *Triloculina tricarinata* d'Orbigny in Deshayes, 1832 | Margreth, 2010 (Pl. 11, Fig. 3); Milker and Schmiedl, 2012 (Fig. 17. 23-24) |
| *Uvigerina auberiana* d'Orbigny, 1839 | *Uvigerina auberiana* d'Orbigny, 1839 | Erdem and Schönfeld, 2017 (Fig. 10. 17); Margreth, 2010 (Pl. 29, Fig. 3) |
| *Uvigerina mediterranea* Hofker, 1932 | *Uvigerina mediterranea* Hofker, 1932 | Margreth, 2010 (Pl. 29, Fig. 1); Milker and Schmiedl, 2012 (Fig. 20. 28) |
| *Uvigerina peregrina* Cushman, 1923 | *Uvigerina peregrina* Cushman, 1923 | Milker and Schmiedl, 2012 (Fig. 20. 29); Murray, 2003 (Fig. 10. 6) |
| *Valvulineria bradyana* (Fornasini, 1900) | *Discorbina bradyana* Fornasini, 1900 | Margreth, 2010 (Pl. 30, Fig. 1) |

**Annex 2 - References**

Cimerman, F., Langer, M.R., 1991. Mediterranean Foraminifera. Academia Scientarium et Artium Slovenica, Dela, Opera 30, Classis IV: Historia Naturalis, Ljubljana, 107 pp.

Darling, K., Schweizer, M., Knudsen, K., Evans, K., Bird, C., Roberts, A., Filipsson, H., Kim, J., Gudmundsson, G., Wade, C., Sayer, M., Austin, W., 2016. The genetic diversity, phylogeography and morphology of Elphidiidae (Foraminifera) in the Northeast Atlantic. Marine Micropaleontology 129, 1-23.

Erdem, Z., Schönfeld, J., 2017. Pleistocene to Holocene benthic foraminiferal assemblages from the Peruvian continental margin. Palaeontologia Electronica 20.2.35A, 1-32.

Fontanier, C., Metzger, E., Waelbroeck, C., Jouffreau, M., LeFloch, N., Jorissen, F., Etcheber, H., Bichon, S., Chabaud, G., Poirier, D., Grémare, A., Deflandre, B., 2013. Live (Stained) Benthic Foraminifera Off Walvis Bay, Namibia: A Deep-Sea Ecosystem under the Influence of Bottom Nepheloid Layers. Journal of Foraminiferal Research 43(1), 55-71.

Fornasini, C., 1908. Illustrazione di specie orbignyane di Nodosaridi, di Rotalidi e d'altri foraminiferi. Memorie della Reale Accademie della Scienze dell'Istituto di Bologna, Scienze Naturali 6(5).

Frontalini, F., Kaminski, M., Mikellidou, I., Armynot du Châtelet, E., 2014. Checklist of benthic foraminifera (class Foraminifera: d'Orbigny 1826; phylum Granuloreticulosa) from Saros Bay, northern Aegean Sea: a biodiversity hotspot. Marine Biodiversity 45(3), 549-567.

Holbourn, A.E., Hendreson, A.S., 2002. Re-illustration and Revised Taxonomy for Selected Deep-sea Benthic Foraminifers. Palaeontologia Electronica 4(2), 34 pp.

Hottinger, L., Halicz, E., Reiss, Z., 1993. Recent Foraminifera from the Gulf of Aqaba, Red Sea. Academia Scientarium et Artium Slovenica, Dela, Opera 33, Classis IV: Historia Naturalis, Ljubljana, 423 pp.

Jones, R.W., 1994. The Challenger Foraminifera. Oxford University Press. 291 pp.

Kaminski, M.A., Aksu, A., Box, M., Hiscott, R.N., Filipescu, S., Al-Salameen, M., 2002. Late Glacial to Holocene benthic foraminifera in the Marmara Sea: implications for Black Sea Mediterranean Sea connections following the last deglaciation. Marine Geology 190, 165-202.

Loeblich, A.R., Tappan, H., 1988. Foraminiferal Genera and Their Classification. Van Nostrand Reinhold Company, New York, 2 Volumes, 970 pp.

Loeblich, A.R., Tappan, H., 1994. Foraminifera of the Sahul Shelf and Timor Sea. Cushman Foundation for Foraminiferal Research Special Publication 31, 13–630.

Margerel, J-P., 2016. Étude critique des genres Favulina, Homalohedra, Oolina, Entosolenia et Pseudofavulina n. gen. du Pliocène et du Pléistocène inférieur de la France occidentale et du Sud de l'Angleterre. Geodiversitas 38(4), 559-578.

Margreth, S., 2010. Benthic foraminifera associated to cold-water coral ecosystems. PhD Thesis n°1686, University of Fribourg. Geofocus 24, 248 pp.

Milker, Y., Schmiedl, G., 2012. A taxonomic guide to modern benthic shelf foraminifera of the western Mediterranean Sea. Palaeontologia Electronica 15.2. 16A, 134 pp.

Murray, J.W., 2003. An illustrated guide to the benthic foraminifera of the Hebridean Shelf, West of Scotland, with notes on their mode of life. Palaeontologia Electronica 5(1), 31 pp.

Plummer, H.J., 1926. Foraminifera of the Midway Formation in Texas. University of Texas Bulletin 2644, 209 pp.

Setoyama, E., Kaminski, M., 2015. Neogene benthic foraminifera from the southern Bering Sea (IODP Expedition 323). 18.2.38A, 1-30.

---

## Short Comment (SC2) · 6 Sep 2020

This is a reply of the authors to the Short Comment of Silvia Spezzaferri from 05.09.2020:

First of all we would like to acknowledge all referees that they took their time and effort to critically review our manuscript. We appreciate all the comments and suggestions and integrated all in our best way possible!

In the following, we dispute (indicated by "Reply") the scientific comments from Dr.

[Figure]

Spezzaferri's (indicated by "SC").

SC: In general, the interpretation is forced, giving CWC foraminifera a "clear, fixed and not questionable" significance, which may not be the case, especially for these ecosystems that are not completely well understood. These organisms can easily adapt and change their ecological preferences according to geographical location, oceanographic parameters, e.g., water depth, substratum, salinity, temperature, etc: : :: : :.(e.g., the same species can live in relatively shallower or deeper water according to the type of substratum, the same applies for all other parameters, e.g., salinity, nutrient and oxygen availability). In the manuscript all the discussion is based on given and fixed foraminifera ecological preference taken from the literature and in different geographical setting, instead of starting from establish proxies (e.g. TOC) and then interpret foraminiferal data.

Reply: It should be stated that foraminifera species and assemblages are a very powerful proxy to study past environmental settings. Lots of studies have been proven this approach. …..please add references. The ecological preference of certain species and assemblages is used to interpret past environmental settings and compared to other ecological or environmental parameters based on different sedimentological, macrofaunal and geochemical proxies. In this manuscript, several proxies are combined and compared to the studied foraminiferal assemblages to draw the final conclusions. This classical multiproxy approach has been used to get solid interpretations. It should be mentioned that often lots of weight is only put on one or the other proxy, while this study is integrating both, foraminiferal proxy and geochemical datasets. To highlight the critical note of the authors towards using foraminifera having a fixed ecological preference being bound to certain settings, just one example: Earlier publications (Margreth et al., 2009, 2011; Stalder et al., 2014; Spezzaferri et al., 2013) use single species (i.e. Discanomalina coronata) as bioindicator for cold-water coral reefs, although other studies showed that this species occurs in high-energy environments (Schönfeld, 2002) or is not or very low abundant in other CWC reef sediments (e.g., Smeulders et al., 2014).

In this manuscript we discuss that D. coronata not directly associates to periods where CWC reefs thrive and question their ecological preference.

SC: Every situation must be evaluated case by case and anyhow a complete dataset including fractions smaller than 125 m should be presented. Explainig everything with displacement is not a real reason. The same applies to the counting of the plankton, is more a problem of time consuming than scientific. To demonstrate that it is a scientific reason, data sghould be presented first and tehn excluded. The >125 m can be useful when making taxonomic work e.g., taxonomic atlases and guides with plates (e.g., Milker and Schmiedl, 2012) but not for ecological purposes, in this case the 40 m frection counts should be presented and eventually afterward not included in the discussion.

Reply: The fraction > 125 $\mu$m has been often used in settings characterized by high currents (e.g., Lutze and Coulbourn, 1984). Also the study of Stalder et al. (2015) based on sediment cores from nearby CWC mounds of the Melilla Carbonate Mound Field presents data and interpretation only based on foraminiferal identification and interpretation of fractions >125 $\mu$m, this for the same reasons we mention in our manuscript ("[Stalder et al., 2015] decided to focus on specimens larger than 125 $\mu$m to exclude smaller forms, which are often displaced by redeposition (Lutze and Coulbourn, 1984), and to make the data comparable to other benthic foraminiferal studies in adjacent areas (Caralp, 1988; Vergnaud-Grazzini et al., 1989; Schönfeld, 2002; Milker and Schmiedl, 2012"). To integrate planktonic data is of course very important for future studies.

SC: It is not clear how he density of benthic foraminifera has been calculated. The method used should be better explained and should be specified the reason for the choice. The method used in the manuscript does not correspond to any of the generally used in micropaleontology...

Reply: Indeed, there are different ways to present total benthic foraminifera densities with respect to sample weight or accumulations rates (including time). As mentioned above, calculating foraminiferal densities on sample weight is biased in CWC settings due to the large abundance of macrofauna. Often, direct comparison to other foraminiferal studies is not possible. Therefore, the number of individuals per gram of fraction was chosen. However, to avoid confusion we will present the relative abundance (in %) of the benthic foraminifera to be comparable to the study of Stalder et al. (2015) and others.

SC: I would like also to comment on Figure 10, which looks very fancy but presents a few problems. First of all it is upside down (even if the cardinal points are marked), we conventionally (and geographically) see the African margin at South and European Margin at North. Not the vice versa. This confuses the reader.

Reply: This is a discussion of an artistic point of view. We prefer to present the Alboran Sea in this way looking towards the SW to better visualize the patterns occurring at the Melilla Carbonate Mound Field. The authors have made different versions of this figure and this orientation illustrates much better the interpretation. Sometimes a different perspective might change the view on traditional conventions. Of course, the authors make sure that N and S are clearly indicated on the respective figure.

SC: As commented above (WHERE?) during glacials the thermocline and pycnocline should be very shallow favoring water and nutrient mixing. In Figure 10 glacials are on the contrary described as stratified, the explanation for this is based only on comparison with modern times, it is generally confused and/or based on assumptions and circular reasoning. No clear evidence is presented. On the contrary interglacial are represented as are the typical models for high latitudes/glacial times e.g., with strong mixing of water mass and nutrients. I First of all in the Mediterranean this cannot be possible, even in the past, also considering the temperate latitude and seasonality. Additionally, if during interglacials fluvial input increased, then the fresh water plume arriving into the sea must have produced a clear separation of water masses (fluids with different densities) and not mixing. The closest large river is only at 50 km (Mouloya)

, if the fluvial input was so massive to trigger coral growth, then also the fresh water plume must have been significant enough to produce stratification not mixing. Other alternative processes must be discussed?

Reply: The authors have difficulties to understand the reasoning outlined above. To clarify the outcome, we clearly conclude the following in the manuscript: 1) Cold-water corals develop mainly during interglacial periods. Their growth is promoted by the combination of increased fluvial input and enhanced influence of Alboran Gyres. Increased fluvial organic matter inputs are driven by the increased impact of warm and moist Atlantic air masses with intensified Western and Eastern Alboran Gyres that lead to more important turnover between surface and intermediate water masses. This phenomenon is promoted by enhanced Modified Atlantic Water inflow at the Strait of Gibraltar. . . . These results demonstrate the paramount importance of enhanced fluvial input as a trigger for cold-water growth in the Southeastern Alboran Sea. This is illustrated in Figure 10. Interglacial. 2) Glacial periods are unfavourable for cold-water corals; in contrast the bryozoan Buskea dichotoma is more suited to glacial environmental conditions. The retreat of corals during glacial periods is triggered by arid continental conditions that lead to reduced fluvial input and nutrient supply. Moreover, reduced inflow of Modified Atlantic Water at the Strait of Gibraltar results in a lower contribution of surface waters to intermediate waters. In contrast, the contribution of Western Mediterranean Deep Water to intermediate water masses increased. Weaker Alboran Gyres and increased contribution of well-ventilated deep waters at intermediate depths resulted in increased stratification. Lower input of organic matter, but less degraded, further characterizes glacial environmental conditions. Aeolian dust was the main fertilizing influence and may have enabled corals to survive throughout glacial periods. This is illustrated in Figure 10. Glacial.

SC: If responsible for stratification in glacials are the stronger ShW then it must be demonstrated that they are indeed stronger (what ever "stronger" means: denser? colder?) and remarkably colder than at the surface to justify such a stratification acting

a physical barrier between the sea floor and the surface. And this is not possible with the present data. At least an intermediate water species should have been analyzed for oxygen isotopes and not only at the BRI site but also in the Atlantic waters, e.g., Cadiz to have the ShW signature, as these are the waters that are supposed to influence the Alboran Sea (e.g, as in the title). Only Atlantic or Mediterranean waters are marked in the figures. If there were rivers they have to be documented as they are not only today but also how they were in the past 400.000 years, according to geological information.

Reply: It is clearly described in the discussion (chapters 5.1 and 5.2) what the processes are with respect to water mass dynamics during glacial and interglacial periods. The increased stratification during cold periods is discussed in detail in chapter 5.2.2 where we integrate grain size data, carbon isotope data and benthic foraminiferal ecology. As the benthic foraminifera analyzed for oxygen isotopes at BRI represents the situation at intermediate water depths, only a transect from the deeper basin across the slope to the shelf would be the most suitable approach to characterize different water mass signature. However, such a transect of cores was not available (yet). Furthermore, we cannot follow the reference to the Gulf of Cadiz with respect to the ShW, as the ShW is a local phenomenon <300 m described for the Eastern Alboran Sea offshore Morocco as a mixture of MAW and WMDW (see Ercilla et al., 2016) - this (strictly not a) water mass does not occur in the Gulf of Cadiz. Rivers are indicated in figure 10, however, where exactly the rivers were draining into the Mediterranean Sea during glacials is less certain but possibly similar to today.

SC: Last but not least and for respect to the funding agency the first author Robin Fentimen should also acknowledge the Swiss National Science Foundation Project Ref. 200020_153125 "Faunal assemblages from active, declining and buried cold-water coral ecosystems" that payed his salary for 3 years over the 4 years of his PhD, and that has co-funded with the amount of 54.000 Euro the cruise Eurofleets GATEWAY, MD194 during which the cores investigated in this research were retrieved.

Reply: We thank you for this comment. Of course, we will acknowledge this project

properly.

References in this Reply:

Caralp MH (1988) Late glacial to recent deep-sea benthic foraminifera from the Northeastern Atlantic (Cadiz Gulf) and Western Mediterranean (Alboran Sea): paleaooceanographic results. Marine Micropaleaotology 13: 265–289.

Ercilla, G., Juan, C., Hernández-Molina, J., Bruno, M., Estrada, F., Alonso, B., Casas, D., Farran, M., Llave, E., García, Vázquez, J.T., D'Acremont, E., Gorini, C., Palomino, D., Valencia, J., El Moumni, B., and Ammar, A. (2016) Significance of bottom currents in deep-sea morphodynamics: An example from the Alboran Sea, Mar Geol, 378, 157-170.

Lutze GF, Coulbourn WT (1984) Recent benthic foraminifera from the continental margin of northwest Africa: Community structure and distribution. Marine Micropaleontology 8: 361–401.

Margreth, S., Rüggeberg, A., and Spezzaferri, S. (2009) Benthic foraminifera as bioindicator for cold-water coral reef ecosystems along the Irish margin, Deep-Sea Res Pt I, 56, 2216-2234.

Margreth, S., Gennari, G., Rüggeberg, A., Comas, M.C., Pinheiro, L.M., Spezzaferri, S. (2011) Growth and demise of cold-water coral ecosystems on mud volcanoes in the West Alboran Sea: The message from planktonic and benthic foraminifera. Marine Geology 282: 26–39.

Milker Y, Schmiedl G (2012) A taxonomic guide to modern benthic shelf foraminifera of the western Mediterranean Sea. Palaeontologia Electronica 15(2): 1–134.

Schönfeld, J. (2002) Recent benthic foraminiferal assemblages in deep high-energy environments from the Gulf of Cadiz (Spain). Mar. Micropaleontol. 44: 141–162.

Schönfeld J (2002) A new benthic foraminiferal proxy for near-bottom current velocities

in the Gulf of Cadiz, northeastern Atlantic Ocean. Deep-Sea Research Part I 49: 1853–1875.

Smeulders, G.G.B., Koho, K.A., De Stigter, H.C., Mienis, F., de Haas, H., van Weering, T.C.E. (2014) Cold-water coral habitats of Rockall and Porcupine Bank, NE Atlantic Ocean: Sedimentary facies and benthic foraminiferal assemblages: Deep-Sea Research II 99: 270–285.

Spezzaferri, S., Rüggeberg, A., Stalder, C., Margreth, S. (2013) Benthic foraminifer assemblages from Norwegian cold-water coral reffs. Journal of Foraminiferal Research 43(1): 21–39.

Stalder, C., Vertino, A., Rosso, A., Ruggeberg, A., Pirkenseer, C., Spangenberg, J.E., Spezzaferri, S., Camozzi, O., Rappo, S., and Hajdas, I. (2015) Microfossils, a Key to Unravel Cold-Water Carbonate Mound Evolution through Time: Evidence from the Eastern Alboran Sea, PLoS One, 10, e0140223.

Vergnaud-Grazzini C, Caralp M, Faugères J-C, Gonthier E, Grousset F, Pujol C, et al. (1989) Mediterranean outflow through the Strait of Gibraltar since 18 000 years B.P. Oceanologica Acta 12: 305–324.

---

## Referee Comment (RC3) · Anonymous Referee #1 · 7 Sep 2020

Response to the authors' response: Despite the points raised by the authors in their response, I am still convinced that this manuscript should not be published. Although I still do not agree with some of the interpretations (which could be improved in a revision process), the main point for suggesting a rejection is the poor age model as pointed out before. In my eyes, the lack of a convincing age model correctly reflecting the presumably intermittent development of the sampled coral mound/ridge, precludes the publication with the available data base. And increasing this data base will take time and most likely significantly change the interpretations. I will not respond point-

by-point the comments by the authors. But below I will highlight a few points made by them. TOC data: The authors argue in their response: At no point in the manuscript is TOC data used as an indicator for productivity. In the discussion section, the mention to TOC can be found twice Line 399: "The overall higher TOC levels during interglacials confirm that the sediment during these periods was relatively enriched in organic matter in comparison to glacial periods . . ." Okay, this does not necessarily relate to productivity, but seeing this in context with the sentence just before (line 397): ". . . the benthic foraminiferal assemblage during interglacials would support a high organic matter export to the seafloor" indeed relates the TOC content to productivity. If the high organic matter export to the seafloor is "confirmed" by the TOC record, then I only can read it as TOC being used as a productivity indicator. Thus, I cannot follow the response by the authors mentioned above. Furthermore, the authors did not resolve the question about the meaning of the TOC record for this paper if it only represents terrigenous organic matter.

Stratigraphy: The authors respond that the chronology of the core has been developed based on the coral ages in conjunction with the d18O data and that this manuscript concentrates rather on characterizing changes between interglacial and glacial periods. However, in the ms it is stated that "due to difficulties to define precisely the stratigraphy of this section of the core (MIS 6 to MIS 9), it will not be considered in detail during this study (line 286)." Thus, if MIS 7 and MIS 9 will not be considered in detail and as MIS 1 is not discussed, how can generally conclusions about "interglacials" be made if "solid" information only exists for MIS 5? Basically the same refers to glacials . . . As pointed out in the first comments made, the stratigraphy of this core is not sufficiently constrained to allow the conclusions made. Furthermore, the authors refer in their response to the pioneering work of Dorschel et al. (2005) and Rüggeberg et al. (2007) from the Irish margin. As nicely pointed out there, coral mound records might have considerable age offsets between coral fragments and foraminifera dated from the same core level. This is also reflected in peculiar d18O data reflecting mainly intermediate (non-interglacial, non-glacial) climate stages while the corals are from interglacial periods. Thus, at Propeller Mound at the Irish margin the foram data do not allow to make any conclusions about the environment at times of coral growth. That is not necessarily the case for the Alboran Sea, but it might be and only a better age control could help here.

Foraminifera analyses The authors state that most of their conclusions are mainly built on the foram data, while other data sets (e.g., TOC, sortable silt) are only meant to support these. This is a valid approach, however, seen the diversity of benthic forams and their different behavior in different settings, often they are used to back up other proxies. In the first comment, I questioned the use of sortable silt in a coral mound setting. Now the authors state that despite all the problems raised, it is used to support the current strength record provided by the infaunal (!) T. angulosa. But to my perception, there is no fit at all between these two records. As pointed out before, it seems that first there was the interpretation and afterwards the data "have to fit". The authors also state that the interpretation of more humid conditions also is based mainly on the foram assemblage and only backed up by the XRF data (see comment on XRF data below). Well, higher abundances of some infaunal species probably points to higher OM fluxes driven by higher productivity. However, if this was fueled by fluvial discharge, by enhanced mixing or by advection of nutrients cannot be resolved from the benthic foraminifera assemblage . . . Overall, as the benthic forams are the most important proxy here, a more critical discussion about the interpretations based on the benthic forams provided here is lacking.

Mound aggradation In my response I made a hint to high mound ARs of >400 cm kyr-1 in the early Holocene as reported by other studies from the region. Now, thanks to the authors, I learned that these only partly occurred during the Early Holocene, but partly during the very end of MIS 2. Nevertheless, high ratios of >200 cm kyr-1 extend well into the Holocene. Indeed, this is a nice example that during the last deglaciation the corals were most active between ∼13.5 kyr BP and 9 kyrs BP, neither giving a clear hint to preferred glacial or interglacial conditions. As this was the state of the art, on

which the present ms should be built, why do the authors look for glacial – interglacial differences?

XRF Sorry, but the hint given in the response that the running mean on the XRF data would help solving the problems with measurements on corals is not sufficient at all. Talking about coral contents of up to >30% this does not work. Still, in my eyes the XRF data are very problematical to use. In addition, as pointed out before, the minor changes in the elemental records are heavily overinterpreted.

Finally, the authors respond: Moreover, we believe again that the time scale considered in this study (300 ky for a 920 cm long core) allows to identify more long-term environmental changes than those from Fink et al. (2013), Stalder et al. (2015) or Fentimen et al. (2020). The time-scale covered by these studies allow to identify precisely short but rapid periods of mound aggradation. This study does not aim to do this, but rather to look at the wider picture. This only will work based on a solid stratigraphy. And I am pretty convinced, a solid stratigraphy will show, that also this record provides a serious of short intermittent pulses of mound aggradation, which very nicely will fit the records of these previous studies (as at almost all up to now known coral mound sites). Thus, it can at least be questioned if there is such an impact of long-term environmental changes . . . In addition, the authors "believe that a core covering the last two interglacials allows to draw more solid conclusions about the impact of environmental changes on mound development than a precise study of the last 15 ky.". Well, assuming intermittent mound aggradation also prior to 15 kyr BP (what is not questioned by the authors), I cannot follow this statement.

---

## Author Comment (AC5) · 7 Oct 2020

We first of all would like to thank both reviewers for the time spent reviewing and commenting this manuscript. We agree that some points raised need to be addressed and this shall be done in a revised version of this manuscript. Indeed, in the same way as Reviewer #2, we believe that the novelty of this dataset which for the first time investigates the development of the Melilla Mounds over the last 300 ka, deserves to be published. Regarding the suggestion to reject this manuscript proposed by Reviewer #1, we do not believe that the issues raised cannot be corrected promptly and as such

we find this opinion harsh (see fist response to Reviewer #1). However, we accept the final decision although we regret the lack of discussion offered by Reviewer #1. We will come back with a stronger manuscript and a stratigraphy constructed around additional U-series isotope datings. Below we raise a number of concerns and scientific disagreements that we have with Reviewer #1. A majority of these have already been addressed in the first Reply. Thus, readers may refer to this first response.

Regarding benthic foraminifera, Reviewer #1 states "seen the diversity of benthic foraminifera and their different behavior in different settings, often they are used to back up other proxies". This is strongly disputable, since the diversity of benthic foraminifera is rather a strength than a weakness, since this allows to track minute environmental changes (see for example the reviews by Jorissen et al., 2007 or Gooday, 2014). The papers of Rüggeberg et al. (2007), Margreth et al. (2011); Stalder et al. (2015; 2018); Fentimen et al. (2020) all use benthic foraminifera as a key and main proxy in cold-water coral environments. The papers of Fink et al. (2012) and Matos et al. (2017) indeed use benthic foraminifera rather as a supporting proxy. Yet, this is due to a choice in investigation strategy rather than a scientific assessment that benthic foraminifera can only be used as a supporting proxy (see again for counter-examples, Gooday et al., 2014). Moreover, the low number of investigated samples in the studies of Fink et al (2012) and Matos et al. (2017), may reduce in these two studies the confidence in conclusions drawn from benthic foraminiferal assemblages. However, the present study presents a consequent number of samples (92, 1 sample every 10 cm), hence benthic foraminiferal assemblages can be used as a main paleoenvironmental proxy. Thus, the comment made by Reviewer #1, like other comments made (see previous reply) is a personal opinion and is not supported by the literature on the subject.

Another comment concerning macrofaunal abundances is in our opinion over-critical: Reviewer #1 states: "That the authors are looking for glacial - interglacial differences". We at no moment have been looking for these differences; however based on the

present stratigraphy (which may or may not be subject to change after addition of supplementary coral ages) it is clear that coral and bryozoan show a relation to interglacial - glacial variations. This should be clear to any neutral reader of this preprint.

Finally, as suggested by both Reviewers, some interpretations need to be clarified and down-toned. We fully agree with this and will take into account these comments in a revised version of this manuscript. However, we believe it is out of place to accuse us of "fitting our data to our interpretations". This is an important accusation since it questions our scientific integrity. Although interpretations may be discussed, at no time do we try to fit the data to a hypothetic model. We believe that the scientific record of all the authors associated with this manuscript is sufficient to suggest that Reviewer #1 has made a false and over-zealous accusation.

Sincerely,

Robin Fentimen, Andres Rüggeberg, Valentin Rime, Eline Feenstra, Norbert Frank, Antonietta Rosso, David Van Rooij, Torsten Vennemann, Thierry Adatte, Irka Hajdas, Anneleen Foubert